# Non-Adversarial Imitation Learning Provably Free of Compounding Errors: The Value Flow Mechanism

Tian Xu [1]  Chenyang Wang [1]  Xiaochen Zhai [1]  Ziniu Li [2 3]  Yi-Chen Li [1]  Yang Yu [1]

## Abstract

Adversarial imitation learning (AIL) achieves high-quality imitation by mitigating compounding errors inherent to behavioral cloning (BC), yet its adversarial optimization frequently leads to training instability. A class of non-adversarial Q-based imitation learning (IL) methods, exemplified by IQ-Learn, has emerged to address this instability and is widely believed to outperform BC by leveraging online environment interactions. In this paper, we revisit IQ-Learn and prove that it in fact reduces to BC: it admits an imitation gap lower bound with quadratic dependence on the horizon and therefore remains susceptible to compounding errors. Our theoretical analysis reveals why online interactions fail to help: IQ-Learn uniformly suppresses Q-values for all actions at states not covered by demonstrations, preventing generalization beyond demonstrations. To address this fundamental limitation, we introduce Dual Q-DM, a new Q-based IL method built on Bellman constraints. Crucially, Bellman constraints drive value flow: Q-values propagate from demonstrated to unvisited states through environment dynamics, enabling generalization beyond demonstrations. We prove that Dual Q-DM is equivalent to AIL and can recover expert actions at unvisited states, thereby mitigating compounding errors. To the best of our knowledge, Dual Q-DM is the first non-adversarial IL method that is theoretically guaranteed to eliminate compounding errors. Experimental results further corroborate our theoretical findings.

[1]National Key Laboratory for Novel Software Technology and School of Artificial Intelligence, Nanjing University, China [2]The Chinese University of Hong Kong, Shenzhen [3]Shenzhen Research Institute of Big Data. Correspondence to: Yang Yu <yuy@nju.edu.cn>.

*Proceedings of the $43^{rd}$ International Conference on Machine Learning*, Seoul, South Korea. PMLR 306, 2026. Copyright 2026 by the author(s).

## 1. Introduction

Imitation learning (IL) trains policies from expert demonstrations (Osa et al., 2018), serving as a fundamental paradigm across diverse domains including large language models (Ouyang et al., 2022) and generalist embodied agents (Black et al., 2024).

Behavioral cloning (BC) and adversarial imitation learning (AIL) are two conventional IL approaches. BC (Pomerleau, 1991) directly applies supervised learning to demonstrations but cannot infer expert actions at unvisited states, leading to compounding errors that scale quadratically with the horizon (Ross & Bagnell, 2010). AIL (Abbeel & Ng, 2004; Syed & Schapire, 2007; Ho & Ermon, 2016) takes a fundamentally different approach by matching state-action distributions through adversarial training. A key component is applying reinforcement learning (RL) to maximize an adversarially learned reward with online environment interactions. In this way, AIL can generalize beyond demonstrations (Reddy et al., 2019; Ghasemipour et al., 2019) and achieve an imitation gap that scales only linearly with the horizon (Xu et al., 2020; Rajaraman et al., 2020), effectively mitigating compounding errors. However, AIL often suffers from training instability due to adversarial optimization (Reddy et al., 2019; Orsini et al., 2021).

To avoid this instability, a wide class of non-adversarial Q-based IL methods (Kostrikov et al., 2020; Garg et al., 2021; Al-Hafez et al., 2023; Karimi & Ebadzadeh, 2025; Li et al., 2025) has emerged that directly optimizes a single maximization objective over Q-functions. Existing literature reports that these methods, exemplified by IQ-Learn (Garg et al., 2021), achieve good empirical performance in certain benchmarks, making them increasingly popular alternatives to AIL. Because their objectives are derived from those of AIL, the prevailing view (Kostrikov et al., 2020; Garg et al., 2021; Li et al., 2025) holds that Q-based IL inherits AIL's ability to mitigate compounding errors. Some works further attribute this capability to the use of online interactions (Kostrikov et al., 2020; Garg et al., 2021; Al-Hafez et al., 2023), widely considered the key mechanism by which AIL resolves compounding errors. Despite their growing popularity, the theoretical foundations of Q-based IL remain largely underdeveloped. To our knowledge, only Moulin

*Table 1.* Comparison of representative classes of IL methods. Here "RL-related Design" refers to whether the method involves an RL-related algorithmic component to solve sequential distribution matching. AIL directly employs a standard RL procedure, and Dual Q-DM involves Bellman constraints, which can be viewed as an analogue of Bellman equations in standard RL.

| Method | Non-Adversarial Optimization | Environment Interactions | RL-related Design | Generalization Beyond Demos | Resolves Compounding Errors |
|---|---|---|---|---|---|
| BC | ✓ | ✗ | ✗ | ✗ | ✗ |
| IQ-Learn | ✓ | ✓ | ✗ | ✗ | ✗ |
| AIL | ✗ | ✓ | ✓ | ✓ | ✓ |
| Dual Q-DM (Ours) | ✓ | ✓ | ✓ | ✓ | ✓ |

et al. (2025) provided a sample complexity analysis for a Q-based method. Therefore, the following fundamental question remains open:

> **Question**
>
> Which algorithmic mechanisms are necessary for Q-based IL to provably generalize beyond demonstrations and eliminate compounding errors?

### 1.1. Our Contribution

This paper investigates this question through systematic theoretical analysis of Q-based IL.

We begin by revisiting IQ-Learn (Garg et al., 2021), a representative Q-based approach claimed to outperform BC by utilizing online environment interactions. Surprisingly, through a re-parameterization argument, we prove that IQ-Learn's recovered policy is nearly equivalent to the BC policy (Theorem 1). Furthermore, we establish that IQ-Learn suffers an imitation gap lower bound of $\Omega(H^2)$ with $H$ being the horizon (Corollary 1), indicating it still experiences compounding errors. Theoretical analysis reveals why online interactions fail to provide benefits: they are merely used to estimate an expected log-partition term that uniformly suppresses Q-values across the action space, preventing generalization beyond demonstrations.

To address this fundamental limitation, we introduce a primal-dual framework for distribution matching, yielding a new Q-based IL method: Dual Q-DM (Dual Q-function Distribution Matching). Specifically, starting from primal distribution matching over occupancy measures, we leverage Lagrangian duality to derive dual distribution matching over Q-functions. Compared with IQ-Learn, the key mechanism in Dual Q-DM is the Bellman constraints, which drive value flow: Q-values propagate from successor states to current states through environment dynamics. Theoretically, we prove that Dual Q-DM's learned policy is equivalent to AIL's policy (Theorem 2), naturally inheriting AIL's ability to mitigate compounding errors. To the best of our knowledge, Dual Q-DM is the first non-adversarial IL method provably free of compounding errors.

Furthermore, for a broad class of IL instances, we prove that Dual Q-DM can identify expert actions at uncovered states, achieving generalization beyond demonstrations (Proposition 1). Crucially, our theoretical analysis reveals that the key mechanism underlying this generalization is value flow, which propagates Q-values from demonstrated to unvisited states through environment dynamics, equipping Dual Q-DM to generalize beyond demonstrations and mitigate compounding errors. Based on these theoretical findings, we present a systematic comparison of major IL algorithms in Table 1, showing that RL-related design provides the key separation between methods that resolve compounding errors and those that do not. Finally, experimental results on the MuJoCo benchmark corroborate our theoretical findings.

## 2. Preliminaries

**Markov Decision Process.** We consider episodic Markov Decision Processes (MDPs) represented by the tuple $\mathcal{M} = (\mathcal{S}, \mathcal{A}, P, r^\star, H, \rho)$, where $\mathcal{S}$ and $\mathcal{A}$ denote the state and action spaces, respectively, $H$ is the planning horizon, and $\rho$ is the initial state distribution. Here, $P : \mathcal{S} \times \mathcal{A} \to \Delta(\mathcal{S})$ is the transition function where $\Delta(\mathcal{S})$ denotes the set of distributions over $\mathcal{S}$, and $r^\star : \mathcal{S} \times \mathcal{A} \to [0, 1]$ is the true reward function. We assume without loss of generality that the state space is layered such that $\mathcal{S} = \mathcal{S}_1 \cup \mathcal{S}_2 \cup \cdots \cup \mathcal{S}_H$, where $\mathcal{S}_h$ is the set of states reachable at step $h$, and $\mathcal{S}_h \cap \mathcal{S}_{h'} = \emptyset$ for $h \neq h'$. A policy $\pi : \mathcal{S} \to \Delta(\mathcal{A})$ maps states to action distributions. We evaluate policy performance using the expected cumulative reward: $V^\pi := \mathbb{E}[\sum_{h=1}^{H} r^\star(s_h, a_h)|s_1 \sim \rho, a_h \sim \pi(\cdot|s_h), s_{h+1} \sim P_h(\cdot|s_h, a_h), \forall h \in [H]]$, where the expectation is taken over the trajectory distribution induced by $\pi$. Another important concept is the state-action visitation distribution $d_h^\pi(s, a) := \mathbb{P}^\pi(s_h = s, a_h = a)$, which quantifies the probability of visiting $(s, a)$ at step $h$ by following $\pi$.

**Imitation Learning.** The goal of IL is to acquire a high-quality policy *without* access to the reward function $r^\star$. To achieve this, we assume access to an expert dataset $\mathcal{D}^{\mathrm{E}} = \{\tau^i = (s_1^i, a_1^i, s_2^i, a_2^i, \ldots, s_H^i, a_H^i); \tau^i \sim \pi^{\mathrm{E}}\}_{i=1}^N$, which is collected by an expert policy $\pi^{\mathrm{E}}$. The learner uses this dataset $\mathcal{D}^{\mathrm{E}}$ to learn a policy that mimics the expert's

behavior. We measure imitation quality using the *imitation gap* (Abbeel & Ng, 2004; Ross & Bagnell, 2010; Rajaraman et al., 2020), defined as $V^{\pi^{\mathrm{E}}} - V^{\widehat{\pi}}$, where $\widehat{\pi}$ is the learned policy. Essentially, we hope that the learned policy can perfectly mimic the expert such that the imitation gap is small.

**Behavioral Cloning.** Behavioral cloning (BC) (Pomerleau, 1991) performs maximum likelihood estimation (MLE) to mimic the expert.

$$\pi^{\mathrm{BC}} = \underset{\pi \in \Pi}{\operatorname{argmax}} \sum_{i=1}^{N} \sum_{h=1}^{H} \log\left(\pi(a_h^i | s_h^i)\right). \tag{1}$$

Here $\Pi$ is the set of all policies. BC is trained purely on demonstrations and thus cannot infer the expert action at states not covered by demonstrations. Due to this limitation, BC suffers from compounding errors and has been proven to have an imitation gap bound of $\mathcal{O}(H^2)$ (Ross & Bagnell, 2010; Xu et al., 2020; Rajaraman et al., 2020), which increases quadratically with the horizon.

**Adversarial Imitation Learning.** Adversarial imitation learning (AIL) imitates expert behavior through an adversarial process. This paper considers the maximum entropy AIL approach (Ho & Ermon, 2016), which can be formulated as the following minimax objective.

$$\min_{\pi \in \Pi} \left( \max_{r \in \mathcal{R}} \mathbb{E}_{\tau \sim \mathcal{D}^{\mathrm{E}}} \left[ \sum_{h=1}^{H} r(s_h, a_h) \right] \right.$$
$$\left. - \mathbb{E}_{\tau \sim \pi} \left[ \sum_{h=1}^{H} \left( r(s_h, a_h) + \alpha \mathcal{H}(\pi(\cdot | s_h)) \right) \right] \right). \tag{2}$$

Here $\mathcal{R} = \{r : \mathcal{S} \times \mathcal{A} \to [0,1]\}$ is the class of bounded reward functions, $\mathcal{H}(\pi(\cdot|s)) = \mathbb{E}_{a \sim \pi(\cdot|s)}[-\log(\pi(a|s))]$ denotes the entropy and $\alpha$ denotes the coefficient. Unless explicitly stated otherwise, this paper uses $\alpha = 1$ by default. As shown in (Ho & Ermon, 2016; Ghasemipour et al., 2019), AIL essentially minimizes the state-action distribution discrepancy with the expert.

$$\min_{\pi \in \Pi} \sum_{h=1}^{H} \left( D_{\mathrm{TV}}(\widehat{d_h^{\pi^{\mathrm{E}}}}, d_h^{\pi}) - \overline{\mathcal{H}}(d_h^{\pi}) \right) \tag{3}$$

Here $D_{\mathrm{TV}}(p, q) = (1/2) \cdot \sum_{x \in \mathcal{X}} |p(x) - q(x)|, \forall p, q \in \Delta(\mathcal{X})$ denotes the total variation distance, $\widehat{d_h^{\pi^{\mathrm{E}}}}$ is the empirical expert's state-action distribution from $\mathcal{D}^{\mathrm{E}}$ and $\overline{\mathcal{H}}(d_h^{\pi}) = \mathbb{E}_{(s,a) \sim d_h^{\pi}}[-\log(d_h^{\pi}(s,a)/(\sum_{a \in \mathcal{A}} d_h^{\pi}(s,a)))]$ is the entropy of state-action distribution. Due to the global state-action distribution matching principle, AIL addresses the compounding errors issue in BC and achieves an imitation gap bound of $\mathcal{O}(H)$ (Agarwal et al., 2019; Xu et al., 2020) with only linear dependence on $H$.

## 3. Revisiting Inverse Soft Q-Learning

To investigate the central question raised in the Introduction, we revisit inverse soft Q-learning (IQ-Learn) (Garg et al., 2021), a representative Q-based IL method claimed to alleviate compounding errors. We first review its formulation, then demonstrate that despite its Q-based appearance, IQ-Learn essentially reduces to BC and consequently still suffers from compounding errors.

### 3.1. An Introduction to Inverse Soft Q-Learning

To circumvent the adversarial optimization in AIL, Garg et al. (2021) proposed IQ-Learn, which directly optimizes a single maximization objective over Q-functions. Starting from the AIL objective in Eq. (2), Garg et al. (2021) derived the following single maximization objective[1]:

$$\max_{Q \in \mathcal{Q}} \mathbb{E}_{\tau \sim \mathcal{D}^{\mathrm{E}}} \left[ \sum_{h=1}^{H} Q(s_h, a_h) - (\mathrm{LSE}\, Q)(s_{h+1}) \right]$$
$$- \mathbb{E}_{s_1 \sim \rho} \left[ (\mathrm{LSE}\, Q)(s_1) \right]. \tag{4}$$

Here, $(\mathrm{LSE}\, Q)(s) := \log(\sum_{a \in \mathcal{A}} \exp(Q(s,a)))$ denotes the log-partition function and $\mathcal{Q} = \{Q : \mathcal{S} \times \mathcal{A} \to [0,C]\}$ denotes the class of bounded Q-functions. To align with the AIL objective in Eq. (2), we focus on IQ-Learn instantiated with the total variation distance, which corresponds to choosing $\phi(x) = x$ in Eq. (9) of Garg et al. (2021). Empirical results in Figure 4 and the discussion in Section C.6 indicate that the specific choice of divergence has minimal effect on performance. After obtaining $\widehat{Q}$ by solving Eq. (4), IQ-Learn derives the corresponding softmax policy $\pi_{\widehat{Q}} = \mathrm{softmax}(\widehat{Q})$.

$$\pi_{\widehat{Q}}(a|s) = \mathrm{softmax}(\widehat{Q}(s, \cdot)) \propto \exp(\widehat{Q}(s,a)). \tag{5}$$

From Eq. (4), IQ-Learn seeks a Q-function that assigns high values to expert actions in $\mathcal{D}^{\mathrm{E}}$ through the term $Q(s_h, a_h)$, while suppressing non-expert actions through $(\mathrm{LSE}\, Q)(s_{h+1})$. As a result, $\pi_{\widehat{Q}}$ places higher probability mass on expert actions, thereby imitating expert behavior. Garg et al. (2021) report that IQ-Learn achieves good empirical performance in certain benchmarks.

Crucially, like AIL, IQ-Learn incorporates online environment interactions when optimizing Eq. (4). Specifically, these interactions provide an accurate estimate of the final term in Eq. (4) through two approaches: (i) by directly using online initial states, or (ii) by exploiting online trajectories via the identity that $\forall \pi, \mathbb{E}_{s_1 \sim \rho}[(\mathrm{LSE}\, Q)(s_1)] =$

---

[1] IQ-Learn was originally formulated in the infinite-horizon setting. Here we translate Eq. (9) of Garg et al. (2021) into our finite-horizon formulation. The main results of this paper also extend to the infinite-horizon setting; please refer to Appendix D for details.

$\mathbb{E}_{\tau \sim \pi}[\sum_{h=1}^{H} (\text{LSE } Q)(s_h) - (\text{LSE } Q)(s_{h+1})]$. In summary, IQ-Learn is derived from AIL and similarly leverages online interactions. This has led to the conventional wisdom that by exploiting environment interactions, IQ-Learn transcends pure BC and mitigates compounding errors. For instance, Al-Hafez et al. (2023); Wulfmeier et al. (2024); Karimi & Ebadzadeh (2025) characterize IQ-Learn as a variant of AIL with implicit reward. In the next subsection, we examine this prevailing belief through theoretical analysis and demonstrate that IQ-Learn actually reduces to BC.

## 3.2. Inverse Soft Q-Learning Reduces to Behavioral Cloning

In this subsection, we conduct a theoretical analysis of IQ-Learn through a re-parameterization argument. Specifically, rather than viewing $Q(s_h, a_h) - (\text{LSE } Q)(s_{h+1})$ as a Bellman-error-related term as in (Garg et al., 2021), we apply a telescoping transformation to Eq. (4) and re-parameterize $Q$ in terms of $\pi$ via Eq. (5).

$$
\begin{aligned}
& \mathbb{E}_{\tau \sim \mathcal{D}^{\text{E}}} \left[ \sum_{h=1}^{H} Q(s_h, a_h) - (\text{LSE } Q)(s_{h+1}) \right] \\
& \quad - \mathbb{E}_{s_1 \sim \rho} \left[ (\text{LSE } Q)(s_1) \right] \\
& = \mathbb{E}_{(s_1, a_1) \sim \mathcal{D}^{\text{E}}} \left[ Q(s_1, a_1) \right] - \mathbb{E}_{s_1 \sim \rho} \left[ \text{LSE}(Q)(s_1) \right] \\
& \quad + \mathbb{E}_{\tau \sim \mathcal{D}^{\text{E}}} \left[ \sum_{h=2}^{H} \underbrace{(Q(s_h, a_h) - \text{LSE}(Q)(s_h))}_{\log(\pi_Q(a_h|s_h))} \right].
\end{aligned}
$$

Strikingly, this reveals that the IQ-Learn objective coincides with the BC objective for time steps $h = 2, \cdots, H$. Further analysis shows that at the initial time step, the IQ-Learn policy assigns a uniform distribution over actions at unvisited states—identical to BC. Formally, we characterize the optimal solution to IQ-Learn as follows.

**Theorem 1.** *Suppose that $\widehat{Q}$ is the optimal solution to IQ-Learn in Eq. (4) and $\pi_{\widehat{Q}}$ is the derived policy in Eq. (5). Assume the softmax policy class realizes the BC policy, i.e., $\pi^{\text{BC}} \in \{\pi_Q : Q \in \mathcal{Q}\}$. Then the following holds:*

- $\forall 2 \leq h \leq H, \forall s_h \in \mathcal{S}_h,\ \pi_{\widehat{Q}}(\cdot|s_h) = \pi^{\text{BC}}(\cdot|s_h).$

- *In the initial time step $h = 1$,*

$$
\forall s_1 \notin \mathcal{D}^{\text{E}}, \pi_{\widehat{Q}}(\cdot|s_1) = \pi^{\text{BC}}(\cdot|s_1) = \text{Unif}(\mathcal{A}),
$$

*where $\text{Unif}(\mathcal{A})$ denotes a uniform distribution over $\mathcal{A}$.*

The detailed proof is presented in Appendix B.1.

**Remark 1.** *Theorem 1 indicates that IQ-Learn is almost equivalent to BC. At all time steps except the initial one, the IQ-Learn policy is identical to the BC policy. At the initial time step, both policies follow a uniform distribution at* states uncovered by $\mathcal{D}^{\text{E}}$. *Collectively, these results demonstrate that neither IQ-Learn nor BC can recover expert behavior at states outside the demonstrations. This result challenges the prevailing belief that IQ-Learn resolves compounding errors of BC.*

**Remark 2.** *We now explain why IQ-Learn does not benefit from online environment interactions to go beyond BC. Recall that in IQ-Learn, online interactions are used to obtain an accurate estimate of the term $\mathbb{E}_{s_1 \sim \rho}[(\text{LSE } Q)(s_1)] = \mathbb{E}_{s_1 \sim \rho}[\log(\sum_{a \in \mathcal{A}} \exp(Q(s_1, a)))]$. This term is monotonically increasing w.r.t $Q(s, a)$ for all actions, meaning it uniformly suppresses Q-values across the entire action space. As a result, it provides no discriminative signal for identifying expert actions from non-expert ones and thus cannot facilitate generalization beyond BC.*

**Remark 3.** *The re-parameterization argument for proving Theorem 1 extends beyond IQ-Learn. As shown in Appendix B.2, it can similarly reveal the connection between another Q-based IL approach ValueDICE (Kostrikov et al., 2020) and BC.*

Building on Theorem 1, we further demonstrate that IQ-Learn still suffers from the compounding errors issue.

**Corollary 1.** *Suppose that $\widehat{Q}$ is the optimal solution to IQ-Learn in Eq. (4) and $\pi_{\widehat{Q}}$ is the derived policy in Eq. (5). There exists an IL instance $(\mathcal{M}, \pi^{\text{E}})$ such that*

$$
V^{\pi^{\text{E}}} - \mathbb{E}\left[ V^{\widehat{\pi}} \right] \geq \Omega\left( \min\left\{ H, \frac{H^2}{N} \right\} \right).
$$

*Here the expectation is taken over the randomness of $\mathcal{D}^{\text{E}}$.*

The detailed proof is presented in Section B.3.

**Remark 4.** *Corollary 1 establishes that IQ-Learn incurs an imitation gap scaling quadratically with the horizon in the worst case, confirming the presence of compounding errors.*

**Remark 5.** *The proof idea is outlined as follows. Following (Rajaraman et al., 2020), we consider a hard IL instance where the agent receives no further rewards after taking a non-expert action. Theorem 1 establishes that IQ-Learn cannot recover the expert action at states not covered by demonstrations. By carefully characterizing the probability of visiting such uncovered states, we prove that the IQ-Learn policy must suffer a $\Omega(H^2)$ imitation gap.*

We conclude this section by discussing why IQ-Learn, despite being derived from the AIL objective, produces an optimization objective that is almost equivalent to BC. Through a careful step-by-step analysis of the IQ-Learn derivation, we trace this discrepancy to the change-of-variables step, where overlooking certain constraints leads to an inequivalent formulation. Please refer to Appendix B.4 for a detailed discussion of this derivation gap.

# 4. Truly Q-based Distribution Matching via Value Flow

The previous section demonstrates that IQ-Learn reduces to BC and thus fails to eliminate compounding errors. To address this fundamental limitation, we develop a genuinely Q-based distribution matching approach and demonstrate that it provably eliminates compounding errors and generalizes beyond demonstrations.

## 4.1. Primal-Dual Framework for Distribution Matching

AIL relies on the state-action distribution matching principle to provably eliminate compounding errors (Xu et al., 2020). Our goal is therefore to faithfully inherit AIL's theoretical guarantees by rigorously implementing this principle without deviation. To this end, we introduce a primal-dual framework that formalizes distribution matching as constrained optimization, which allows us to establish theoretical equivalence with AIL. We present three main formulations in this framework and defer the derivation to Section C.1.

**Primal Distribution Matching (Primal DM).** We begin by transforming the distribution matching objective from policy space (i.e., Eq. (3)) into occupancy measure space.

$$\min_d \sum_{h=1}^{H} \left( D_{\text{TV}}(\widehat{d_h^{\pi^{\text{E}}}}, d_h) - \overline{H}(d_h) \right),$$

$$\text{s.t.} \sum_{a \in \mathcal{A}} d_1(s_1, a) = \rho(s_1), \forall s_1 \in \mathcal{S}_1, \qquad (6)$$

$$\sum_{(s_h, a) \in \mathcal{S}_h \times \mathcal{A}} d_h(s_h, a) P_h(s'|s_h, a) = \sum_{a \in \mathcal{A}} d_{h+1}(s', a),$$

$$\forall 1 \leq h \leq H - 1, \forall s' \in \mathcal{S}_{h+1},$$

$$d_h(s_h, a) \geq 0, \forall (h, s_h, a) \in [H] \times \mathcal{S}_h \times \mathcal{A}.$$

Primal DM performs state-action distribution matching directly over occupancy measures under the constraints that $d$ is a feasible state-action distribution (Puterman, 2014).

**Dual V-Function Distribution Matching (Dual V-DM).** Notice that Eq. (6) is a convex program with a strictly convex objective (Ho & Ermon, 2016) and linear polytope constraints. Thus, we can apply Lagrangian duality (Boyd et al., 2004) to Eq. (6), yielding the following dual V-function distribution matching.

$$\max_{r,V} \mathbb{E}_{\tau \sim \mathcal{D}^{\text{E}}} \left[ \sum_{h=1}^{H} r(s_h, a_h) \right] - \mathbb{E}_{s_1 \sim \rho} \left[ V(s_1) \right], \qquad (7)$$

$$\text{s.t.} V(s_h) = \text{LSE}(r + PV)(s_h), \forall (h, s_h) \in [H] \times \mathcal{S}_h,$$

$$0 \leq r(s_h, a) \leq 1, \forall (h, s_h, a) \in [H] \times \mathcal{S}_h \times \mathcal{A}.$$

Here $(r + PV)(s, a) = r(s, a) + \mathbb{E}_{s' \sim P(\cdot|s,a)}[V(s')]$ denotes the Bellman backup. The dual problem optimizes over a reward-value pair $(r, V)$ and aims to maximize the

gap between the expert's value and the learner's value, in alignment with AIL in Eq. (2). The constraints serve complementary roles: the first requires $V$ to satisfy the soft Bellman optimality equation with respect to $r$, ensuring $V$ is the corresponding soft optimal value function; the second enforces boundedness of $r$, which arises from the dual representation of total variation distance.

**Dual Q-Function Distribution Matching (Dual Q-DM).** Recall that our goal is to develop a Q-based distribution matching approach. To this end, we introduce a Q-function $Q(s, a) := r(s, a) + \mathbb{E}_{s' \sim P(\cdot|s,a)}[V(s')]$ and perform the change-of-variables on Eq. (7), yielding the following dual Q-function distribution matching.

$$\max_Q \mathbb{E}_{\tau \sim \mathcal{D}^{\text{E}}} \left[ \sum_{h=1}^{H} Q(s_h, a_h) - \mathbb{E}_{s' \sim P(\cdot|s_h, a_h)} \left[ \text{LSE}(Q)(s') \right] \right],$$

$$- \mathbb{E}_{s_1 \sim \rho} \left[ \text{LSE}(Q)(s_1) \right] \qquad (8)$$

$$\text{s.t.} \ 0 \leq Q(s_h, a) - \mathbb{E}_{s' \sim P(\cdot|s_h, a)} \left[ \text{LSE}(Q)(s') \right] \leq 1,$$

$$\forall (h, s_h, a) \in [H] \times \mathcal{S}_h \times \mathcal{A}. \qquad (9)$$

We compare this formulation with IQ-Learn in Eq. (4). While Dual Q-DM shares a similar objective to IQ-Learn, the key distinguishing feature is Bellman constraints in Eq. (9), which act as an inequality analogue of the Bellman equations. Crucially, Bellman constraints drive value flow: Q-values at successor states propagate into the current Q-values through environment dynamics, i.e., $Q(s_h, a) \in [\mathbb{E}_{s' \sim P(\cdot|s_h, a)}[\text{LSE}(Q)(s')], \mathbb{E}_{s' \sim P(\cdot|s_h, a)}[\text{LSE}(Q)(s')] + 1]$. Later we will demonstrate that value flow is the key mechanism for generalization beyond demonstrations.

From another perspective, Bellman constraints provide the mechanism for achieving multi-step state-action distribution matching—the same role that the direct RL procedure plays in AIL. On one hand, AIL explicitly learns a reward function and thus can apply a direct RL procedure (e.g., Bellman equations) to implement multi-step distribution matching. On the other hand, Dual Q-DM bypasses explicit reward learning and instead applies Bellman constraints for achieving distribution matching.

*Table 2.* Comparison between AIL and Dual Q-DM.

| Method | Imitation Principle | Explicit Reward | RL-related Mechanism |
|---|---|---|---|
| AIL | Distribution Matching | ✓ | Direct RL (e.g., Bellman equations) |
| Dual Q-DM | Distribution Matching | ✗ | Bellman constraints |

Interestingly, Bellman constraints can also be recovered by fixing the flaw in IQ-Learn's derivation: the constraints omitted in its change-of-variables step are precisely the Bellman constraints introduced here; see Appendix B.4 for

a detailed analysis. Rather than following this derivation path, we adopt the primal-dual framework to rigorously establish the following theoretical equivalence.

**Theorem 2.** *Suppose that $\widetilde{Q}$ is the optimal solution to Dual Q-DM in Eq. (8) and $\pi_{\widetilde{Q}} = \mathrm{softmax}(\widetilde{Q})$ is the derived policy. Then $\pi_{\widetilde{Q}}$ is the optimal solution to AIL in Eq. (3).*

$$\pi_{\widetilde{Q}} \in \operatorname*{argmin}_{\pi \in \Pi} \sum_{h=1}^{H} \left( D_{\mathrm{TV}}(\widehat{d_h^{\pi^{\mathrm{E}}}}, d_h^{\pi}) - \overline{\mathcal{H}}(d_h^{\pi}) \right).$$

The detailed proof is presented in Appendix C.2.

**Remark 6.** *Theorem 2 shows that the policy obtained via Dual Q-DM is the optimal solution to AIL, thereby inheriting AIL's key advantage of mitigating compounding errors. The proof proceeds by establishing equivalence among three formulations—Primal DM, Dual V-DM, and Dual Q-DM—which together constitute a comprehensive primal-dual framework for distribution matching. We refer the reader to Appendix C.2 for details.*

### 4.2. Value Flow: How Bellman Constraints Enable Generalization

In this subsection, we demonstrate that value flow, the propagation of learning signals from demonstrated to unvisited states through Bellman constraints, is the central mechanism underlying generalization beyond demonstrations. We begin by defining a class of MDPs called transition-discriminative MDPs (TD MDPs), which we use to characterize when value flow enables effective generalization.

**Definition 1** (TD MDPs). *Consider a deterministic expert policy $\pi^{\mathrm{E}}$. We define the set of expert-reachable states as $\mathcal{S}_{h+1}^{\mathrm{E}} = \{s_{h+1} : \exists s_h \in \mathcal{S}_h, P(s_{h+1}|s_h, \pi^{\mathrm{E}}(s_h)) > 0\}$. A transition-discriminative MDP satisfies the following two properties:*

- $\forall h \in [H-1], s_h \in \mathcal{S}_h, s_{h+1}^{\mathrm{E}} \in \mathcal{S}_{h+1}^{\mathrm{E}},$

  $P(s_{h+1}^{\mathrm{E}}|s_h, \pi^{\mathrm{E}}(s_h)) > P(s_{h+1}^{\mathrm{E}}|s_h, a), \forall a \neq \pi^{\mathrm{E}}(s_h).$

- $\forall h \in [H-1], s_{h+1}^{\mathrm{E}} \in \mathcal{S}_{h+1}^{\mathrm{E}}, a \in \mathcal{A},$

  $P(s_{h+1}^{\mathrm{E}}|s_h^{\mathrm{E}}, a) \geq P(s_{h+1}^{\mathrm{E}}|s_h, a), \forall s_h^{\mathrm{E}} \in \mathcal{S}_h^{\mathrm{E}}, s_h \notin \mathcal{S}_h^{\mathrm{E}}.$

These properties establish transition discriminability by ensuring that both expert actions and expert-type states lead to expert-type successor states with higher probability. This condition is natural, as it simply formalizes the intuition that expert behavior exhibits internal consistency in its transition patterns. In TD MDPs, we investigate the generalization performance of Dual Q-DM and IQ-Learn by characterizing the learned Q-values at states uncovered by demonstrations.

**Proposition 1.** *Suppose that $\widetilde{Q}$ and $\widehat{Q}$ are Q-functions learned by Dual Q-DM via Eq. (8) and IQ-Learn via Eq. (4), respectively. For deterministic expert policies and TD MDPs, when $N \geq 1$, for any timestep $h \in [H-1]$ and any states uncovered by demonstrations $s_h \notin \mathcal{D}^{\mathrm{E}}$, we have*

$$\widetilde{Q}(s_h, \pi^{\mathrm{E}}(s_h)) > \widetilde{Q}(s_h, a), \forall a \neq \pi^{\mathrm{E}}(s_h),$$
$$\widehat{Q}(s_h, \pi^{\mathrm{E}}(s_h)) = \widehat{Q}(s_h, a), \forall a \neq \pi^{\mathrm{E}}(s_h).$$

The proof of Proposition 1 is presented in Appendix C.3.

**Remark 7.** *Proposition 1 reveals a fundamental difference in generalization capability stemming from the presence or absence of Bellman constraints. Dual Q-DM, equipped with Bellman constraints, successfully identifies expert actions at states unvisited by demonstrations for the first $H-1$ timesteps. In contrast, IQ-Learn, lacking Bellman constraints, assigns uniform Q-values across all actions at unseen states and thus cannot recover expert behavior beyond demonstrations.*

**The Value Flow Mechanism.** The proof of Proposition 1 reveals that value flow is the key mechanism underlying generalization. Specifically, via a dynamic programming analysis, we show that value flow, driven by Bellman constraints, propagates Q-values from demonstrated to unvisited states through environment dynamics, thereby enabling generalization beyond demonstrations. Figure 1 provides empirical evidence and illustration of this mechanism.

We first present a technical lemma useful for the proof. Lemma 1 shows that the Q-function learned by Dual Q-DM saturates Bellman constraints in a structured manner: it achieves the upper bound on certain demonstrated state–action pairs and the lower bound on all unvisited ones.

$$\exists (s_h^{\mathrm{E}}, a_h^{\mathrm{E}}) \in \mathcal{D}^{\mathrm{E}}, r_{\widetilde{Q}}(s_h^{\mathrm{E}}, a_h^{\mathrm{E}}) = 1,$$
$$\forall (s_h, a_h) \notin \mathcal{D}^{\mathrm{E}}, r_{\widetilde{Q}}(s_h, a_h) = 0. \tag{10}$$

$r_{\widetilde{Q}}(s_h, a_h) := \widetilde{Q}(s_h, a_h) - \mathbb{E}_{s' \sim P(\cdot|s_h, a_h)}[\mathrm{LSE}(\widetilde{Q})(s')]$ can be regarded as the reward function induced by $\widetilde{Q}$.

**Step I: Value Flow Source.** We characterize the Q-function at the terminal step. Since Q-values at the terminal step $H$ equal single-step rewards, Eq. (10) implies that they are high on visited pairs and low on unvisited ones. Formally, $\exists (s_H^{\mathrm{E}}, a_H^{\mathrm{E}}) \in \mathcal{D}^{\mathrm{E}}, \forall (s_H, a_H) \notin \mathcal{D}^{\mathrm{E}}, \widetilde{Q}(s_H^{\mathrm{E}}, a_H^{\mathrm{E}}) > \widetilde{Q}(s_H, a_H)$. By defining the value function $\widetilde{V}(s) := \mathrm{LSE}(\widetilde{Q})(s)$, we derive that $\exists s_H^{\mathrm{E}} \in \mathcal{D}^{\mathrm{E}}, \forall s_H \notin \mathcal{D}^{\mathrm{E}}, \widetilde{V}(s_H^{\mathrm{E}}) > \widetilde{V}(s_H)$, meaning the values on visited states are higher than those on unvisited states. As visualized in Figure 1(c), demonstration states acquire substantially higher V-values than unvisited states. These high V-values serve as the source from which value flows to unvisited states in the subsequent steps.

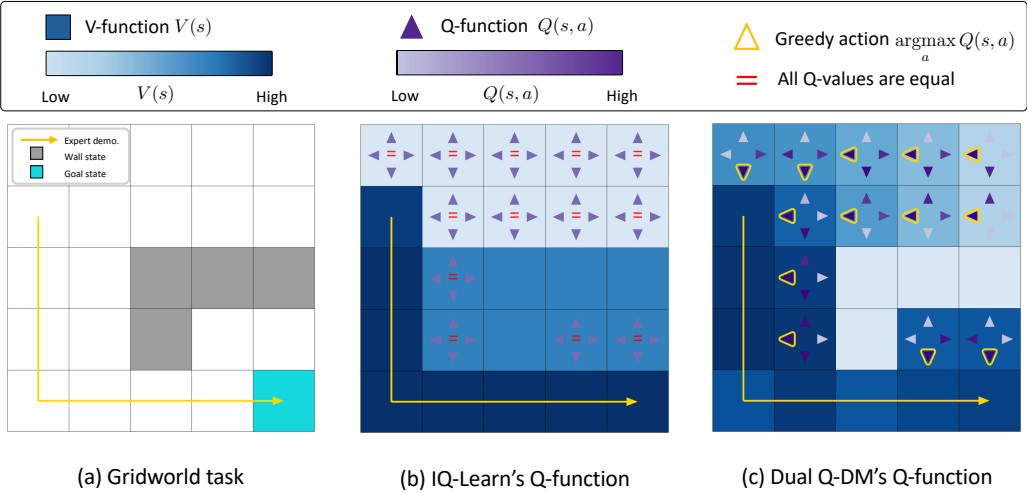

*Figure 1.* Illustration of the learned Q-functions on a Gridworld task. (a): The Gridworld environment with wall states (gray), goal state (cyan), and one demonstration trajectory (yellow). (b): IQ-Learn's Q-function: visited states acquire high V-values (dark blue), but unvisited states exhibit equal Q-values across all actions (red equal signs), confirming that IQ-Learn cannot identify expert actions beyond demonstrations (Theorem 1). (c): Dual Q-DM's Q-function: Bellman constraints enable value flow from demonstrated to unvisited states, yielding a structured Q-function whose greedy action (yellow-outlined triangle) at unvisited states correctly points toward the expert trajectory, demonstrating generalization beyond demonstrations (Proposition 1).

**Step II: Value Flow Propagation.** We characterize the Q-function at step $H - 1$, establishing the base case for induction. $\widetilde{Q}$ satisfies the soft Bellman optimality equation:

$$\widetilde{Q}(s_{H-1}, a_{H-1}) = r_{\widetilde{Q}}(s_{H-1}, a_{H-1}) \\ + \mathbb{E}_{s' \sim P(\cdot|s_{H-1}, a_{H-1})}\left[\widetilde{V}(s')\right],$$

decomposing the Q-value into a current reward and a future value. This decomposition exposes the value flow mechanism: the high V-values at demonstrated successor states flow into the current Q-value through $\mathbb{E}_{s'}[\widetilde{V}(s')]$, even when the current state itself is unvisited by demonstrations.

To see why this enables generalization, consider an unvisited expert-type pair $(s_{H-1}, \pi^{\mathrm{E}}(s_{H-1}))$. Although its current reward is zero by Eq. (10), the expert action yields a high expected future value because, in TD MDPs, expert actions preferentially transition to expert-type states. Consequently, $\widetilde{Q}(s_{H-1}, \pi^{\mathrm{E}}(s_{H-1})) > \widetilde{Q}(s_{H-1}, a), \forall a \neq \pi^{\mathrm{E}}(s_{H-1})$. Figure 1 provides empirical confirmation: in Dual Q-DM (Figure 1(c)), the greedy actions at unvisited states correctly point toward the expert trajectory, whereas in IQ-Learn (Figure 1(b)), Q-values are uniform across actions at such states.

In short, value flow ensures that the Q-value of an unobserved expert action inherits high values from its successor states, enabling generalization to unvisited state–action pairs. A parallel argument establishes that $\forall s_{H-1}^{\mathrm{E}} \in \mathcal{S}^{\mathrm{E}}, s_{H-1} \notin \mathcal{S}^{\mathrm{E}}, \widetilde{V}(s_{H-1}^{\mathrm{E}}) \geq \widetilde{V}(s_{H-1})$—that is, V-values at expert states uniformly dominate those at non-expert states.

**Step III: Recursive Value Flow.** For the inductive step,

we assume that at step $h + 1$: (i) $\widetilde{Q}(s_{h+1}, \pi^{\mathrm{E}}(s_{h+1})) > \widetilde{Q}(s_{h+1}, a), \forall a \neq \pi^{\mathrm{E}}(s_{h+1})$ and (ii) $\forall s_{h+1}^{\mathrm{E}} \in \mathcal{S}^{\mathrm{E}}, s_{h+1} \notin \mathcal{S}^{\mathrm{E}}, \widetilde{V}(s_{h+1}^{\mathrm{E}}) \geq \widetilde{V}(s_{h+1})$. By the same argument as the base case, both conclusions carry over to step $h$. Applying this induction over $h = H - 1, \ldots, 1$ establishes that Dual Q-DM correctly identifies expert actions at unvisited states throughout the entire horizon.

The reach of this mechanism extends further than it might initially appear. Even a single demonstration trajectory induces a potentially large predecessor tree under the environment dynamics—the set of all states from which the demonstrated trajectory is reachable. Recursive value flow propagates high Q-values through this tree, enabling Dual Q-DM to recover expert behavior well beyond demonstrations. Figure 1(c) confirms this effect: greedy actions correctly point toward the expert trajectory across the entire grid, including states far from the demonstration.

This dynamic programming analysis thus yields a new mechanistic understanding of Q-based IL: value flow, not merely online interactions, is the essential mechanism for generalization beyond demonstrations.

> **Takeaway:**
> Value flow, driven by Bellman constraints, propagates learning signals from demonstrated to unvisited states through environment dynamics, equipping Dual Q-DM to generalize beyond demonstrations and mitigate compounding errors.

**Practical Implementation.** We conclude this section with

a deep implementation of Dual Q-DM that incorporates Bellman constraints via penalization.

$$\max_Q \mathbb{E}_{\tau \sim \mathcal{D}^{\mathrm{E}}} \left[ \sum_{h=1}^{H} Q(s_h, a_h) - \mathrm{LSE}(Q)(s_{h+1}) \right]$$
$$- \mathbb{E}_{s_1 \sim \rho} \left[ \mathrm{LSE}(Q)(s_1) \right]$$
$$- \beta \mathbb{E}_{\tau \sim \mathcal{D}^{\mathrm{online}}} \left[ \sum_{h=1}^{H} \left( \mathrm{ReLU}(0 - Q(s_h, a_h) + \mathrm{LSE}(Q)(s_{h+1})) \right)^2 \right.$$
$$\left. + \left( \mathrm{ReLU}(Q(s_h, a_h) - \mathrm{LSE}(Q)(s_{h+1}) - 1) \right)^2 \right],$$

where $\mathrm{ReLU}(x) = \max\{0, x\}$, $\mathcal{D}^{\mathrm{online}}$ is the online replay buffer and $\beta > 0$ controls the strength of Bellman constraints. This penalty term imposes a quadratic penalty whenever the learned Q-function violates Bellman constraints, encouraging the solution to remain feasible. More importantly, it is evaluated over the *entire online dataset*, enabling value flow to propagate beyond the immediate neighborhood of demonstrations and reach a wider range of unvisited states, which is essential for broad generalization as suggested by our theoretical analysis. Algorithm 1 in Section C.5 outlines the full procedure.

Several recent Q-based IL methods (Al-Hafez et al., 2023; Karimi & Ebadzadeh, 2025) incorporate regularization terms with similar forms, interpreting them as "implicit reward regularizers" for training stability. In contrast, our work uncovers a fundamental value flow mechanism essential for generalization. A detailed comparison between different Q-based IL methods is provided in Appendix C.6.

## 5. Related Work

We briefly review relevant studies here, with a comprehensive discussion deferred to Appendix A.

**Conventional Imitation Learning.** The theoretical foundations of conventional IL methods have been extensively studied (Ross & Bagnell, 2010; Sun et al., 2019; Rajaraman et al., 2020; Xu et al., 2023; 2026a; Viano et al., 2024; Foster et al., 2024). Beyond the results discussed in the introduction, we highlight additional insights here. Rajaraman et al. (2020); Xu et al. (2021) established a fundamental $\Omega(H^2)$ lower bound for any offline IL algorithm, including BC, demonstrating that online interaction is necessary for breaking this barrier. For AIL, Xu et al. (2026a) proved a horizon-free imitation gap bound in the small-sample regime, revealing that the coupling structure in state-action distributions is critical for generalization. From the dual perspective, our work reveals a parallel mechanism in Q-based IL: value flow establishes coupling across Q-functions and serves as the key mechanism for eliminating compounding errors and generalizing beyond demonstrations.

**Q-based Imitation Learning.** Q-based IL methods have been explored empirically in numerous works (Kostrikov

et al., 2020; Garg et al., 2021; Al-Hafez et al., 2023; Karimi & Ebadzadeh, 2025; Wulfmeier et al., 2024; Li et al., 2025), yet their theoretical foundations remain poorly understood. To the best of our knowledge, only Moulin et al. (2025) have provided a theoretical analysis, introducing an offline Q-based method with a minimax formulation and establishing sample complexity guarantees under Q-function realizability. Our work differs from this in two fundamental ways: we prove that IQ-Learn effectively reduces to BC, and introduce Dual Q-DM, a novel online method with provable guarantees for eliminating compounding errors.

**Primal-Dual Perspective in Imitation Learning.** Our work relates to primal-dual approaches in IL (Viano et al., 2022; Kim et al., 2022; Ma et al., 2022; Sikchi et al., 2024). The most closely related is Sikchi et al. (2024), which derives a Q-based algorithm from primal distribution matching that can leverage supplementary data. Like IQ-Learn, their approach does not incorporate Bellman constraints. In contrast, our work develops a primal-dual framework for distribution matching where Bellman constraints play a central role. We rigorously establish the equivalence between this framework and AIL, thereby unifying Q-based and AIL methods under the primal-dual theoretical lens.

## 6. Experimental Validation

This section validates our theoretical findings through experiments. A brief overview of the experimental setup follows, with details in Appendix F. The implementation is available at https://github.com/LAMDA-RL/Dual-Q-DM.

### 6.1. Experimental Set-up

**Environment.** We evaluate our method on two types of environments. First, we use the hard tabular instance from Corollary 1, which allows us to precisely characterize compounding errors. Second, we evaluate on all 5 tasks from the MuJoCo benchmark (Todorov et al., 2012), a widely adopted continuous control benchmark in IL. Each algorithm is evaluated over three independent runs with different random seeds. Policy performance is measured using Monte Carlo estimation over 10 rollout trajectories per evaluation.

**Baselines.** We consider three categories of baselines. The first is BC (Pomerleau, 1991). For AIL, we consider DAC (Kostrikov et al., 2019) and the SOTA method HyPE (Ren et al., 2024). For Q-based IL, we include ValueDICE (Kostrikov et al., 2020), IQ-Learn (Garg et al., 2021), LS-IQ (Al-Hafez et al., 2023), and ReCOIL (Sikchi et al., 2024).

### 6.2. Experimental Results

**Results on the Hard Instance.** We evaluate different methods on the hard instance from Corollary 1 and report their imitation gaps across varying horizons. As shown in Figure 4, BC, IQ-Learn (with TV and $\chi^2$ divergences), and Val-

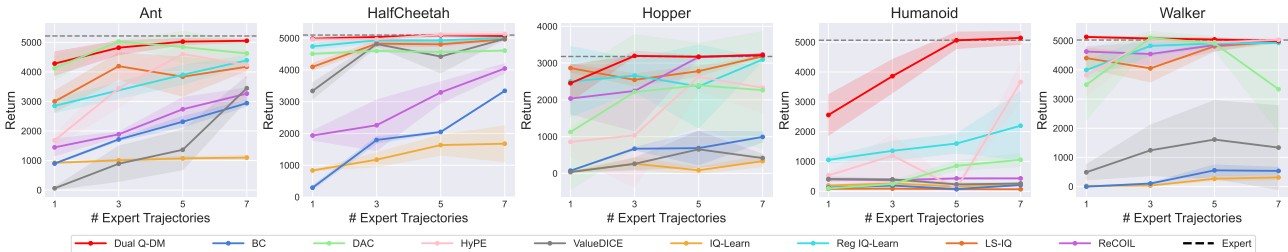

*Figure 2.* Learning curves regarding online environment interactions on 5 MuJoCo tasks.

*Figure 3.* Final performance regarding different number of expert trajectories on 5 MuJoCo tasks.

ueDICE all exhibit imitation gaps growing linearly with $H^2$, confirming compounding errors predicted by Corollary 1. In contrast, Dual Q-DM and AIL maintain consistently small imitation gaps as the horizon increases, confirming Theorem 2, which establishes that Dual Q-DM inherits AIL's ability to eliminate compounding errors.

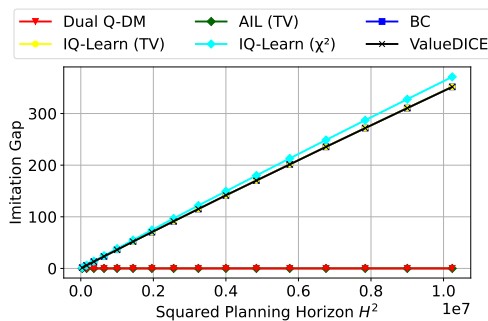

*Figure 4.* Imitation gaps across different horizons.

**Results on MuJoCo Tasks.** For MuJoCo tasks, we evaluate two variants of IQ-Learn: the standard IQ-Learn as originally proposed, and Reg IQ-Learn, which corresponds to the official implementation incorporating additional regularization[2]. LS-IQ and ReCOIL follow Reg IQ-Learn in also incorporating regularization. From Figure 2, we draw four conclusions. First, IQ-Learn and ValueDICE exhibit poor performance comparable to BC, confirming our theoretical prediction in Theorem 1. Second, Reg IQ-Learn, LS-IQ, and ReCOIL achieve substantial improvements over IQ-Learn through regularization. Notably, this regularization shares

similarities with our Bellman constraints, and thus our analysis provides a theoretical explanation for its effectiveness from a fundamental value flow perspective. Third, Dual Q-DM demonstrates superior performance compared to other Q-based methods, validating Theorem 2 and Proposition 1: due to the value flow driven by Bellman constraints, Dual Q-DM effectively generalize beyond demonstrations and mitigate compounding errors. Finally, relative to AIL methods, Dual Q-DM exhibits faster convergence and improved training stability due to its non-adversarial training.

Figure 3 further presents the final performance of all methods across varying numbers of expert trajectories. Dual Q-DM consistently achieves higher final performance than competing Q-based methods across all trajectory counts, demonstrating superior demonstration efficiency.

## 7. Conclusion

This paper provides a comprehensive theoretical analysis of Q-based IL. We prove that IQ-Learn, a representative Q-based method, reduces to BC and inherits its compounding errors. To address this fundamental limitation, we propose Dual Q-DM, which introduces Bellman constraints as its core algorithmic design. We establish theoretical equivalence between Dual Q-DM and AIL, making it the first non-adversarial IL method provably free from compounding errors. Crucially, Bellman constraints drive value flow: Q-values propagates from demonstrated to unvisited states through environment dynamics. Our analysis reveals it is this value flow, not merely online interactions, that enables generalization beyond demonstrations, offering principled guidance for designing effective Q-based IL algorithms.

---

[2]The official IQ-Learn implementation on MuJoCo tasks includes a regularization term in the objective function that is absent from the paper. See Appendix C.6 for details.

## Acknowledgement

Tian Xu thanks Yushun Zhang for his valuable discussions on constrained optimization. This work was supported by NSFC (No. 62495090, No. 62495093), the Jiangsu Science Foundation (No. BK20243039), the Fundamental and Interdisciplinary Disciplines Breakthrough Plan of the Ministry of Education of China (No. JYB2025XDXM118), and the "111 Center" (No. B26023).

## Impact Statement

This work improves the theoretical understanding of imitation learning and provides a stable non-adversarial method with provable guarantees. By enabling reliable generalization beyond demonstrations, it may benefit applications such as robotics and autonomous systems. While stronger imitation capabilities may raise concerns in certain misuse scenarios, we expect this work to primarily support the development of robust and responsible learning systems.

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

# A. Additional Related Work

**Behavior Cloning.** BC directly performs maximum likelihood estimation (MLE) on the expert dataset, enabling accurate reproduction of expert actions within the support of demonstrations. However, its offline learning nature prevents BC from recovering expert actions on states unvisited in demonstrations, leading to the compounding error problem (Ross & Bagnell, 2010). Ross & Bagnell (2010) first established that BC achieves an imitation gap bound with quadratic dependence on the horizon H at the population level. Rajaraman et al. (2020) and Xu et al. (2020) subsequently extended this analysis to the finite sample regime. Critically, Rajaraman et al. (2020) proved a fundamental $\Omega(H^2)$ lower bound for any offline IL algorithm, including BC, demonstrating that online interaction is essential for breaking the quadratic barrier. Li et al. (2023) further analyzed an importance-sampling weighted BC (ISW-BC) approach for IL from imperfect demonstrations and proved that ISW-BC can effectively leverage imperfect demonstrations to reduce imitation gap. Recently, Foster et al. (2024) developed a sharp analysis of BC with general function approximation under stationarity assumptions, while Simchowitz et al. (2025) investigated BC in continuous control settings and proved that BC can suffer an exponentially growing imitation gap with respect to the horizon in certain constructed hard instances.

**Adversarial Imitation Learning.** Unlike BC, AIL operates on the principle of state-action distribution matching. Building on this foundation, numerous practical AIL algorithms have been developed and applied across diverse domains (Ho & Ermon, 2016; Fu et al., 2018; Ke et al., 2019; Kostrikov et al., 2019; Dadashi et al., 2021; Viano et al., 2022; Swamy et al., 2023; Watson et al., 2023; Ren et al., 2024; Viel et al., 2025). Empirical studies (Ho & Ermon, 2016; Kostrikov et al., 2019; Ghasemipour et al., 2019) have revealed a key advantage: AIL can mitigate the compounding error problem that plagues BC. This observation has motivated extensive theoretical investigation (Zhang et al., 2020; Wang et al., 2020; Rajaraman et al., 2020; 2021; Xu et al., 2020; Liu et al., 2021; Viano et al., 2024; Xu et al., 2024; Zhang et al., 2025; Xu et al., 2026a;b) into AIL's underlying mechanisms. Notably, Agarwal et al. (2019); Xu et al. (2020) proved that AIL achieves an imitation gap bound of $\mathcal{O}(H)$ with only linear dependence on the horizon $H$, thereby provably eliminating BC's compounding error problem. Rajaraman et al. (2020) advanced this line of work by introducing improved state-action distribution estimation for AIL, yielding better sample complexity bounds. Xu et al. (2026a) further established a sharp horizon-free bound for AIL. This bound is meaningful in the low data regime and thus can explain the good empirical performance of AIL with limited demonstrations (Ghasemipour et al., 2019). More recently, Liu et al. (2021); Viano et al. (2024); Xu et al. (2026b) extended AIL theory to the function approximation setting. In this work, we focus on the theoretical analysis of Q-based IL and reveal that Bellman constraints play a role analogous to the direct RL optimization in AIL, serving as the crucial mechanism for eliminating compounding errors.

**Q-learning-based Imitation Learning.** To circumvent adversarial optimization in AIL, a family of non-adversarial Q-based IL methods (Kostrikov et al., 2020; Garg et al., 2021; Al-Hafez et al., 2023; Karimi & Ebadzadeh, 2025; Li et al., 2025) has emerged that directly optimize a single maximization objective over Q-functions. ValueDICE, the pioneering Q-based IL method, derived a Q-function objective from KL-divergence-based distribution matching through a change-of-variables transformation of the reward function. Building on this foundation, IQ-Learn (Garg et al., 2021) exploited the closed-form solution to entropy-regularized RL to convert AIL's minimax objective into a single maximization problem. Importantly, while IQ-Learn's official MuJoCo implementation includes a regularization term that substantially improves performance, this term was not presented in the original paper. LS-IQ (Al-Hafez et al., 2023) subsequently interpreted this regularization as $\chi^2$-divergence minimization over mixture distributions and introduced additional techniques to enhance training stability. RIZE (Karimi & Ebadzadeh, 2025) further extended LS-IQ through adaptive regularization. In contrast to these empirical advances, this paper provides a systematic theoretical analysis of Q-based IL. We identify conceptual connections between existing regularization techniques and our proposed Bellman constraints, offering a fundamental RL-based explanation for their effectiveness: propagating Q-value information through environment dynamics from visited to unvisited states is essential for Q-based IL to generalize beyond demonstrations and mitigate compounding errors.

Despite substantial empirical progress, theoretical understanding of Q-based IL remains limited. To our knowledge, only Moulin et al. (2025) has provided theoretical analysis, introducing an offline Q-based method with minimax formulation and proving finite sample complexity under Q-function realizability. Our work differs fundamentally by theoretically investigating which algorithmic mechanisms are necessary for eliminating compounding errors. We prove that IQ-Learn essentially reduces to BC and propose a novel online Dual Q-DM method that provably avoids compounding errors. Our theoretical analysis reveals that Bellman constraints enable generalization beyond demonstrations by propagating Q-value information through environment dynamics from visited to unvisited states—a mechanism essential for mitigating compounding errors in Q-based IL.

**Primal-Dual Perspective in Imitation Learning.** Several works have explored primal-dual formulations for distribution matching in IL (Viano et al., 2022; Kim et al., 2022; Ma et al., 2022; Sikchi et al., 2024). Viano et al. (2022) derives a minimax objective over Q-functions and reward functions from the primal distribution matching problem and establishes imitation gap bounds for their algorithm. Kim et al. (2022); Ma et al. (2022) focus on IL with supplementary datasets. Similarly, starting from the primal distribution matching formulation, they employ Lagrangian duality theory to derive a minimax objective over distribution ratios and value functions. In contrast, we focus on the pure IL setting without supplementary data and introduce a primal-dual framework that yields a single maximization problem over Q-functions rather than a minimax formulation.

The most closely related is Sikchi et al. (2024), which derives a Q-based algorithm from primal distribution matching that can leverage supplementary data. Like IQ-Learn, their approach does not incorporate Bellman constraints into the objective. In contrast, our work develops a primal-dual framework for distribution matching where Bellman constraints play a central role. We rigorously establish the equivalence between this framework and AIL, thereby unifying Q-based and AIL methods under a single theoretical lens.

# B. Omitted Results in Section 3

## B.1. Proof of Theorem 1

We prove Theorem 1 by considering the following two cases.

**Case I:** $2 \leq h \leq H$. We first consider the time step $2 \leq h \leq H$. Since the state space is layered, we can analyze the IQ-Learn objective at each time step. For any $h \in [2, H]$, we have that

$$\mathbb{E}_{(s_h, a_h) \sim \widehat{d_h^{\pi^{\mathrm{E}}}}} \left[ Q(s_h, a_h) - \log(\sum_{a \in \mathcal{A}} \exp(Q(s_h, a))) \right] = \mathbb{E}_{(s_h, a_h) \sim \widehat{d_h^{\pi^{\mathrm{E}}}}} \left[ \log \left( \frac{\exp(Q(s_h, a_h))}{\sum_{a \in \mathcal{A}} \exp(Q(s_h, a))} \right) \right]. \quad (11)$$

For any Q-function $Q$, the corresponding softmax policy $\pi_Q$ is defined as $\pi_Q(a|s) = \exp(Q(s, a)) / \sum_{a' \in \mathcal{A}} \exp(Q(s, a'))$. Furthermore, for the Q-function class $\mathcal{Q} := \{Q : \mathcal{S} \times \mathcal{A} \to [0, C]\}$, we define the derived policy class $\Pi_{\mathcal{Q}} = \{\pi_Q : Q \in \mathcal{Q}\}$. Then the IQ-Learn objective over the Q-function class in Eq. (12) is equivalent to the following maximum likelihood estimation objective over the policy class.

$$\max_{Q \in \mathcal{Q}} \mathbb{E}_{(s_h, a_h) \sim \widehat{d_h^{\pi^{\mathrm{E}}}}} \left[ \log \left( \frac{\exp(Q(s_h, a_h))}{\sum_{a \in \mathcal{A}} \exp(Q(s_h, a))} \right) \right] \quad (12)$$

$$\Leftrightarrow \max_{\pi \in \Pi_{\mathcal{Q}}} \mathbb{E}_{(s_h, a_h) \sim \widehat{d_h^{\pi^{\mathrm{E}}}}} \left[ \log \left( \pi(s_h|a_h) \right) \right]. \quad (13)$$

In particular, suppose that $\widehat{Q}$ is the optimal solution to Eq. (12), then the corresponding softmax policy $\pi_{\widehat{Q}}$ is the optimal solution to Eq. (13). Notice that Eq. (13) is exactly the BC objective in time step $h$. Furthermore, due to the realizability of $\Pi_{\mathcal{Q}}$, i.e., $\pi^{\mathrm{BC}} \in \Pi_{\mathcal{Q}}$, $\pi^{\mathrm{BC}}$ is also the optimal solution to Eq. (13). Then we can obtain that $\forall 2 \leq h \leq H, \forall s_h \in \mathcal{S}_h, \pi_{\widehat{Q}}(\cdot|s_h) = \pi^{\mathrm{BC}}(\cdot|s_h)$, which proves the first claim in Theorem 1.

**Case II:** $h = 1$. Then we consider the initial time step $h = 1$. For BC, we can write down the closed-form solution based on the first order optimality condition.

$$\pi^{\mathrm{BC}}(a_h|s_h) = \begin{cases} \frac{n(s_h, a_h)}{n(s_h)} & \text{if } n(s_h) > 0 \\ \frac{1}{|\mathcal{A}|} & \text{otherwise} \end{cases} \quad (14)$$

Here $n(s_h, a_h)$ ($n(s_h)$) refers to the number of times that the state-action pair $(s_h, a_h)$ (state $s_h$) has appeared in $\mathcal{D}^{\mathrm{E}}$ in time step $h$. From Eq. (14), we can conclude that for an unvisited state $s_1 \notin \mathcal{D}^{\mathrm{E}}$, $\pi^{\mathrm{BC}}(a|s_1) = 1/|\mathcal{A}|, \forall a \in \mathcal{A}$.

For IQ-Learn, we have that

$$\widehat{Q} = \underset{Q \in \mathcal{Q}}{\operatorname{argmax}} \mathbb{E}_{(s_1, a_1) \sim \widehat{d_1^{\pi^{\mathrm{E}}}}} [Q(s_1, a_1)] - \mathbb{E}_{s_1 \sim \rho} \left[ \log(\sum_{a \in \mathcal{A}} \exp(Q(s_1, a))) \right]$$

$$= \underset{Q \in \mathcal{Q}}{\operatorname{argmax}} \sum_{(s_1, a) \in \mathcal{S}_1 \times \mathcal{A}} \widehat{d_1^{\pi^{\mathrm{E}}}}(s_1, a) Q(s_1, a) - \sum_{s_1 \in \mathcal{S}_1} \rho(s_1) \log(\sum_{a \in \mathcal{A}} \exp(Q(s_1, a)))$$

We analyze $\widehat{Q}$ on each unvisited state $s_1 \notin \mathcal{D}^{\mathrm{E}}$. In particular, we consider the following two cases.

- Case II (a): For state $s_1$ such that $\widehat{d_1^{\pi^{\mathrm{E}}}}(s_1) = 0, \rho(s_1) > 0$, the optimization problem can be formulated as

$$\operatorname*{argmin}_{Q \in \mathcal{Q}} \rho(s_1) \log(\sum_{a \in \mathcal{A}} \exp(Q(s_1, a))).$$

  This is a monotonically increasing function w.r.t $Q(s_1, a)$ for each action $a \in \mathcal{A}$. Thus, it is direct to write down the optimal solution $\forall a \in \mathcal{A}, \widehat{Q}(s_1, a) = 0$. The corresponding softmax policy can be formulated as $\forall a \in \mathcal{A}, \pi_{\widehat{Q}}(a|s_1) = 1/|\mathcal{A}|$.

- Case II (b): For states $s_1$ where $\widehat{d_1^{\pi^{\mathrm{E}}}}(s_1) = 0$ and $\rho(s_1) = 0$, the objective function provides no learning signal, so the Q-function retains its initialization. Assuming a constant initialization for simplicity, the corresponding softmax policy is uniform: $\pi_{\widehat{Q}}(a \mid s_1) = 1/|\mathcal{A}|$ for all $a \in \mathcal{A}$.

In both Case II (a) and Case II (b), we can prove that for any unvisited state $s_1 \notin \mathcal{D}^{\mathrm{E}}, \forall a \in \mathcal{A}, \pi_{\widehat{Q}}(a|s_1) = 1/\mathcal{A}$, which proves the second claim in Theorem 1.

## B.2. Connection Between ValueDICE and BC

As discussed in Remark 3, the re-parameterization perspective could also help analyze the connection between another famous Q-based method ValueDICE (Kostrikov et al., 2020) and BC. In particular, ValueDICE first learns a Q-function via

$$\max_{Q \in \mathcal{Q}} - \log \left( \mathbb{E}_{\tau \sim \mathcal{D}^{\mathrm{E}}} \left[ \sum_{h=1}^{H} \exp\left( - Q(s_h, a_h) + \mathrm{MAX}(Q)(s_{h+1})\right) \right] \right) - \mathbb{E}_{s_1 \sim \rho} \left[ \mathrm{MAX}(Q)(s_1) \right]. \tag{15}$$

and derives argmax policy $\pi_Q(s) = \operatorname{argmax}_{a \in \mathcal{A}} Q(s, a)$ where $\mathrm{MAX}(Q)(s) = \max_{a \in \mathcal{A}} Q(s, a)$. Comparing Eq. (15) with Eq. (4), ValueDICE shares the same structural form as IQ-Learn, with $\mathrm{MAX}(Q)(s)$ playing the role of $\mathrm{LSE}(Q)(s)$. Applying the re-parameterization argument from Remark 3, one can identify a term of the form $\max_Q(Q(s_h, a_h) - \mathrm{MAX}(Q)(s_h)) = \max_{\pi} C \cdot \mathbb{I}\{\pi_Q(s_h) = a_h\}$ embedded in Eq. (15), which corresponds to BC with a 0-1 loss.

## B.3. Proof of Corollary 1

We consider the Reset Cliff MDP introduced in (Rajaraman et al., 2020). The environment is defined as follows. The state space at time step $h$ is $\mathcal{S}_h$, where $|\mathcal{S}_1| = |\mathcal{S}_2| = \cdots = |\mathcal{S}_H| = S$, and each $\mathcal{S}_h$ contains a designated bad state $b_h$. The initial state distribution is given by

$$\rho = \left( \tfrac{1}{N+1}, \ldots, \tfrac{1}{N+1}, 1 - \tfrac{S-2}{N+1}, 0 \right),$$

where $N = |\mathcal{D}^E|$ is the number of trajectories in the expert dataset.

We consider a fixed deterministic expert policy $\pi^{\mathrm{E}}$. At any non-bad state $s_h \in \mathcal{S}_h \setminus \{b_h\}$, taking the expert action $\pi^{\mathrm{E}}(s_h)$ resets the agent to a state drawn from $\rho$ and yields a reward of 1; any other action transitions deterministically to the bad state $b_{h+1}$ and yields a reward of 0. The bad states are absorbing: every action from $b_h$ returns to $b_{h+1}$ with reward 0. Formally, the transition and reward functions are

$$P(\cdot \mid s_h, a_h) = \begin{cases} \rho, & s_h \in \mathcal{S}_h \setminus \{b_h\}, \ a_h = \pi^{\mathrm{E}}(s_h), \\ \delta_{b_h}, & \text{otherwise,} \end{cases}$$

$$r(s_h, a_h) = \begin{cases} 1, & s_h \in \mathcal{S}_h \setminus \{b_h\}, \ a_h = \pi^{\mathrm{E}}(s_h), \\ 0, & \text{otherwise.} \end{cases}$$

Here $\delta_{b_h}$ is a dirac delta distribution on the bad state $b_h$.

We begin with the following imitation gap decomposition.

$$V^{\pi^{\mathrm{E}}} - \mathbb{E}\left[ V^{\widehat{\pi}} \right] = V^{\pi^{\mathrm{E}}} - \mathbb{E}\left[ V^{\pi^{\mathrm{BC}}} \right] + \mathbb{E}\left[ V^{\pi^{\mathrm{BC}}} - V^{\widehat{\pi}} \right].$$

To prove the lower bound, we first analyze the term $V^{\pi^{\mathrm{BC}}} - V^{\widehat{\pi}}$. In particular, according to the policy difference lemma, we can obtain that

$$V^{\pi^{\mathrm{BC}}} - V^{\widehat{\pi}} = \mathbb{E}\left[\sum_{h=1}^{H} \mathbb{E}_{a \sim \pi^{\mathrm{BC}}(\cdot|s_h)}\left[Q^{\widehat{\pi}}(s_h, a)\right] - \mathbb{E}_{a \sim \widehat{\pi}(\cdot|s_h)}\left[Q^{\widehat{\pi}}(s_h, a)\right] \bigg| \pi^{\mathrm{BC}}\right].$$

Theorem 1 discloses that $\forall 2 \leq h \leq H, \ \forall s \in \mathcal{S}_h, \widehat{\pi}(a|s) = \pi^{\mathrm{BC}}(a|s)$. Then we can obtain that

$$V^{\pi^{\mathrm{BC}}} - V^{\widehat{\pi}} = \mathbb{E}\left[\mathbb{E}_{a \sim \pi^{\mathrm{BC}}(\cdot|s_1)}\left[Q^{\widehat{\pi}}(s_1, a)\right] - \mathbb{E}_{a \sim \widehat{\pi}(\cdot|s_1)}\left[Q^{\widehat{\pi}}(s_1, a)\right] \bigg| \pi^{\mathrm{BC}}\right].$$

In the Reset Cliff MDP, if the agent takes a non-expert action, then it will transition into the absorbing states and cannot receive positive rewards anymore. This implies that $Q^{\widehat{\pi}}(s_1, a) = 0, \forall a \neq \pi^{\mathrm{E}}(s_1)$. Then we can obtain that

$$V^{\pi^{\mathrm{BC}}} - V^{\widehat{\pi}} = \mathbb{E}\left[Q^{\widehat{\pi}}(s_1, \pi^{\mathrm{E}}(s_1))\left(\pi^{\mathrm{BC}}(\pi^{\mathrm{E}}(s_1)|s_1) - \widehat{\pi}(\pi^{\mathrm{E}}(s_1)|s_1)\right) \bigg| \pi^{\mathrm{BC}}\right].$$

For any step $h$, we define $\mathcal{S}_h(\mathcal{D}^{\mathrm{E}}) := \{s_h \in \mathcal{S}_h : s_h \in \mathcal{D}^{\mathrm{E}}\}$ as the set of states visited in $\mathcal{D}^{\mathrm{E}}$ in time step $h$. According to Eq. (14), the BC policy can exactly recover the expert action in any visited state $s \in \mathcal{S}_1(\mathcal{D}^{\mathrm{E}})$ and thus $\pi^{\mathrm{BC}}(\pi^{\mathrm{E}}(s)|s) - \widehat{\pi}(\pi^{\mathrm{E}}(s)|s) = 1 - \widehat{\pi}(\pi^{\mathrm{E}}(s)|s) \geq 0$. For any unvisited state $s \notin \mathcal{S}_1(\mathcal{D}^{\mathrm{E}}) \cup \{b\}$, Theorem 1 implies that $\pi^{\mathrm{BC}}(\pi^{\mathrm{E}}(s)|s) - \widehat{\pi}(\pi^{\mathrm{E}}(s)|s) = 1/|\mathcal{A}| - 1/|\mathcal{A}| = 0$. Then we can obtain $V^{\pi^{\mathrm{BC}}} - V^{\widehat{\pi}} \geq 0$ almost surely. Then we have that

$$V^{\pi^{\mathrm{E}}} - \mathbb{E}\left[V^{\widehat{\pi}}\right] = V^{\pi^{\mathrm{E}}} - \mathbb{E}\left[V^{\pi^{\mathrm{BC}}}\right] + \mathbb{E}\left[V^{\pi^{\mathrm{BC}}} - V^{\widehat{\pi}}\right] \geq V^{\pi^{\mathrm{E}}} - \mathbb{E}\left[V^{\pi^{\mathrm{BC}}}\right].$$

Now, we can turn to analyze the imitation gap of $V^{\pi^{\mathrm{E}}} - \mathbb{E}[V^{\pi^{\mathrm{BC}}}]$. While Rajaraman et al. (2020) have established the imitation gap lower bound for any offline IL method, including BC, we identify a flaw in their proof. We therefore develop a new analysis to establish the imitation gap lower bound specifically for BC.

In the Reset Cliff MDP, the agent suffers a reward loss when it takes a non-expert action for the first time. Therefore, we define $t$ as the first time that the agent takes a non-expert action in a trajectory $\tau = \{s_1, a_1, s_2, \ldots, s_H, a_H\}$.

$$t = \begin{cases} \inf\{h : a_h \neq \pi^{\mathrm{E}}(s_h)\}, & \text{if } \exists h \in [H], a_h \neq \pi^{\mathrm{E}}(s_h), \\ H + 1, & \text{otherwise.} \end{cases}$$

Then we have that

$$V^{\pi^{\mathrm{E}}} - V^{\pi^{\mathrm{BC}}} = H - \mathbb{E}\left[\sum_{h=1}^{H} r(s_h, a_h) \bigg| \pi^{\mathrm{BC}}\right] \overset{(a)}{=} \mathbb{E}\left[H - t + 1 \big| \pi^{\mathrm{BC}}\right] = \sum_{h=1}^{H+1} \mathbb{P}^{\pi^{\mathrm{BC}}}(t = h)(H - h + 1)$$

$$= \sum_{h=1}^{H} \mathbb{P}^{\pi^{\mathrm{BC}}}(t = h)(H - h + 1).$$

Equation $(a)$ holds because in the Reset Cliff MDP, the agent receives zero reward once it takes a non-expert action. To analyze the probability $\mathbb{P}^{\pi^{\mathrm{BC}}}(t = h)$, we define another random variable $t^u$ as the first time that the agent visits a state uncovered in demonstrations $\mathcal{D}^{\mathrm{E}}$.

$$t^u = \begin{cases} \inf\{h : s_h \notin \mathcal{S}_h(\mathcal{D}^{\mathrm{E}}) \cup \{b_h\}\}, & \text{if } \exists h \in [H], s_h \notin \mathcal{S}_h(\mathcal{D}^{\mathrm{E}}) \cup \{b_h\}, \\ H + 1, & \text{otherwise.} \end{cases}$$

Here $\mathcal{S}_h(\mathcal{D}^{\mathrm{E}})$ denotes the set of states that are visited in $\mathcal{D}^{\mathrm{E}}$ in time step $h$. Then we can obtain that

$$\mathbb{P}^{\pi^{\mathrm{BC}}}(t = h) \geq \mathbb{P}^{\pi^{\mathrm{BC}}}(t = h, t^u = h) = \mathbb{P}^{\pi^{\mathrm{BC}}}(t^u = h)\mathbb{P}^{\pi^{\mathrm{BC}}}(t = h|t^u = h).$$

For the conditional probability, we have that

$$\mathbb{P}^{\pi^{\mathrm{BC}}}(t = h|t^u = h) = \sum_{s \in \mathcal{S}_h} \mathbb{P}^{\pi^{\mathrm{BC}}}(t = h, s_h = s|t^u = h)$$

$$= \sum_{s \in \mathcal{S}_h} \mathbb{P}^{\pi^{\mathrm{BC}}} (t = h|s_h = s, t^u = h) \, \mathbb{P}^{\pi^{\mathrm{BC}}} (s_h = s|t^u = h)$$

$$= \sum_{s \notin \mathcal{S}_h(\mathcal{D}^{\mathrm{E}}) \cup \{b_h\}} \mathbb{P}^{\pi^{\mathrm{BC}}} (t = h|s_h = s, t^u = h) \, \mathbb{P}^{\pi^{\mathrm{BC}}} (s_h = s|t^u = h).$$

The last equation follows that for any $s \in \mathcal{S}_h(\mathcal{D}^{\mathrm{E}}) \cup \{b_h\}$, $\mathbb{P}^{\pi^{\mathrm{BC}}}(s_h = s|t^u = h) = 0$ since $h$ is the first time step that $\pi^{\mathrm{BC}}$ encounters a state uncovered in $\mathcal{D}^{\mathrm{E}}$. For any $s \notin \mathcal{S}_h(\mathcal{D}^{\mathrm{E}}) \cup \{b_h\}$, we have

$$\mathbb{P}^{\pi^{\mathrm{BC}}} (t = h|s_h = s, t^u = h)$$

$$= \sum_{a \neq \pi^{\mathrm{E}}(s)} \mathbb{P}^{\pi^{\mathrm{BC}}} \left( a_h = a, a_{1:h-1} = \pi^{\mathrm{E}}(s_{1:h-1})|s_h = s, t^u = h \right)$$

$$= \sum_{a \neq \pi^{\mathrm{E}}(s)} \mathbb{P}^{\pi^{\mathrm{BC}}} \left( a_h = a|a_{1:h-1} = \pi^{\mathrm{E}}(s_{1:h-1}), s_h = s, t^u = h \right) \mathbb{P}^{\pi^{\mathrm{BC}}} \left( a_{1:h-1} = \pi^{\mathrm{E}}(s_{1:h-1})|s_h = s, t^u = h \right)$$

$$\overset{(a)}{=} \sum_{a \neq \pi^{\mathrm{E}}(s)} \mathbb{P}^{\pi^{\mathrm{BC}}} \left( a_h = a|a_{1:h-1} = \pi^{\mathrm{E}}(s_{1:h-1}), s_h = s, t^u = h \right)$$

$$= \sum_{a \neq \pi^{\mathrm{E}}(s)} \mathbb{P}^{\pi^{\mathrm{BC}}} \left( a_h = a|a_{1:h-1} = \pi^{\mathrm{E}}(s_{1:h-1}), s_h = s, \forall 1 \leq t \leq h-1, s_t \in \mathcal{S}_t(\mathcal{D}^{\mathrm{E}}) \cup \{b_t\}, s_h \notin \mathcal{S}_h(\mathcal{D}^{\mathrm{E}}) \cup \{b_h\} \right)$$

$$\overset{(b)}{=} \sum_{a \neq \pi^{\mathrm{E}}(s)} \mathbb{P}^{\pi^{\mathrm{BC}}} \left( a_h = a|s_h = s, s_h \notin \mathcal{S}_h(\mathcal{D}^{\mathrm{E}}) \cup \{b_h\} \right)$$

$$\overset{(c)}{=} \sum_{a \neq \pi^{\mathrm{E}}(s)} \mathbb{P}^{\pi^{\mathrm{BC}}} \left( a_h = a|s_h = s \right)$$

$$= \sum_{a \neq \pi^{\mathrm{E}}(s)} \pi^{\mathrm{BC}}(a|s)$$

$$\overset{(d)}{=} 1 - \frac{1}{|\mathcal{A}|}.$$

In equation $(a)$, we leverage the property that $\mathbb{P}^{\pi^{\mathrm{BC}}} \left( a_{1:h-1} = \pi^{\mathrm{E}}(s_{1:h-1})|s_h = s, t^u = h \right) = 1$ because the event $t^u = h$ means the agent visits a state $s_h \neq b$ in time step $h$, implying that it must always takes the expert action before $h$. Equation $(b)$ follows that $a_h$ is independent on $s_{1:h-1}$ and $a_{1:h-1}$ conditioned on $s_h$. Equation $(c)$ holds because the event $s_h = s$ for state $s \notin \mathcal{S}_h(\mathcal{D}^{\mathrm{E}}) \cup \{b_h\}$ implies the event $s_h \notin \mathcal{S}_h(\mathcal{D}^{\mathrm{E}}) \cup \{b_h\}$. In equation (d), we leverage the property that in a state $s$ unvisited in $\mathcal{D}^{\mathrm{E}}$, $\forall a \in \mathcal{A}, \pi^{\mathrm{BC}}(a|s) = 1/|\mathcal{A}|$ due to Eq. (14). Then we can calculate the conditional probability.

$$\mathbb{P}^{\pi^{\mathrm{BC}}} (t = h|t^u = h) = \sum_{s \notin \mathcal{S}_h(\mathcal{D}^{\mathrm{E}}) \cup \{b_h\}} \mathbb{P}^{\pi^{\mathrm{BC}}} (t = h|s_h = s, t^u = h) \, \mathbb{P}^{\pi^{\mathrm{BC}}} (s_h = s|t^u = h)$$

$$= \left( 1 - \frac{1}{|\mathcal{A}|} \right) \sum_{s \notin \mathcal{S}_h(\mathcal{D}^{\mathrm{E}}) \cup \{b_h\}} \mathbb{P}^{\pi^{\mathrm{BC}}} (s_h = s|t^u = h)$$

$$= 1 - \frac{1}{|\mathcal{A}|}.$$

Then we have that

$$\mathbb{P}^{\pi^{\mathrm{BC}}} (t = h) \geq \mathbb{P}^{\pi^{\mathrm{BC}}} (t^u = h) \, \mathbb{P}^{\pi^{\mathrm{BC}}} (t = h|t^u = h) \geq \left( 1 - \frac{1}{|\mathcal{A}|} \right) \mathbb{P}^{\pi^{\mathrm{BC}}} (t^u = h).$$

Then we proceed to analyze the imitation gap.

$$V^{\pi^{\mathrm{E}}} - V^{\pi^{\mathrm{BC}}} = \sum_{h=1}^{H} \mathbb{P}^{\pi^{\mathrm{BC}}} (t = h) \, (H - h + 1)$$

$$\geq \left( 1 - \frac{1}{|\mathcal{A}|} \right) \sum_{h=1}^{H} \mathbb{P}^{\pi^{\mathrm{BC}}} (t^u = h) \, (H - h + 1).$$

Taking an expectation over $\mathcal{D}^{\mathrm{E}}$ on both sides yields that

$$\mathbb{E}_{\mathcal{D}^{\mathrm{E}}}\left[V^{\pi^{\mathrm{E}}} - V^{\pi^{\mathrm{BC}}}\right] = \left(1 - \frac{1}{|\mathcal{A}|}\right) \sum_{h=1}^{H} \mathbb{E}_{\mathcal{D}^{\mathrm{E}}}\left[\mathbb{P}^{\pi^{\mathrm{BC}}}\left(t^u = h\right)\right] (H - h + 1). \tag{16}$$

Then we can calculate the probability $\mathbb{P}^{\pi^{\mathrm{BC}}}\left(t^u = h\right)$ as follows.

$$
\begin{aligned}
\mathbb{P}^{\pi^{\mathrm{BC}}}\left(t^u = h\right) &= \mathbb{P}^{\pi^{\mathrm{BC}}}\left(s_h \notin \mathcal{S}_h(\mathcal{D}^{\mathrm{E}}) \cup \{b_h\}, \forall h' \in [h-1], s_{h'} \in \mathcal{S}_{h'}(\mathcal{D}^{\mathrm{E}}) \cup \{b_{h'}\}\right) \\
&\overset{(a)}{=} \mathbb{P}^{\pi^{\mathrm{BC}}}\left(s_h \notin \mathcal{S}_h(\mathcal{D}^{\mathrm{E}}) \cup \{b_h\}, \forall h' \in [h-1], s_{h'} \in \mathcal{S}_{h'}(\mathcal{D}^{\mathrm{E}})\right) \\
&\overset{(b)}{=} \mathbb{P}^{\pi^{\mathrm{E}}}\left(s_h \notin \mathcal{S}_h(\mathcal{D}^{\mathrm{E}}) \cup \{b_h\}, \forall h' \in [h-1], s_{h'} \in \mathcal{S}_{h'}(\mathcal{D}^{\mathrm{E}})\right) \\
&\overset{(c)}{=} \left(\prod_{\ell=1}^{h-1} \mathbb{P}^{\pi^{\mathrm{E}}}(s_\ell \in \mathcal{S}_h(\mathcal{D}^{\mathrm{E}}))\right) \mathbb{P}^{\pi^{\mathrm{E}}}\left(s_h \notin \mathcal{S}_h(\mathcal{D}^{\mathrm{E}}) \cup \{b_h\}\right) \\
&= \left(\prod_{\ell=1}^{h-1}\left(\sum_{s \in \mathcal{S}_\ell(\mathcal{D}^{\mathrm{E}})} \rho(s)\right)\right)\left(\sum_{s \notin \mathcal{S}_h(\mathcal{D}^{\mathrm{E}}) \cup \{b_h\}} \rho(s)\right).
\end{aligned}
$$

Equation (a) holds because if the agent visits the state $b_{h'}$ in some preceding time step $h' < h$, it must visit the state $b_h$ in time step $h$. Equation (b) holds because that $\forall h' \in [h-1], \forall s_{h'} \in \mathcal{S}_{h'}(\mathcal{D}^{\mathrm{E}}), \pi^{\mathrm{BC}}(\pi^{\mathrm{E}}(s_{h'})|s_{h'}) = 1$. Equation (c) leverages the property that under the expert policy, the state in each timestep is i.i.d. drawn from the initial state distribution $\rho$. Taking an expectation over $\mathcal{D}^{\mathrm{E}}$ on both sides yields that

$$
\begin{aligned}
\mathbb{E}_{\mathcal{D}^{\mathrm{E}}}\left[\mathbb{P}^{\pi^{\mathrm{BC}}}\left(t^u = h\right)\right] &= \mathbb{E}_{\mathcal{D}^{\mathrm{E}}}\left[\left(\prod_{\ell=1}^{h-1}\left(\sum_{s \in \mathcal{S}_\ell(\mathcal{D}^{\mathrm{E}})} \rho(s)\right)\right)\left(\sum_{s \notin \mathcal{S}_h(\mathcal{D}^{\mathrm{E}}) \cup \{b_h\}} \rho(s)\right)\right] \\
&\overset{(a)}{=} \left(\prod_{\ell=1}^{h-1} \mathbb{E}_{\mathcal{D}^{\mathrm{E}}}\left[\sum_{s \in \mathcal{S}_\ell(\mathcal{D}^{\mathrm{E}})} \rho(s)\right]\right) \mathbb{E}_{\mathcal{D}^{\mathrm{E}}}\left[\sum_{s \notin \mathcal{S}_h(\mathcal{D}^{\mathrm{E}}) \cup \{b_h\}} \rho(s)\right].
\end{aligned}
$$

Equation (a) holds because the state at each time step of $\mathcal{D}^{\mathrm{E}}$ is i.i.d drawn from the initial state distribution $\rho$. For each term in the RHS, we have that

$$\mathbb{E}_{\mathcal{D}^{\mathrm{E}}}\left[\sum_{s \in \mathcal{S}_\ell(\mathcal{D}^{\mathrm{E}})} \rho(s)\right] = 1 - \sum_{s \in \mathcal{S}_\ell} \rho(s)\mathbb{P}\left(s \notin \mathcal{S}_\ell(\mathcal{D}^{\mathrm{E}})\right) = 1 - \sum_{s \in \mathcal{S}_\ell} \rho(s)(1 - \rho(s))^N,$$

$$\mathbb{E}_{\mathcal{D}^{\mathrm{E}}}\left[\sum_{s \notin \mathcal{S}_h(\mathcal{D}^{\mathrm{E}}) \cup \{b_h\}} \rho(s)\right] = \sum_{s \in \mathcal{S}_\ell} \rho(s)\mathbb{P}\left(s \notin \mathcal{S}_h(\mathcal{D}^{\mathrm{E}}) \cup \{b_h\}\right) = \sum_{s \in \mathcal{S}_\ell} \rho(s)(1 - \rho(s))^N.$$

We define $\varepsilon := \sum_{s \in \mathcal{S}_\ell} \rho(s)(1 - \rho(s))^N$ and get that $\mathbb{E}_{\mathcal{D}^{\mathrm{E}}}[\mathbb{P}^{\pi^{\mathrm{BC}}}(t^u = h)] = (1 - \varepsilon)^{h-1}\varepsilon$. Substituting this result into Eq.(16) yields that

$$
\begin{aligned}
\mathbb{E}_{\mathcal{D}^{\mathrm{E}}}\left[V^{\pi^{\mathrm{E}}} - V^{\pi^{\mathrm{BC}}}\right] &= \left(1 - \frac{1}{|\mathcal{A}|}\right) \sum_{h=1}^{H} (1 - \varepsilon)^{h-1}(H - h + 1)\varepsilon \\
&\geq \left(1 - \frac{1}{|\mathcal{A}|}\right) \varepsilon \sum_{h=1}^{\lfloor H/2 \rfloor} (1 - \varepsilon)^{h-1}(H - h + 1) \\
&\geq \left(1 - \frac{1}{|\mathcal{A}|}\right) \frac{H\varepsilon}{2} \sum_{h=1}^{\lfloor H/2 \rfloor} (1 - \varepsilon)^{h-1} \\
&= \left(1 - \frac{1}{|\mathcal{A}|}\right) \frac{H}{2}\left(1 - (1 - \varepsilon)^{\lfloor H/2 \rfloor}\right)
\end{aligned}
$$

$$\geq \left(1 - \frac{1}{|\mathcal{A}|}\right) \frac{H}{2} \left(1 - \exp\left(-\varepsilon \lfloor \frac{H}{2} \rfloor\right)\right).$$

The last inequality follows that $(1+x/n)^n \leq \exp(x), \forall |x| \leq n, n \geq 1$. Besides, Lemma 4 implies that $\varepsilon \geq (S-2)/(e(N+1))$. Then we can obtain that

$$\mathbb{E}_{\mathcal{D}^{\mathrm{E}}}\left[V^{\pi^{\mathrm{E}}} - V^{\pi^{\mathrm{BC}}}\right] \geq \left(1 - \frac{1}{|\mathcal{A}|}\right) \frac{H}{2} \left(1 - \exp\left(-\frac{(S-2)}{e(N+1)} \lfloor \frac{H}{2} \rfloor\right)\right)$$

When $-\frac{(S-2)}{e(N+1)} \lfloor \frac{H}{2} \rfloor \in [-1, 0]$, according to $\exp(x) \leq 1 + x/2, \forall x \in [-1, 0]$, we have

$$\exp\left(-\frac{(S-2)}{e(N+1)} \lfloor \frac{H}{2} \rfloor\right) \leq 1 - \frac{(S-2)}{2e(N+1)} \lfloor \frac{H}{2} \rfloor.$$

Then we have that

$$\mathbb{E}_{\mathcal{D}^{\mathrm{E}}}\left[V^{\pi^{\mathrm{E}}} - V^{\pi^{\mathrm{BC}}}\right] \geq \left(1 - \frac{1}{|\mathcal{A}|}\right) \frac{(S-2)H}{4e(N+1)} \lfloor \frac{H}{2} \rfloor \gtrsim \frac{SH^2}{N}.$$

When $-\frac{(S-2)}{e(N+1)} \lfloor \frac{H}{2} \rfloor < -1$, we have that

$$\mathbb{E}_{\mathcal{D}^{\mathrm{E}}}\left[V^{\pi^{\mathrm{E}}} - V^{\pi^{\mathrm{BC}}}\right] \geq \left(1 - \frac{1}{|\mathcal{A}|}\right) \frac{H}{2} \left(1 - \frac{1}{e}\right) \gtrsim H.$$

Combining the above two inequalities finishes the proof.

## B.4. Derivation Gap in IQ-Learn

In this section, we examine why IQ-Learn, despite originating from the AIL objective, produces an optimization objective that closely resembles BC. We present the key steps in the derivation of IQ-Learn (Garg et al., 2021):

- Step I. IQ-Learn starts with the following minimax objective of AIL.

$$\min_{\pi \in \Pi} \left(\max_{r \in \mathcal{R}} \mathbb{E}_{\mathcal{D}^{\mathrm{E}}}\left[\sum_{h=1}^{H} r(s_h, a_h)\right] - \mathbb{E}_{\pi}\left[\sum_{h=1}^{H} r(s_h, a_h) + \mathcal{H}(\pi(\cdot|s_h))\right]\right).$$

  Here $\mathcal{R} = \{r : \mathcal{S} \times \mathcal{A} \to [0, 1]\}$ is the class with bounded reward functions.

- Step II. IQ-Learn transforms this primal problem into the dual formulation.

$$\max_{r \in \mathcal{R}} \left(\min_{\pi \in \Pi} \mathbb{E}_{\mathcal{D}^{\mathrm{E}}}\left[\sum_{h=1}^{H} r(s_h, a_h)\right] - \mathbb{E}_{\pi}\left[\sum_{h=1}^{H} r(s_h, a_h) + \mathcal{H}(\pi(\cdot|s_h))\right]\right). \tag{17}$$

- Step III. The inner minimization over policies in Eq. (17) corresponds to a maximum entropy RL problem with a closed-form solution. IQ-Learn plugs this closed-form solution into Eq.(17) and obtains the following single maximization objective.

$$\max_{r \in \mathcal{R}} \mathbb{E}_{\mathcal{D}^{\mathrm{E}}}\left[\sum_{h=1}^{H} r(s_h, a_h)\right] - \mathbb{E}_{s_1 \sim \rho}\left[\log\left(\sum_{a \in \mathcal{A}} \exp\left(Q^{\star, \mathrm{soft}, r}(s_1, a)\right)\right)\right]. \tag{18}$$

  Here $Q^{\star, \mathrm{soft}, r}$ is the soft-optimal Q-function regarding reward $r$.

- Step IV. IQ-Learn applies the change-of-variable $r(s_h, a_h) = Q(s_h, a_h) - \mathbb{E}_{s' \sim P(\cdot|s_h, a_h)}[\log(\sum_{a' \in \mathcal{A}} \exp(Q(s', a')))]$, yielding the following objective w.r.t Q-functions.

$$\max_{Q \in \mathcal{Q}} \mathbb{E}_{\mathcal{D}^{\mathrm{E}}}\left[\sum_{h=1}^{H} Q(s_h, a_h) - \mathbb{E}_{s' \sim P(\cdot|s_h, a_h)}\left[\log\left(\sum_{a' \in \mathcal{A}} \exp\left(Q(s', a')\right)\right)\right]\right] - \mathbb{E}_{s_1 \sim \rho}\left[\log\left(\sum_{a' \in \mathcal{A}} \exp\left(Q(s_1, a')\right)\right)\right]. \tag{19}$$

  Here $\mathcal{Q} = \{Q : \mathcal{S} \times \mathcal{A} \to [0, C]\}$ is the class of Q-functions.

We identify Step IV as the source of the equivalence breakdown. The change of variables introduces a Q-function in place of the reward, but IQ-Learn overlooks the constraint that the induced reward must remain in $\mathcal{R}$. However, IQ-Learn ignores that the new Q-variable must satisfy that the derived reward variable should satisfy the original constraints. Specifically, for any $Q \in \mathcal{Q}$, the induced reward function $r_Q(s_h, a) = Q(s_h, a) - \mathbb{E}_{s' \sim P(\cdot|s_h,a)}[\log(\sum_{a' \in \mathcal{A}} \exp(Q(s', a')))]$ must belong to $\mathcal{R}$. This requirement translates to:

$$0 \leq Q(s_h, a) - \mathbb{E}_{s' \sim P(\cdot|s_h,a)}[\text{LSE}(Q)(s')] \leq 1, \quad \forall (h, s_h, a) \in [H] \times \mathcal{S}_h \times \mathcal{A}.$$

These are precisely the Bellman constraints from Eq. (9) in our primal-dual framework. Thus, the necessity of Bellman constraints emerges naturally even when viewed through IQ-Learn's derivation idea.

## C. Omitted Results in Section 4

### C.1. Derivation of the Primal Dual Framework in Section 4.1

In this part, we derive the primal dual framework in Section 4.1 using Lagrangian duality theory. We first derive the dual V-function distribution matching formulation from the primal distribution matching.

The primal distribution matching problem is as follows:

$$\min_d \sum_{h=1}^{H} \left( D_{\text{TV}}(\widehat{d_h^{\pi^{\text{E}}}}, d_h) - \overline{H}(d_h) \right),$$

$$\text{s.t.} \sum_{a \in \mathcal{A}} d_1(s_1, a) = \rho(s_1), \forall s_1 \in \mathcal{S}_1,$$

$$\sum_{(s_h,a) \in \mathcal{S}_h \times \mathcal{A}} d_h(s_h, a) P_h(s_{h+1}|s_h, a) = \sum_{a \in \mathcal{A}} d_{h+1}(s_{h+1}, a), \forall 1 \leq h \leq H-1, \forall s_{h+1} \in \mathcal{S}_{h+1},$$

$$d_h(s_h, a) \geq 0, \forall (h, s_h, a) \in [H] \times \mathcal{S}_h \times \mathcal{A}.$$

where

$$D_{\text{TV}}(\widehat{d_h^{\pi^{\text{E}}}}, d_h) = \frac{1}{2} \sum_{(s_h,a) \in \mathcal{S}_h \times \mathcal{A}} |\widehat{d_h^{\pi^{\text{E}}}}(s_h, a) - d_h(s_h, a)| \text{ and } \overline{H}(d_h) = \mathbb{E}_{(s_h,a) \sim d_h} \left[ -\log \left( \frac{d_h(s_h, a)}{\sum_{a \in \mathcal{A}} d_h(s, a)} \right) \right].$$

Because the minimization objective $D_{\text{TV}}(\widehat{d_h^{\pi^{\text{E}}}}, d_h) - \overline{H}(d_h)$ is convex and the constraint set is a polytope, this problem is a convex problem and we can apply Lagrangian duality (Boyd et al., 2004). First, we introduce the variable $u_h(s, a)$ to eliminate the absolute value.

$$\min_{d,u} \sum_{h=1}^{H} \sum_{(s_h,a) \in \mathcal{S}_h \times \mathcal{A}} \frac{1}{2} u(s_h, a) + d_h(s_h, a) \log \left( \frac{d_h(s_h, a)}{\sum_a d_h(s_h, a)} \right),$$

$$\text{s.t.} u(s_h, a) \geq \widehat{d_h^{\pi^{\text{E}}}}(s_h, a) - d_h(s_h, a), \forall (s_h, a, h) \in \mathcal{S}_h \times \mathcal{A} \times [H],$$

$$u(s_h, a) \geq -\widehat{d_h^{\pi^{\text{E}}}}(s_h, a) + d_h(s_h, a), \forall (s_h, a, h) \in \mathcal{S}_h \times \mathcal{A} \times [H],$$

$$\sum_{a \in \mathcal{A}} d_1(s_1, a) = \rho(s_1), \forall s_1 \in \mathcal{S}_1,$$

$$\sum_{(s_h,a) \in \mathcal{S}_h \times \mathcal{A}} d_h(s_h, a) P_h(s_{h+1}|s_h, a) = \sum_{a \in \mathcal{A}} d_{h+1}(s_{h+1}, a), \forall 1 \leq h \leq H-1, \forall s_{h+1} \in \mathcal{S}_{h+1},$$

$$d_h(s_h, a) \geq 0, \forall (h, s_h, a) \in [H] \times \mathcal{S}_h \times \mathcal{A}.$$

We now introduce Lagrange multipliers for each constraint:

$$\beta(s_h, a) \geq 0 \text{ for } u(s_h, a) \geq \widehat{d_h^{\pi^{\text{E}}}}(s_h, a) - d_h(s_h, a)$$

$$\alpha(s_h, a) \geq 0 \text{ for } u(s_h, a) \geq -\widehat{d_h^{\pi^{\text{E}}}}(s_h, a) + d_h(s_h, a)$$

$$-V(s_1) \text{ for } d_1(s_1, a) = \rho(s_1)$$

$$-V(s_{h+1}) \text{ for } \sum_{(s_h,a)\in\mathcal{S}_h\times\mathcal{A}} d_h(s_h, a)P_h(s_{h+1}|s_h, a) = \sum_{a\in\mathcal{A}} d_{h+1}(s_{h+1}, a)$$

$$\lambda(s_h, a) \text{ for } d_h(s_h, a) \geq 0$$

Then we provide the Lagrange Function of the original optimization problem.

$$\mathcal{L}(d, u, \alpha, \beta, V, \lambda)$$

$$= \sum_{(s_h,a,h)\in\mathcal{S}\times\mathcal{A}\times[H]} \frac{1}{2}u(s_h, a_h) + d_h(s_h, a_h)\log\left(\frac{d_h(s_h, a_h)}{\sum_a d_h(s_h, a)}\right)$$

$$+ \sum_{(s_h,a,h)\in\mathcal{S}\times\mathcal{A}\times[H]} \beta(s_h, a)(\widehat{d_h^{\pi^E}}(s_h, a_h) - d_h(s_h, a_h) - u(s_h, a))$$

$$+ \sum_{(s_h,a,h)\in\mathcal{S}\times\mathcal{A}\times[H]} \alpha(s_h, a)(-\widehat{d_h^{\pi^E}}(s_h, a_h) + d_h(s_h, a_h) - u(s_h, a))$$

$$- \sum_{s_1\in\mathcal{S}_1} V(s_1)(\rho(s_1) - \sum_{a\in\mathcal{A}} d_1(s_1, a))$$

$$- \sum_{h=2}^{H} \sum_{s_h\in\mathcal{S}_h} V(s_h)\left(\sum_{(s_{h-1},a)\in\mathcal{S}_{h-1}\times\mathcal{A}} d_{h-1}(s_{h-1}, a)P_h(s_h|s_{h-1}, a) - \sum_{a\in\mathcal{A}} d_h(s_h, a)\right)$$

$$- \sum_{(s_h,a,h)\in\mathcal{S}\times\mathcal{A}\times[H]} \lambda(s_h, a)d_h(s_h, a)$$

Now we extract the coefficients of the variables $u_h(s_h, a), d_h(s_h, a)$ and rearrange the term:

$$\mathcal{L}(d, u, \alpha, \beta, V, \lambda)$$

$$= \sum_{(s_h,a,h)\in\mathcal{S}\times\mathcal{A}\times[H]} u(s_h, a)\left(\frac{1}{2} - \beta(s_h, a) - \alpha(s_h, a)\right)$$

$$+ \sum_{h=1}^{H-1} \sum_{(s_h,a)\in\mathcal{S}_h\times\mathcal{A}} d_h(s_h, a)[-\beta(s_h, a) + \alpha(s_h, a) + V(s_h)$$

$$- \sum_{s_{h+1}\in\mathcal{S}_{h+1}} V(s_{h+1})P_h(s_{h+1}|s_h, a) - \lambda(s_h, a) + \log(\frac{d_h(s_h, a)}{\sum_a d_h(s_h, a)})]$$

$$+ \sum_{(s_H,a)\in\mathcal{S}_H\times\mathcal{A}} d_H(s_H, a)(-\beta(s_H, a) + \alpha(s_H, a) + V(s_H) - \lambda(s_H, a) + \log(\frac{d_H(s_H, a)}{\sum_a d_H(s_H, a)}))$$

$$+ \sum_{(s_h,a,h)\in\mathcal{S}\times\mathcal{A}\times[H]} (\beta(s_h, a) - \alpha(s_h, a))\widehat{d_h^{\pi^E}}(s_h, a) - \sum_{s_1\in\mathcal{S}_1} V(s_1)\rho(s_1)$$

According to the KKT condition, we take the gradient of each original variable $u_h(s_h, a), d_h(s_h, a)$ and set them to 0.

$$\frac{\partial L(d, u, \alpha, \beta, V, \lambda)}{\partial u(s_h, a)} = \frac{1}{2} - \beta(s_h, a) - \alpha(s_h, a) = 0. \tag{20}$$

$$\forall h \in [H-1], \frac{\partial L(d, u, \alpha, \beta, V, \lambda)}{\partial d_h(s_h, a)} = \alpha(s_h, a) - \beta(s_h, a) + V(s_h) - \sum_{s_{h+1}\in\mathcal{S}_{h+1}} V(s_{h+1})P_h(s_{h+1}|s_h, a)$$

$$- \lambda(s_h, a) + \log(d_h(s_h, a)) + 1 - \log(\sum_a d_h(s_h, a)) - \sum_a \frac{d_h(s_h, a)}{\sum_a d_h(s_h, a)}$$

$$= \alpha(s_h, a) - \beta(s_h, a) + V(s_h) - \mathbb{E}_{s_{h+1}\sim P_h(\cdot|s_h, a)}[V(s_{h+1})]$$

$$- \lambda(s_h, a) + \log(\frac{d_h(s_h, a)}{\sum_a d_h(s_h, a)})$$

$$= 0 \tag{21}$$

Similarly, we have that

$$\frac{\partial L(d, u, \alpha, \beta, V, \lambda)}{\partial d_H(s_H, a)} = \alpha(s_H, a) - \beta(s_H, a) + V(s_H) - \lambda(s_H, a) + \log(\frac{d_H(s_H, a)}{\sum_a d_H(s_H, a)}) = 0. \tag{22}$$

Using the gradient of $u(s_h, a), d_h(s_h, a)$, we can get

$$\beta(s_h, a) + \alpha(s_h, a) = \frac{1}{2}$$

$$\forall h \in [H-1], \alpha(s_h, a) - \beta(s_h, a) + V(s_h) - \mathbb{E}_{s_{h+1} \sim P_h(\cdot|s_h, a)}[V(s_{h+1})] + \log(\frac{d_h(s_h, a)}{\sum_a d_h(s_h, a)}) = \lambda(s_h, a) \geq 0$$

$$\alpha(s_H, a) - \beta(s_H, a) + V(s_H) + \log(\frac{d_H(s_H, a)}{\sum_a d_H(s_H, a)}) = \lambda(s_H, a) \geq 0$$

Now we do the following derivation to eliminate $d_h(s_h, a)$.

$$\log(\frac{d_h(s_h, a)}{\sum_a d_h(s_h, a)}) \geq \beta(s_h, a) - \alpha(s_h, a) - V(s_h) + \mathbb{E}_{s_{h+1} \sim P_h(\cdot|s_h, a)}[V(s_{h+1})]$$

$$\Rightarrow \frac{d_h(s_h, a)}{\sum_a d_h(s_h, a)} \geq \exp(\beta(s_h, a) - \alpha(s_h, a) - V(s_h) + \mathbb{E}_{s_{h+1} \sim P_h(\cdot|s_h, a)}[V(s_{h+1})])$$

$$\Rightarrow 1 \geq \sum_a \exp(\beta(s_h, a) - \alpha(s_h, a) - V(s_h) + \mathbb{E}_{s_{h+1} \sim P_h(\cdot|s_h, a)}[V(s_{h+1})])$$

$$\Rightarrow \exp(V(s_h)) \geq \sum_a \exp(\beta(s_h, a) - \alpha(s_h, a) + \mathbb{E}_{s_{h+1} \sim P_h(\cdot|s_h, a)}[V(s_{h+1})])$$

$$\Rightarrow V(s_h) \geq \log(\sum_a \exp(\beta(s_h, a) - \alpha(s_h, a) + \mathbb{E}_{s_{h+1} \sim P_h(\cdot|s_h, a)}[V(s_{h+1})]))$$

The derivation for $d_H(s_H, a)$ is similar and thus we omit the derivation. Now we use $w(s_h, a) = \beta(s_h, a) - \alpha(s_h, a)$ to eliminate the variables $\beta$ and $\alpha$, and due to the equation Eq. (20) we can derive the scope of $w(s_h, a)$ is $-\frac{1}{2} \leq w_h(s_h, a) \leq \frac{1}{2}$.

Substituting the equations obtained by setting the gradients to zero back into the Lagrangian function, we can get the dual problem.

$$\max_{w,V} \sum_{(s_h, a, h) \in \mathcal{S}_h \times \mathcal{A} \times [H]} w(s_h, a) \widehat{d_h^{\pi^E}}(s_h, a) - \mathbb{E}_{s_1 \sim \rho}[V(s_1)]$$

$$\text{s.t.} V(s_h) \geq \log(\sum_a \exp(w(s_h, a) + \mathbb{E}_{s_{h+1} \sim P_h(\cdot|s_h, a)}[V(s_{h+1})])), \forall (h, s_h) \in [H-1] \times \mathcal{S}_h,$$

$$V(s_H) \geq \log(\sum_a \exp(w(s_H, a))),$$

$$-\frac{1}{2} \leq w(s_h, a) \leq \frac{1}{2}, \forall (h, s_h, a) \in [H] \times \mathcal{S}_h \times \mathcal{A}.$$

Since the optimization objective is monotonically decreasing with respect to variable $V$, we can replace the inequality with equality.

$$\max_{w,V} \sum_{(s_h, a, h) \in \mathcal{S}_h \times \mathcal{A} \times [H]} w(s_h, a) \widehat{d_h^{\pi^E}}(s_h, a) - \mathbb{E}_{s_1 \sim \rho}[V(s_1)]$$

$$\text{s.t.} V(s_h) = \log(\sum_a \exp(w(s_h, a) + \mathbb{E}_{s_{h+1} \sim P_h(\cdot|s_h, a)}[V(s_{h+1})])), \forall (h, s_h) \in [H-1] \times \mathcal{S}_h,$$

$$V(s_H) = \log(\sum_a \exp(w(s_H, a))),$$

$$-\frac{1}{2} \le w(s_h, a) \le \frac{1}{2}, \forall (h, s_h, a) \in [H] \times \mathcal{S}_h \times \mathcal{A}.$$

We introduce another two variables $r(s_h, a) = w(s_h, a) + \frac{1}{2}, V^{\text{new}}(s_h) = V(s_h) + \frac{1}{2}$. By noting that $\sum_{(s_h, a, h) \in \mathcal{S}_h \times \mathcal{A} \times [H]} r(s_h, a) \widehat{d_h^{\pi^{\text{E}}}}(s_h, a) = \mathbb{E}_{\tau \sim \mathcal{D}^{\text{E}}}[r(s_h, a_h)]$, we can get the following optimization problem:

$$\max_{r, V^{\text{new}}} \mathbb{E}_{\tau \sim \mathcal{D}^{\text{E}}}[r(s_h, a_h)] - \mathbb{E}_{s_1 \sim \rho}[V^{\text{new}}(s_1)]$$

$$\text{s.t.} V^{\text{new}}(s_h) = \log(\sum_a \exp(r(s_h, a) + \mathbb{E}_{s_{h+1} \sim P_h(\cdot|s_h, a)}[V^{\text{new}}(s_{h+1})])), \forall (h, s_h) \in [H-1] \times \mathcal{S}_h,$$

$$V^{\text{new}}(s_H) = \log(\sum_a \exp(r(s_H, a))),$$

$$0 \le r(s_h, a) \le 1, \forall (h, s_h, a) \in [H] \times \mathcal{S}_h \times \mathcal{A}.$$

Up to now, we have derived the dual V-function distribution matching problem in Eq. (7). Then we start to derive the dual Q-function distribution matching problem in Eq. (8). In particular, we introduce the Q-function:

$$\forall (s_h, a, h) \in \mathcal{S}_h \times \mathcal{A} \times [H-1], Q(s_h, a) = r(s_h, a) + \mathbb{E}_{s_{h+1} \sim P_h(\cdot|s_h, a)}[V^{\text{new}}(s_{h+1})], Q(s_H, a) = r(s_H, a).$$

We plug the above equation into the formulation of dual V-function distribution matching, yielding dual Q-function distribution matching in Eq. (8).

$$\max_Q \mathbb{E}_{\tau \sim \mathcal{D}^{\text{E}}}\left[\sum_{h=1}^H Q(s_h, a_h) - \mathbb{E}_{s' \sim P(\cdot|s_h, a_h)}\big[\text{LSE}(Q)(s')\big]\right] - \mathbb{E}_{s_1 \sim \rho}\big[\text{LSE}(Q)(s_1)\big]$$

$$\text{s.t. } 0 \le Q(s_h, a) - \mathbb{E}_{s' \sim P(\cdot|s_h, a)}\big[\text{LSE}(Q)(s')\big] \le 1, (h, s_h, a) \in [H] \times \mathcal{S}_h \times \mathcal{A}.$$

### C.2. Proof of Theorem 2

To prove Theorem 2, we establish the equivalence among AIL in Eq. (2), Dual V-DM in Eq. (7) and Dual Q-DM in Eq. (8).

**Proposition 2.** *Suppose that $(\widehat{r}, \widehat{V})$ is the optimal solution to Dual V-DM in Eq. (7) and $\widehat{\pi}$ is the deriving policy, i.e.,*

$$\forall h \in [H], s_h \in \mathcal{S}_h, a \in \mathcal{A}, \widehat{\pi}(a|s_h) \propto \exp\left(\widehat{r}(s_h, a) + \mathbb{E}_{s' \sim P(\cdot|s_h, a)}\left[\widehat{V}(s')\right]\right).$$

*Then $\widehat{\pi}$ is the optimal solution to AIL in Eq. (2).*

*Proof.* Let $(\widehat{r}, \widehat{V})$ be the optimal solution to the Dual V-DM problem in Eq. (7). Let $\mathcal{R} = \{r : r(s, a) \in [0, 1], \forall (s, a) \in \mathcal{S} \times \mathcal{A}\}$ denote the set of bounded reward functions.

In the Dual V-DM problem, the constraints imposed on $V$,

$$V(s_h) = \text{LSE}(r + PV)(s_h), \quad \forall (h, s_h) \in [H] \times \mathcal{S}_h,$$

imply that $V$ is exactly the soft optimal value function with respect to the reward $r$, i.e., $V = V^{\star, \text{soft}, r}$. Consequently, the expected initial value can be equivalently evaluated by solving the standard entropy-regularized reinforcement learning problem:

$$\mathbb{E}_{s_1 \sim \rho}[V(s_1)] = \max_{\pi \in \Pi} \mathbb{E}_{\tau \sim \pi}\left[\sum_{h=1}^H (r(s_h, a_h) + \mathcal{H}(\pi(\cdot|s_h)))\right]. \tag{23}$$

Substituting Eq. (23) back into Eq. (7), we can reformulate the Dual V-DM problem as the following max-min optimization problem:

$$\max_{r \in \mathcal{R}} \min_{\pi \in \Pi} \left(\mathbb{E}_{\tau \sim \mathcal{D}^{\text{E}}}\left[\sum_{h=1}^H r(s_h, a_h)\right] - \mathbb{E}_{\tau \sim \pi}\left[\sum_{h=1}^H (r(s_h, a_h) + \mathcal{H}(\pi(\cdot|s_h)))\right]\right). \tag{24}$$

Let $\phi(\pi, r)$ denote the objective function inside the parentheses of Eq. (24). The optimal reward $\widehat{r}$ is the solution to the outer maximization of Eq. (24), i.e., $\widehat{r} = \text{argmax}_{r \in \mathcal{R}} \underline{\phi}(r)$, where $\underline{\phi}(r) = \min_{\pi \in \Pi} \phi(\pi, r)$. Furthermore, the deriving policy $\widehat{\pi}$ is formulated by

$$\widehat{\pi}(a|s_h) = \frac{\exp\left(\widehat{r}(s_h, a) + \mathbb{E}_{s' \sim P(\cdot|s_h, a)}\left[\widehat{V}(s')\right]\right)}{\sum_{a' \in \mathcal{A}} \exp\left(\widehat{r}(s_h, a') + \mathbb{E}_{s' \sim P(\cdot|s_h, a')}\left[\widehat{V}(s')\right]\right)} = \frac{Q^{\star, \text{soft}, \widehat{r}}(s, a)}{\sum_{a' \in \mathcal{A}} Q^{\star, \text{soft}, \widehat{r}}(s, a')},$$

indicating that $\widehat{\pi}$ is the soft optimal policy regarding $\widehat{r}$. Therefore, we have that $\widehat{\pi}$ is the solution to the inner minimization of Eq. (24) given $\widehat{r}$, and this solution is unique due to the property of maximum entropy RL.

The AIL problem in Eq. (2) is exactly the min-max counterpart: $\min_{\pi \in \Pi} \max_{r \in \mathcal{R}} \phi(\pi, r)$. Let $\pi^{\text{AIL}} \in \text{argmin}_{\pi \in \Pi} \overline{\phi}(\pi), \overline{\phi}(\pi) = \max_{r \in \mathcal{R}} \phi(\pi, r)$ be the AIL's solution. Our target is to prove that $\widehat{\pi} = \pi^{\text{AIL}}$. We first prove that $(\pi^{\text{AIL}}, \widehat{r})$ is the saddle point of the min-max problem using the minimax theorem (Sion, 1958). In particular, we transition from the policy space to the state-action visitation distribution space (occupancy measure) D, where

$$\text{D} = \{\forall (h, s_h, a) \in [H] \times \mathcal{S}_h \in \mathcal{A}, d_h(s_h, a) \geq 0,$$
$$\forall (h, s_{h+1}) \in [H-1] \times \mathcal{S}_{h+1}, \sum_{s_h \in \mathcal{S}_h} \sum_{a \in \mathcal{A}} d_h(s_h, a) P(s_{h+1}|s_h, a) = d_{h+1}(s_{h+1})\}.$$

It is a well-known result that there exists a bijective mapping between the occupancy measure $d$ and the policy $\pi$ (Syed et al., 2008). Then we rewrite the objective function as

$$\phi(\pi, r) = \sum_{h=1}^{H} \sum_{(s_h, a) \in \mathcal{S}_h \times \mathcal{A}} \left(\widehat{d_h^{\pi^{\text{E}}}}(s_h, a) r(s_h, a)\right) - \sum_{h=1}^{H} \sum_{(s_h, a) \in \mathcal{S}_h \times \mathcal{A}} \left(d_h^{\pi}(s_h, a) r(s_h, a) - \overline{H}(d_h^{\pi})\right).$$

Accordingly, we introduce the objective function in the occupancy measure space,

$$J(d, r) = \sum_{h=1}^{H} \sum_{(s_h, a) \in \mathcal{S}_h \times \mathcal{A}} \left(\widehat{d_h^{\pi^{\text{E}}}}(s_h, a) r(s_h, a)\right) - \sum_{h=1}^{H} \sum_{(s_h, a) \in \mathcal{S}_h \times \mathcal{A}} \left(d_h(s_h, a) r(s_h, a) - \overline{H}(d_h)\right),$$

and consider the min-max problem in the occupancy measure space $\min_{d \in \text{D}} \max_{r \in \mathcal{R}}$. Observe that $J(d, r)$ is linear (and thus concave) in $r$ and strictly convex in $d^{\pi}$ according to (Ho & Ermon, 2016, Lemma 3.1). Since the feasible domains D and $\mathcal{R}$ are both compact and convex, Sion's minimax theorem (Sion, 1958) applies, yielding strong duality:

$$\min_{d \in \text{D}} \max_{r \in \mathcal{R}} J(d, r) = \max_{r \in \mathcal{R}} \min_{d \in \text{D}} J(d, r). \tag{25}$$

This directly implies strong duality in the policy space.

$$\min_{\pi \in \Pi} \max_{r \in \mathcal{R}} \phi(\pi, r) = \min_{d \in \text{D}} \max_{r \in \mathcal{R}} J(d, r) = \max_{r \in \mathcal{R}} \min_{d \in \text{D}} J(d, r) = \max_{r \in \mathcal{R}} \min_{\pi \in \Pi} \phi(\pi, r).$$

Recall that $\pi^{\text{AIL}} \in \text{argmin}_{\pi \in \Pi} \overline{\phi}(\pi)$ and $\widehat{r} \in \text{argmax}_{r \in \mathcal{R}} \underline{\phi}(r)$. Due to strong duality, we have that $(\pi^{\text{AIL}}, \widehat{r})$ constitutes a saddle point for $\phi(\pi, r)$, which implies:

$$\pi^{\text{AIL}} \in \underset{\pi \in \Pi}{\text{argmin}} \, \phi(\pi, \widehat{r}).$$

Recall that given $\widehat{r}$, $\text{argmin}_{\pi \in \Pi} \phi(\pi, \widehat{r})$ is a maximum entropy RL problem, which admits a unique optimal solution $\widehat{\pi}$. Then we have that $\pi^{\text{AIL}} = \widehat{\pi}$, which finishes the proof. $\square$

**Proposition 3.** *Suppose that $\widehat{Q}$ is the optimal solution to Dual Q-DM in Eq. (8), and $\widehat{r}$ and $\widehat{V}$ are the deriving reward and V-function, respectively, i.e.,*

$$\forall h \in [H], s_h \in \mathcal{S}_h, a \in \mathcal{A}, \widehat{r}(s_h, a) = \widehat{Q}(s_h, a) - \mathbb{E}_{s' \sim P(\cdot|s_h, a)}\left[\log\left(\sum_{a' \in \mathcal{A}} \exp\left(\widehat{Q}(s', a')\right)\right)\right],$$

$$\widehat{V}(s_h) = \log\left(\sum_{a \in \mathcal{A}} \exp\left(\widehat{Q}(s_h, a)\right)\right).$$

*Then $(\widehat{r}, \widehat{V})$ is the optimal solution to Dual V-DM in Eq. (7).*

*Proof.* We first show that $(\widehat{r}, \widehat{V})$ is a feasible solution to Dual V-DM in Eq. (7). According to Bellman constraints in Eq. (9), we have that $0 \leq \widehat{r}(s_h, a) \leq 1, \forall (h, s_h, a) \in [H] \times \mathcal{S}_h \times \mathcal{A}$. Besides, we have that $\widehat{Q}$ is the soft optimal Q-function regarding $\widehat{r}$ and $\widehat{V}$ is the corresponding soft optimal V-function. This implies that $\widehat{V}$ satisfies the soft Bellman optimality equation regarding $\widehat{r}$, i.e.,

$$\widehat{V}(s_h) = \mathrm{LSE}(\widehat{r} + P\widehat{V})(s_h), \forall (h, s_h) \in [H] \times \mathcal{S}_h.$$

Together, we have that $(\widehat{r}, \widehat{V})$ is a feasible solution to Dual V-DM in Eq. (7).

Then we prove that $(\widehat{r}, \widehat{V})$ is an optimal solution to Dual V-DM in Eq. (7). According to the connection between $\widehat{Q}$ and $(\widehat{r}, \widehat{V})$, we have that

$$\mathbb{E}_{\tau \sim \mathcal{D}^{\mathrm{E}}} \left[ \sum_{h=1}^{H} \widehat{r}(s_h, a_h) \right] - \mathbb{E}_{s_1 \sim \rho} \left[ \widehat{V}(s_1) \right]$$

$$= \mathbb{E}_{\mathcal{D}^{\mathrm{E}}} \left[ \sum_{h=1}^{H} \widehat{Q}(s_h, a_h) - \mathbb{E}_{s' \sim P(\cdot|s_h, a_h)} \left[ \log \left( \sum_{a' \in \mathcal{A}} \exp \left( \widehat{Q}_{h+1}(s', a') \right) \right) \right] \right]$$

$$- \mathbb{E}_{s_1 \sim \rho} \left[ \log \left( \sum_{a' \in \mathcal{A}} \exp \left( \widehat{Q}(s_1, a') \right) \right) \right].$$

For any feasible solution $(r, V)$ to Dual V-DM in Eq. (7), we define the corresponding Q-function as

$$\forall (h, s_h, a) \in [H] \times \mathcal{S}_h \times \mathcal{A}, Q(s_h, a) = r(s_h, a) + \mathbb{E}_{s' \sim P_h(\cdot|s_h, a)} \left[ V(s') \right].$$

Notice that $V$ is feasible and thus satisfies that

$$\forall (h, s_h) \in [H] \times \mathcal{S}_h, V(s_h) = \log \left( \sum_{a \in \mathcal{A}} \exp \left( r(s_h, a) + \mathbb{E}_{s' \sim P(\cdot|s_h, a)} \left[ V(s') \right] \right) \right) = \log \left( \sum_{a \in \mathcal{A}} \exp \left( Q(s_h, a) \right) \right).$$

Then we can have that $\forall (h, s_h, a) \in [H] \times \mathcal{S}_h \times \mathcal{A}$,

$$Q(s_h, a) - \mathbb{E}_{s' \sim P(\cdot|s_h, a)} \left[ \log \left( \sum_{a' \in \mathcal{A}} \exp \left( Q(s', a') \right) \right) \right] = Q(s_h, a) - \mathbb{E}_{s' \sim P(\cdot|s_h, a)} \left[ V(s') \right] = r(s_h, a) \in [0, 1].$$

Therefore, $Q$ is feasible w.r.t Dual Q-DM in Eq. (8). Then for any feasible solution $(r, V)$ to Dual V-DM in Eq. (7), it holds that

$$\mathbb{E}_{\tau \sim \mathcal{D}^{\mathrm{E}}} \left[ \sum_{h=1}^{H} \widehat{r}(s_h, a_h) \right] - \mathbb{E}_{s_1 \sim \rho} \left[ \widehat{V}(s_1) \right]$$

$$= \mathbb{E}_{\mathcal{D}^{\mathrm{E}}} \left[ \sum_{h=1}^{H} \widehat{Q}(s_h, a_h) - \mathbb{E}_{s' \sim P(\cdot|s_h, a_h)} \left[ \log \left( \sum_{a' \in \mathcal{A}} \exp \left( \widehat{Q}(s', a') \right) \right) \right] \right] - \mathbb{E}_{s_1 \sim \rho} \left[ \log \left( \sum_{a' \in \mathcal{A}} \exp \left( \widehat{Q}(s_1, a') \right) \right) \right]$$

$$\overset{(a)}{\geq} \mathbb{E}_{\mathcal{D}^{\mathrm{E}}} \left[ \sum_{h=1}^{H} Q(s_h, a_h) - \mathbb{E}_{s' \sim P(\cdot|s_h, a_h)} \left[ \log \left( \sum_{a' \in \mathcal{A}} \exp \left( Q(s', a') \right) \right) \right] \right] - \mathbb{E}_{s_1 \sim \rho} \left[ \log \left( \sum_{a' \in \mathcal{A}} \exp \left( Q(s_1, a') \right) \right) \right]$$

$$= \mathbb{E}_{\tau \sim \mathcal{D}^{\mathrm{E}}} \left[ \sum_{h=1}^{H} r_h(s_h, a_h) \right] - \mathbb{E}_{s_1 \sim \rho} \left[ V(s_1) \right].$$

Inequality (a) follows that $\widehat{Q}$ is the optimal solution to Dual Q-DM in Eq. (8) and $Q$ is a feasible solution. This implies that $(\widehat{r}, \widehat{V})$ is an optimal solution to Dual V-DM in Eq. (7), which finishes the proof. $\square$

With Proposition 2 and Proposition 3, we can now prove Theorem 2.

*Proof of Theorem 2.* Suppose that $\widetilde{Q}$ is the optimal solution to Dual Q-DM in Eq. (8), we define that Proposition 3:

$$\forall h \in [H], s_h \in \mathcal{S}_h, a \in \mathcal{A}, \widetilde{r}(s_h, a) = \widetilde{Q}(s_h, a) - \mathbb{E}_{s' \sim P(\cdot|s_h, a)} \left[ \log \left( \sum_{a' \in \mathcal{A}} \exp \left( \widetilde{Q}(s', a') \right) \right) \right],$$

$$\widetilde{V}(s_h) = \log \left( \sum_{a \in \mathcal{A}} \exp \left( \widetilde{Q}(s_h, a) \right) \right).$$

According to Proposition 3, $(\widetilde{r}, \widetilde{V})$ is the optimal solution to Dual V-DM. Proposition 2 further implies that

$$\forall h \in [H], s_h \in \mathcal{S}_h, a \in \mathcal{A}, \widetilde{\pi}(a|s_h) \propto \exp \left( \widetilde{r}(s_h, a) + \mathbb{E}_{s' \sim P(\cdot|s_h, a)} \left[ \widetilde{V}(s') \right] \right)$$

is the optimal solution to AIL in Eq. (2). Since $\widetilde{r}(s_h, a) = \widetilde{Q}(s_h, a) - \mathbb{E}_{s' \sim P(\cdot|s_h, a)} \left[ \log \left( \sum_{a' \in \mathcal{A}} \exp \left( \widetilde{Q}(s', a') \right) \right) \right] = \widetilde{Q}(s_h, a) - \mathbb{E}_{s' \sim P(\cdot|s_h, a)} \left[ \widetilde{V}(s') \right]$, $\widetilde{\pi}$ satisfies that

$$\forall h \in [H], s_h \in \mathcal{S}_h, a \in \mathcal{A}, \widetilde{\pi}(a|s_h) \propto \exp \left( \widetilde{Q}(s_h, a) \right),$$

which is exactly the softmax policy $\pi_{\widetilde{Q}} = \text{softmax}(\widetilde{Q})$. This implies that $\pi_{\widetilde{Q}}$ is the optimal solution to AIL in Eq. (2), which finishes the proof.

$\square$

### C.3. Proof of Proposition 1

To prove Proposition 1, we first establish the following technical lemma.

**Lemma 1.** *Suppose that $\widetilde{Q}$ is the Q-function learned by Dual Q-DM via Eq. (8). It holds that*

- $\forall h \in [H], \exists (s_h^{\mathrm{E}}, a_h^{\mathrm{E}}) \in \mathcal{D}^{\mathrm{E}}$ *such that* $\widetilde{Q}(s_h^{\mathrm{E}}, a_h^{\mathrm{E}}) - \mathbb{E}_{s' \sim P(\cdot|s_h^{\mathrm{E}}, a_h^{\mathrm{E}})} \left[ \mathrm{LSE}(\widetilde{Q})(s') \right] = 1$.

- $\forall h \in [H], \forall (s_h, a_h) \notin \mathcal{D}^{\mathrm{E}}$ *such that* $\widetilde{Q}(s_h, a_h) - \mathbb{E}_{s' \sim P(\cdot|s_h, a_h)} \left[ \mathrm{LSE}(\widetilde{Q})(s') \right] = 0$.

The proof can be found in Section E.2. Now, we proceed to prove Proposition 1.

*Proof of Proposition 1.* Suppose that $\widetilde{Q}$ is the Q-function recovered by Dual Q-DM, and $r_{\widetilde{Q}}(s_h, a_h) := \widetilde{Q}(s_h, a_h) - \mathbb{E}_{s' \sim P(\cdot|s_h, a_h)}[\mathrm{LSE}(\widetilde{Q})(s')]$ and $\widetilde{V}(s_h) = \mathrm{LSE}(\widetilde{Q})(s_h)$ are the derived V-functions and reward functions, respectively. We use backward mathematical induction to prove that the solution of Dual Q-DM satisfies the following two properties:

1. $\forall h \in [H-1], s_h \in \mathcal{S}_h, a \neq \pi^{\mathrm{E}}(s_h), \widetilde{Q}(s_h, \pi^{\mathrm{E}}(s_h)) > \widetilde{Q}(s_h, a)$.

2. $\forall h \in [H-1], s_h^{\mathrm{E}} \in \mathcal{S}_h^{\mathrm{E}}, s_h \notin \mathcal{S}_h^{\mathrm{E}}, \widetilde{V}(s_h^{\mathrm{E}}) \geq \widetilde{V}(s_h)$ and $\exists \widehat{s_h^{\mathrm{E}}} \in \mathcal{S}_h^{\mathrm{E}}$ such that $\forall s_h \notin \mathcal{S}_h^{\mathrm{E}}, \widetilde{V}(\widehat{s_h^{\mathrm{E}}}) > \widetilde{V}(s_h)$. Here $\mathcal{S}_{h+1}^{\mathrm{E}} = \{s_{h+1} : \exists s_h \in \mathcal{S}_h, P(s_{h+1}|s_h, \pi^{\mathrm{E}}(s_h)) > 0\}$ and $\mathcal{S}_1^{\mathrm{E}} = \{s_1 \in \mathcal{S}_1 : \rho(s_1) > 0\}$ denote the set of expert-reachable states.

**Initialization (at time step $H$):** Before proceeding to the base case at time step $H-1$, we first show that property (2) holds at time step $H$. According to Lemma 1, $\forall s_H \notin \mathcal{S}_H^{\mathrm{E}}, r_{\widetilde{Q}}(s_H, a) = 0$, thus

$$\widetilde{V}(s_H) = \log \left( \sum_a \exp(r_{\widetilde{Q}}(s_H, a)) \right) = \log(|\mathcal{A}|).$$

That is, the V-function attains the lowest value of $\log(|\mathcal{A}|)$ at any state uncovered by demonstrations. Meanwhile, Lemma 1 shows that $\exists s_H^{\mathrm{E}} \in \mathcal{S}_H^{\mathrm{E}}, r_{\widetilde{Q}}(s_H^{\mathrm{E}}, a) = 1$, which implies $\widetilde{V}(s_H^{\mathrm{E}}) > \log(|\mathcal{A}|)$. Therefore, at time step $H$, we have $\widetilde{V}(s_H^{\mathrm{E}}) \geq \widetilde{V}(s_H)$ for all $s_H^{\mathrm{E}} \in \mathcal{S}_H^{\mathrm{E}}, s_H \notin \mathcal{S}_H^{\mathrm{E}}$, and there strictly exists $\widehat{s_H^{\mathrm{E}}} \in \mathcal{S}_H^{\mathrm{E}}$ such that $\widetilde{V}(\widehat{s_H^{\mathrm{E}}}) > \widetilde{V}(s_H)$ for all $s_H \notin \mathcal{S}_H^{\mathrm{E}}$.

**Base Case (at time step $H-1$):** We first prove that property (1) holds at time step $H-1$. Notice that $\widetilde{Q}$ satisfies the following Bellman optimality equation.

$$\widetilde{Q}(s_{H-1}, a^{\mathrm{E}}) = r_{\widetilde{Q}}(s_{H-1}, a^{\mathrm{E}}) + \mathbb{E}_{s' \sim P(\cdot|s_{H-1}, a^{\mathrm{E}})}[\widetilde{V}(s')],$$
$$\widetilde{Q}(s_{H-1}, a) = r_{\widetilde{Q}}(s_{H-1}, a) + \mathbb{E}_{s' \sim P(\cdot|s_{H-1}, a)}[\widetilde{V}(s')], \quad \forall a \neq a^{\mathrm{E}}.$$

Lemma 1 indicates that $\forall a \neq a^{\mathrm{E}}, r_{\widetilde{Q}}(s_{H-1}, a^{\mathrm{E}}) \geq r_{\widetilde{Q}}(s_{H-1}, a) = 0$. Then we have:

$$\widetilde{Q}(s_{H-1}, a^{\mathrm{E}}) - \widetilde{Q}(s_{H-1}, a)$$
$$\geq \mathbb{E}_{s' \sim P(\cdot|s_{H-1}, a^{\mathrm{E}})}[\widetilde{V}(s')] - \mathbb{E}_{s' \sim P(\cdot|s_{H-1}, a)}[\widetilde{V}(s')]$$
$$= \sum_{s' \in \mathcal{S}_H^{\mathrm{E}}} P(s'|s_{H-1}, a^{\mathrm{E}}) \widetilde{V}(s') - \sum_{s' \in \mathcal{S}_H} P(s'|s_{H-1}, a) \widetilde{V}(s')$$
$$= P(\widehat{s_H^{\mathrm{E}}}|s_{H-1}, a^{\mathrm{E}}) \widetilde{V}(\widehat{s_H^{\mathrm{E}}}) + \sum_{\substack{s_H^{\mathrm{E}} \in \mathcal{S}_H^{\mathrm{E}} \\ s_H^{\mathrm{E}} \neq \widehat{s_H^{\mathrm{E}}}}} P(s_H^{\mathrm{E}}|s_{H-1}, a^{\mathrm{E}}) \widetilde{V}(s_H^{\mathrm{E}})$$
$$- P(\widehat{s_H^{\mathrm{E}}}|s_{H-1}, a) \widetilde{V}(\widehat{s_H^{\mathrm{E}}}) - \sum_{\substack{s_H^{\mathrm{E}} \in \mathcal{S}_H^{\mathrm{E}} \\ s_H^{\mathrm{E}} \neq \widehat{s_H^{\mathrm{E}}}}} P(s_H^{\mathrm{E}}|s_{H-1}, a) \widetilde{V}(s_H^{\mathrm{E}}) - \sum_{s_H \notin \mathcal{S}_H^{\mathrm{E}}} P(s_H|s_{H-1}, a) \widetilde{V}(s_H)$$
$$= \widetilde{V}(\widehat{s_H^{\mathrm{E}}}) \left( P(\widehat{s_H^{\mathrm{E}}}|s_{H-1}, a^{\mathrm{E}}) - P(\widehat{s_H^{\mathrm{E}}}|s_{H-1}, a) \right)$$
$$+ \sum_{\substack{s_H^{\mathrm{E}} \in \mathcal{S}_H^{\mathrm{E}} \\ s_H^{\mathrm{E}} \neq \widehat{s_H^{\mathrm{E}}}}} \left( P(s_H^{\mathrm{E}}|s_{H-1}, a^{\mathrm{E}}) - P(s_H^{\mathrm{E}}|s_{H-1}, a) \right) \widetilde{V}(s_H^{\mathrm{E}}) - \sum_{s_H \notin \mathcal{S}_H^{\mathrm{E}}} P(s_H|s_{H-1}, a) \widetilde{V}(s_H)$$
$$\overset{(a)}{\geq} \widetilde{V}(\widehat{s_H^{\mathrm{E}}}) \left( P(\widehat{s_H^{\mathrm{E}}}|s_{H-1}, a^{\mathrm{E}}) - P(\widehat{s_H^{\mathrm{E}}}|s_{H-1}, a) \right)$$
$$+ \left( \min_{\substack{s_H^{\mathrm{E}} \in \mathcal{S}_H^{\mathrm{E}} \\ s_H^{\mathrm{E}} \neq \widehat{s_H^{\mathrm{E}}}}} \widetilde{V}(s_H^{\mathrm{E}}) \right) \sum_{\substack{s_H^{\mathrm{E}} \in \mathcal{S}_H^{\mathrm{E}} \\ s_H^{\mathrm{E}} \neq \widehat{s_H^{\mathrm{E}}}}} \left( P(s_H^{\mathrm{E}}|s_{H-1}, a^{\mathrm{E}}) - P(s_H^{\mathrm{E}}|s_{H-1}, a) \right) - \left( \max_{s_H \notin \mathcal{S}_H^{\mathrm{E}}} \widetilde{V}(s_H) \right) \sum_{s_H \notin \mathcal{S}_H^{\mathrm{E}}} P(s_H|s_{H-1}, a)$$
$$\overset{(b)}{\geq} \widetilde{V}(\widehat{s_H^{\mathrm{E}}}) \left( P(\widehat{s_H^{\mathrm{E}}}|s_{H-1}, a^{\mathrm{E}}) - P(\widehat{s_H^{\mathrm{E}}}|s_{H-1}, a) \right)$$
$$+ \left( \max_{s_H \notin \mathcal{S}_H^{\mathrm{E}}} \widetilde{V}(s_H) \right) \sum_{\substack{s_H^{\mathrm{E}} \in \mathcal{S}_H^{\mathrm{E}} \\ s_H^{\mathrm{E}} \neq \widehat{s_H^{\mathrm{E}}}}} \left( P(s_H^{\mathrm{E}}|s_{H-1}, a^{\mathrm{E}}) - P(s_H^{\mathrm{E}}|s_{H-1}, a) \right) - \left( \max_{s_H \notin \mathcal{S}_H^{\mathrm{E}}} \widetilde{V}(s_H) \right) \sum_{s_H \notin \mathcal{S}_H^{\mathrm{E}}} P(s_H|s_{H-1}, a)$$
$$= \widetilde{V}(\widehat{s_H^{\mathrm{E}}}) \left( P(\widehat{s_H^{\mathrm{E}}}|s_{H-1}, a^{\mathrm{E}}) - P(\widehat{s_H^{\mathrm{E}}}|s_{H-1}, a) \right)$$
$$+ \max_{s_H \notin \mathcal{S}_H^{\mathrm{E}}} \widetilde{V}(s_H) \left( \sum_{\substack{s_H^{\mathrm{E}} \in \mathcal{S}_H^{\mathrm{E}} \\ s_H^{\mathrm{E}} \neq \widehat{s_H^{\mathrm{E}}}}} \left( P(s_H^{\mathrm{E}}|s_{H-1}, a^{\mathrm{E}}) - P(s_H^{\mathrm{E}}|s_{H-1}, a) \right) - \sum_{s_H \notin \mathcal{S}_H^{\mathrm{E}}} P(s_H|s_{H-1}, a) \right)$$
$$\overset{(c)}{=} \delta \left( \widetilde{V}(\widehat{s_H^{\mathrm{E}}}) - \max_{s_H \notin \mathcal{S}_H^{\mathrm{E}}} \widetilde{V}(s_H) \right)$$
$$\overset{(d)}{>} 0.$$

Inequality (a) holds because in TD MDPs, $\forall h \in [H-1], s_h \in \mathcal{S}_h, s_{h+1}^{\mathrm{E}} \in \mathcal{S}_{h+1}^{\mathrm{E}}, P(s_{h+1}^{\mathrm{E}}|s_h, \pi^{\mathrm{E}}(s_h)) > P(s_{h+1}^{\mathrm{E}}|s_h, a), \forall a \neq \pi^{\mathrm{E}}(s_h)$. Inequality (b) holds because we have established that $\widetilde{V}(s_H^{\mathrm{E}}) \geq \widetilde{V}(s_H)$ for all

$s_H^{\mathrm{E}} \in \mathcal{S}_H^{\mathrm{E}}, s_H \notin \mathcal{S}_H^{\mathrm{E}}$. For equation (c), we define $\delta = P(\widehat{s_H^{\mathrm{E}}}|s_{H-1}, a^{\mathrm{E}}) - P(\widehat{s_H^{\mathrm{E}}}|s_{H-1}, a)$. Because the sum of transition probabilities equals 1, we know that:

$$\delta + \sum_{\substack{s_H^{\mathrm{E}} \in \mathcal{S}_H^{\mathrm{E}} \\ s_H^{\mathrm{E}} \neq \widehat{s_H^{\mathrm{E}}}}} \left( P(s_H^{\mathrm{E}}|s_{H-1}, a^{\mathrm{E}}) - P(s_H^{\mathrm{E}}|s_{H-1}, a) \right) - \sum_{s_H \notin \mathcal{S}_H^{\mathrm{E}}} P(s_H|s_{H-1}, a) = 1 - 1 = 0,$$

which seamlessly justifies substitution (c). To prove the strict inequality (d), we have both $\delta = P(\widehat{s_H^{\mathrm{E}}}|s_{H-1}, a^{\mathrm{E}}) - P(\widehat{s_H^{\mathrm{E}}}|s_{H-1}, a) > 0$ due to property of TD MDPs and $\forall s_H \notin \mathcal{S}_H^{\mathrm{E}}, \widetilde{V}(\widehat{s_H^{\mathrm{E}}}) > \widetilde{V}(s_H)$ and $\delta > 0$ due to our analysis at the initialization stage. Thus, property (1) holds at step $H - 1$.

Next, we verify property (2) at step $H - 1$. By definition, $\widetilde{V}(s_h) = \log \left( \sum_{a \in \mathcal{A}} \exp(\widetilde{Q}(s_h, a)) \right)$. We prove property (2) by showing that $\forall s_{H-1}^{\mathrm{E}} \in \mathcal{S}_{H-1}^{\mathrm{E}}, s_{H-1} \notin \mathcal{S}_{H-1}^{\mathrm{E}}, a \in \mathcal{A}, \widetilde{Q}(s_{H-1}^{\mathrm{E}}, a) \geq \widetilde{Q}(s_{H-1}, a)$. Using the definition of the Q-function and knowing $r_{\widetilde{Q}}(s_{H-1}, a) = 0$, we have that

$$
\begin{aligned}
&\widetilde{Q}(s_{H-1}^{\mathrm{E}}, a) - \widetilde{Q}(s_{H-1}, a) \\
&\overset{(a)}{\geq} \mathbb{E}_{s' \sim P(\cdot|s_{H-1}^{\mathrm{E}}, a)}[\widetilde{V}(s')] - \mathbb{E}_{s' \sim P(\cdot|s_{H-1}, a)}[\widetilde{V}(s')] \\
&= \sum_{s_H^{\mathrm{E}} \in \mathcal{S}_H^{\mathrm{E}}} P(s_H^{\mathrm{E}}|s_{H-1}^{\mathrm{E}}, a)\widetilde{V}(s_H^{\mathrm{E}}) - \sum_{s_H \in \mathcal{S}_H} P(s_H|s_{H-1}, a)\widetilde{V}(s_H) \\
&= \sum_{s_H^{\mathrm{E}} \in \mathcal{S}_H^{\mathrm{E}}} P(s_H^{\mathrm{E}}|s_{H-1}^{\mathrm{E}}, a)\widetilde{V}(s_H^{\mathrm{E}}) - \sum_{s_H^{\mathrm{E}} \in \mathcal{S}_H^{\mathrm{E}}} P(s_H^{\mathrm{E}}|s_{H-1}, a)\widetilde{V}(s_H^{\mathrm{E}}) - \sum_{s_H \notin \mathcal{S}_H^{\mathrm{E}}} P(s_H|s_{H-1}, a)\widetilde{V}(s_H) \\
&= \sum_{s_H^{\mathrm{E}} \in \mathcal{S}_H^{\mathrm{E}}} \left( P(s_H^{\mathrm{E}}|s_{H-1}^{\mathrm{E}}, a) - P(s_H^{\mathrm{E}}|s_{H-1}, a) \right) \widetilde{V}(s_H^{\mathrm{E}}) - \sum_{s_H \notin \mathcal{S}_H^{\mathrm{E}}} P(s_H|s_{H-1}, a)\widetilde{V}(s_H) \\
&\geq \left( \min_{s_H^{\mathrm{E}} \in \mathcal{S}_H^{\mathrm{E}}} \widetilde{V}(s_H^{\mathrm{E}}) \right) \sum_{s_H^{\mathrm{E}} \in \mathcal{S}_H^{\mathrm{E}}} \left( P(s_H^{\mathrm{E}}|s_{H-1}^{\mathrm{E}}, a) - P(s_H^{\mathrm{E}}|s_{H-1}, a) \right) - \left( \max_{s_H \notin \mathcal{S}_H^{\mathrm{E}}} \widetilde{V}(s_H) \right) \sum_{s_H \notin \mathcal{S}_H^{\mathrm{E}}} P(s_H|s_{H-1}, a) \\
&\overset{(b)}{=} \gamma \left( \min_{s_H^{\mathrm{E}} \in \mathcal{S}_H^{\mathrm{E}}} \widetilde{V}(s_H^{\mathrm{E}}) - \max_{s_H \notin \mathcal{S}_H^{\mathrm{E}}} \widetilde{V}(s_H) \right) \\
&\overset{(c)}{\geq} 0.
\end{aligned}
$$

Inequality (a) holds because Lemma 1 shows that the induced reward $r_{\widetilde{Q}}$ attains the lowest value 0 at any unvisited state $s_{H-1}$. Equation (b) defines $\delta = \sum_{s_H^{\mathrm{E}} \in \mathcal{S}_H^{\mathrm{E}}} \left( P(s_H^{\mathrm{E}}|s_{H-1}, a) - P(s_H^{\mathrm{E}}|s_{H-1}, a) \right)$, which equals $\sum_{s_H \notin \mathcal{S}_H^{\mathrm{E}}} P(s_H|s_{H-1}, a)$ because probabilities sum to 1. The property of TD MDPs ensures that $\delta = \sum_{s_H^{\mathrm{E}} \in \mathcal{S}_H^{\mathrm{E}}} \left( P(s_H^{\mathrm{E}}|s_{H-1}, a) - P(s_H^{\mathrm{E}}|s_{H-1}, a) \right) \geq 0$. Besides, we have established that $\forall s_H^{\mathrm{E}} \in \mathcal{S}_H^{\mathrm{E}}, s_H \notin \mathcal{S}_H^{\mathrm{E}}, \widetilde{V}(s_H^{\mathrm{E}}) \geq \widetilde{V}(s_H)$. Together, we can prove inequality (c).

Furthermore, Lemma 1 indicates that there exists $\widehat{s_{H-1}^{\mathrm{E}}} \in \mathcal{S}_{H-1}^{\mathrm{E}}$ and $a_{H-1}^{\mathrm{E}}$ such that $r_{\widetilde{Q}}(\widehat{s_{H-1}^{\mathrm{E}}}, a_{H-1}^{\mathrm{E}}) = 1$. In this case, inequality (a) strictly holds, proving that $\widetilde{V}(\widehat{s_{H-1}^{\mathrm{E}}}) > \widetilde{V}(s_{H-1})$.

**Inductive Hypothesis:** Assume that at time step $h$, properties (1) and (2) hold.

**Inductive Step:** We need to prove the properties hold for time step $h - 1$. The proof mirrors the Base Case exactly. We first prove property (1). According to property (1) in the inductive hypothesis, we have that $\exists \widehat{s_h^{\mathrm{E}}} \in \mathcal{S}_h^{\mathrm{E}}, \forall s_h \notin \mathcal{S}_h^{\mathrm{E}}$, $\widetilde{V}(\widehat{s_h^{\mathrm{E}}}) > \widetilde{V}(s_h)$. Then we have that

$$
\begin{aligned}
&\widetilde{Q}(s_{h-1}, a^{\mathrm{E}}) - \widetilde{Q}(s_{h-1}, a) \\
&\overset{(a)}{\geq} \mathbb{E}_{s' \sim P(\cdot|s_{h-1}, a^{\mathrm{E}})}[\widetilde{V}(s')] - \mathbb{E}_{s' \sim P(\cdot|s_{h-1}, a)}[\widetilde{V}(s')] \\
&= \widetilde{V}(\widehat{s_h^{\mathrm{E}}}) \left( P(\widehat{s_h^{\mathrm{E}}}|s_{h-1}, a^{\mathrm{E}}) - P(\widehat{s_h^{\mathrm{E}}}|s_{h-1}, a) \right)
\end{aligned}
$$

$$+ \sum_{\substack{s_h^{\mathrm{E}} \in \mathcal{S}_h^{\mathrm{E}} \\ s_h^{\mathrm{E}} \neq \widehat{s_h^{\mathrm{E}}}}} \left(P(s_h^{\mathrm{E}}|s_{h-1}, a^{\mathrm{E}}) - P(s_h^{\mathrm{E}}|s_{h-1}, a)\right) \widetilde{V}(s_h^{\mathrm{E}}) - \sum_{s_h \notin \mathcal{S}_h^{\mathrm{E}}} P(s_h|s_{h-1}, a) \widetilde{V}(s_h)$$

$$\geq \widetilde{V}(\widehat{s_h^{\mathrm{E}}}) \left(P(\widehat{s_h^{\mathrm{E}}}|s_{h-1}, a^{\mathrm{E}}) - P(\widehat{s_h^{\mathrm{E}}}|s_{h-1}, a)\right)$$

$$+ \left(\min_{\substack{s_h^{\mathrm{E}} \in \mathcal{S}_h^{\mathrm{E}} \\ s_h^{\mathrm{E}} \neq \widehat{s_h^{\mathrm{E}}}}} \widetilde{V}(s_h^{\mathrm{E}})\right) \sum_{\substack{s_h^{\mathrm{E}} \in \mathcal{S}_h^{\mathrm{E}} \\ s_h^{\mathrm{E}} \neq \widehat{s_h^{\mathrm{E}}}}} \left(P(s_h^{\mathrm{E}}|s_{h-1}, a^{\mathrm{E}}) - P(s_h^{\mathrm{E}}|s_{h-1}, a)\right) - \left(\max_{s_h \notin \mathcal{S}_h^{\mathrm{E}}} \widetilde{V}(s_h)\right) \sum_{s_h \notin \mathcal{S}_h^{\mathrm{E}}} P(s_h|s_{h-1}, a)$$

$$\overset{(b)}{\geq} \widetilde{V}(\widehat{s_h^{\mathrm{E}}}) \left(P(\widehat{s_h^{\mathrm{E}}}|s_{h-1}, a^{\mathrm{E}}) - P(\widehat{s_h^{\mathrm{E}}}|s_{h-1}, a)\right)$$

$$+ \left(\max_{s_h \notin \mathcal{S}_h^{\mathrm{E}}} \widetilde{V}(s_h)\right) \sum_{\substack{s_h^{\mathrm{E}} \in \mathcal{S}_h^{\mathrm{E}} \\ s_h^{\mathrm{E}} \neq \widehat{s_h^{\mathrm{E}}}}} \left(P(s_h^{\mathrm{E}}|s_{h-1}, a^{\mathrm{E}}) - P(s_h^{\mathrm{E}}|s_{h-1}, a)\right) - \left(\max_{s_h \notin \mathcal{S}_h^{\mathrm{E}}} \widetilde{V}(s_h)\right) \sum_{s_h \notin \mathcal{S}_h^{\mathrm{E}}} P(s_h|s_{h-1}, a)$$

$$\overset{(c)}{=} \delta \left(\widetilde{V}(\widehat{s_h^{\mathrm{E}}}) - \max_{s_h \notin \mathcal{S}_h^{\mathrm{E}}} \widetilde{V}(s_h)\right)$$

$$\overset{(d)}{>} 0,$$

Inequality (a) follows that Lemma 1 indicates that the derived reward $r_{\widetilde{Q}}$ attains the lowest value at a non-expert state-action pair $(s_{h-1}, a)$. Inequality (b) follows the inductive hypothesis that $\forall s_h^{\mathrm{E}} \in \mathcal{S}_h^{\mathrm{E}}, s_h \notin \mathcal{S}_h^{\mathrm{E}}, \widetilde{V}(s_h^{\mathrm{E}}) \geq \widetilde{V}(s_h)$. In equation (c), we define $\delta = P(\widehat{s_h^{\mathrm{E}}}|s_{h-1}, a^{\mathrm{E}}) - P(\widehat{s_h^{\mathrm{E}}}|s_{h-1}, a)$ and leverage the property the sum of the transition probabilities equals 1. In the strict inequality (d), we have both $\delta = P(\widehat{s_h^{\mathrm{E}}}|s_{h-1}, a^{\mathrm{E}}) - P(\widehat{s_h^{\mathrm{E}}}|s_{h-1}, a) > 0$ due to the property of TD MDPs and $\widetilde{V}(\widehat{s_h^{\mathrm{E}}}) > \max_{s_h \notin \mathcal{S}_h^{\mathrm{E}}} \widetilde{V}(s_h)$ due to the inductive hypothesis.

Property (2) also follows identical logic. In particular, $\forall s_{h-1}^{\mathrm{E}} \in \mathcal{S}_{h-1}^{\mathrm{E}}, s_{h-1} \notin \mathcal{S}_{h-1}^{\mathrm{E}}$, we have

$$\widetilde{Q}(s_{h-1}^{\mathrm{E}}, a) - \widetilde{Q}(s_{h-1}, a)$$

$$\overset{(a)}{\geq} \mathbb{E}_{s' \sim P(\cdot|s_{h-1}^{\mathrm{E}}, a)}[\widetilde{V}(s')] - \mathbb{E}_{s' \sim P(\cdot|s_{h-1}, a)}[\widetilde{V}(s')]$$

$$= \sum_{s_h^{\mathrm{E}} \in \mathcal{S}_h^{\mathrm{E}}} P(s_h^{\mathrm{E}}|s_{h-1}^{\mathrm{E}}, a) \widetilde{V}(s_h^{\mathrm{E}}) - \sum_{s_h^{\mathrm{E}} \in \mathcal{S}_h^{\mathrm{E}}} P(s_h^{\mathrm{E}}|s_{h-1}, a) \widetilde{V}(s_h^{\mathrm{E}}) - \sum_{s_h \notin \mathcal{S}_h^{\mathrm{E}}} P(s_h|s_{h-1}, a) \widetilde{V}(s_h)$$

$$\overset{(b)}{\geq} \left(\min_{s_h^{\mathrm{E}} \in \mathcal{S}_h^{\mathrm{E}}} \widetilde{V}(s_h^{\mathrm{E}})\right) \sum_{s_h^{\mathrm{E}} \in \mathcal{S}_h^{\mathrm{E}}} \left(P(s_h^{\mathrm{E}}|s_{h-1}^{\mathrm{E}}, a) - P(s_h^{\mathrm{E}}|s_{h-1}, a)\right) - \left(\max_{s_h \notin \mathcal{S}_h^{\mathrm{E}}} \widetilde{V}(s_h)\right) \sum_{s_h \notin \mathcal{S}_h^{\mathrm{E}}} P(s_h|s_{h-1}, a)$$

$$\overset{(c)}{=} \delta \left(\min_{s_h^{\mathrm{E}} \in \mathcal{S}_h^{\mathrm{E}}} \widetilde{V}(s_h^{\mathrm{E}}) - \max_{s_h \notin \mathcal{S}_h^{\mathrm{E}}} \widetilde{V}(s_h)\right)$$

$$\overset{(d)}{\geq} 0.$$

Inequality (a) holds because Lemma 1 shows that the induced reward $\widetilde{r}$ attains the lowest value 0 at any non-expert state-action pair $(s_{h-1}, a)$. Inequality (b) follows the property of TD MDPs. In equation (c), we define that $\delta = \sum_{s_h^{\mathrm{E}} \in \mathcal{S}_h^{\mathrm{E}}} P(s_h^{\mathrm{E}}|s_{h-1}^{\mathrm{E}}, a) - P(s_h^{\mathrm{E}}|s_{h-1}, a)$. In inequality (d), we have both $\delta > 0$ due to the property of TD MDPs and $\min_{s_h^{\mathrm{E}} \in \mathcal{S}_h^{\mathrm{E}}} \widetilde{V}(s_h^{\mathrm{E}}) - \max_{s_h \notin \mathcal{S}_h^{\mathrm{E}}} \widetilde{V}(s_h) \geq 0$ due to the inductive hypothesis. Then we can prove that $\forall s_{h-1}^{\mathrm{E}} \in \mathcal{S}_{h-1}^{\mathrm{E}}, s_{h-1} \notin \mathcal{S}_{h-1}^{\mathrm{E}}, \widetilde{V}(s_{h-1}^{\mathrm{E}}) \geq \widetilde{V}(s_{h-1})$.

Again, Lemma 1 shows that $\exists (\widehat{s_{h-1}^{\mathrm{E}}}, a_{h-1}^{\mathrm{E}}) \in \mathcal{D}^{\mathrm{E}}, r_{\widetilde{Q}}(s_{h-1}^{\mathrm{E}}, a_{h-1}^{\mathrm{E}}) = 1$. In this case, inequality (a) strictly holds. Then we can get that $\exists \widehat{s_{h-1}^{\mathrm{E}}} \in \mathcal{S}_{h-1}^{\mathrm{E}}, \forall s_{h-1} \notin \mathcal{S}_{h-1}^{\mathrm{E}}, \widetilde{V}(\widehat{s_{h-1}^{\mathrm{E}}}) > \widetilde{V}(s_{h-1})$. Up to now, we have finished the induction analysis for the Q-function recovered by Dual Q-DM.

We continue to analyze the Q-function recovered by IQ-Learn. Recall the objective of IQ-Learn:

$$\max_{Q \in \mathcal{Q}} \mathcal{L}(Q) := \mathbb{E}_{\tau \sim \mathcal{D}^E} \left[ \sum_{h=1}^{H} Q(s_h, a_h) - (\text{LSE}\, Q)(s_{h+1}) \right] - \mathbb{E}_{s_1 \sim \rho} \left[ (\text{LSE}\, Q)(s_1) \right].$$

Here $\mathcal{Q} = \{Q : \mathcal{S} \times \mathcal{A} \to [0, C]\}$ denotes the class of bounded Q-functions. In IQ-Learn, there are no other constraints beyond the boundedness constraint. Therefore, we analyze its solution by analyzing the gradient.

In time steps $2 \le h \le H - 1$, for an unvisited state-action pair $(s_h, a_h) \notin \mathcal{D}^E$, we have that

$$\frac{\mathcal{L}(Q)}{\partial Q(s_h, a_h)} = 0.$$

As such, the IQ-Learn's Q-function remains at its initial value, which we assume to be a constant. That is, $\widehat{Q}(s_h, a) = C, \forall a \in \mathcal{A}$. At time step $h = 1$, for all $s_1 \notin \mathcal{D}^E, a \in \mathcal{A}$,

$$\frac{\mathcal{L}(Q)}{\partial Q(s_1, a)} = -\rho(s_1) \frac{\exp(Q(s_1, a))}{\sum_a \exp(Q(s_1, a))} < 0.$$

Consequently, all $Q(s_1, a)$ will be minimized uniformly, leading to the solution $\widehat{Q}(s_1, a) = 0, \forall a \in \mathcal{A}$. We finish the proof. $\square$

## C.4. An Example for Illustrating the Importance of Bellman Constraints

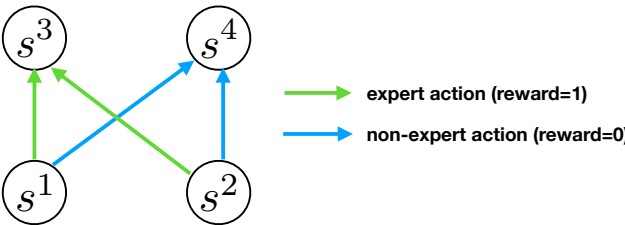

*Figure 5.* An Example of TD MDP. Here arrows denote the corresponding transitions.

Consider the TD MDP shown in Figure 5, where $\mathcal{S} = \{s^1, s^2, s^3, s^4\}$, $\mathcal{A} = \{a^1, a^2\}$ with $a^1$ being the expert action and $H = 2$. We consider a uniform initial state distribution over $\{s^1, s^2\}$ and use an entropy coefficient of $\alpha = 0.1$. The demonstration dataset consists of a single expert trajectory $\mathcal{D}^E = \{s^1, a^1, s^3, a^1\}$, rendering $s^2$ an unvisited state. We study generalization by characterizing the learned Q-function at this unvisited state.

In this example, Dual Q-DM and IQ-Learn share the same objective

$$\mathcal{L}(Q) = Q(s^1, a^1) - \text{LSE}(Q)(s^3) + Q(s^3, a^1) - \tfrac{1}{2} \text{LSE}(Q)(s^1) - \tfrac{1}{2} \text{LSE}(Q)(s^2),$$

but they yield markedly different Q-functions at $s^2$. IQ-Learn minimizes this objective without enforcing Bellman constraints. Moreover, $\mathcal{L}(Q)$ is monotonically decreasing in both $Q(s^2, a^1)$ and $Q(s^2, a^2)$. Consequently, IQ-Learn drives both values to the same lower bound, resulting in $\widehat{Q}(s^2, a^1) = \widehat{Q}(s^2, a^2) = 0$, and thus fails to distinguish the expert action at the unvisited state.

In contrast, Dual Q-DM minimizes the same objective subject to Bellman constraints. By Lemma 1, these constraints attain the maximal value on visited state-action pairs and the minimal value on unvisited ones, yielding

$$r_{\widetilde{Q}}(s^1, a^1) = 1, \quad r_{\widetilde{Q}}(s^1, a^2) = r_{\widetilde{Q}}(s^2, a^1) = r_{\widetilde{Q}}(s^2, a^2) = 0,$$
$$\widetilde{Q}(s^3, a^1) = 1, \quad \widetilde{Q}(s^3, a^2) = \widetilde{Q}(s^4, a^1) = \widetilde{Q}(s^4, a^2) = 0.$$

The resulting Q-function satisfies the soft Bellman optimality equation with respect to this reward. Consequently, the Q-values at $s^2$ are given by

$$\widetilde{Q}(s^2, a^1) = r_{\widetilde{Q}}(s^2, a^1) + \text{LSE}(\widetilde{Q}(s^3)) = \log(e^{10} + 1),$$

$$\widetilde{Q}(s^2, a^2) = r_{\widetilde{Q}}(s^2, a^2) + \mathrm{LSE}(\widetilde{Q}(s^4)) = \log(2).$$

Although $r_{\widetilde{Q}}(s^2, a^1)$ and $r_{\widetilde{Q}}(s^2, a^2)$ are both zero, the Bellman constraints propagate values from successor states. As a result, $\widetilde{Q}(s^2, a^1)$ reflects the higher value of the visited successor state $s^3$, whereas $\widetilde{Q}(s^2, a^2)$ reflects the lower value of the unvisited successor state $s^4$. This enables Dual Q-DM to correctly identify the expert action at the unvisited state $s^2$.

### C.5. Practical Implementation of Dual Q-DM

In this part, we present the practical implementation of Dual Q-DM, which incorporates Bellman constraints via penalization. Algorithm 1 outlines the complete procedure.

$$\ell(Q) = \mathbb{E}_{\tau \sim \mathcal{D}^{\mathrm{E}}} \left[ \sum_{h=1}^{H} Q(s_h, a_h) - \mathrm{LSE}(Q)(s_{h+1}) \right] - \mathbb{E}_{s_1 \sim \rho} \left[ \mathrm{LSE}(Q)(s_1) \right]$$

$$- \beta \mathbb{E}_{\tau \sim \mathcal{D}^{\mathrm{online}}} \left[ \sum_{h=1}^{H} \left( \mathrm{ReLU}(0 - Q(s_h, a_h) + \mathrm{LSE}(\overline{Q})(s_{h+1})) \right)^2 + \left( \mathrm{ReLU}(Q(s_h, a_h) - \mathrm{LSE}(\overline{Q})(s_{h+1}) - 1) \right)^2 \right]$$

$$(26)$$

where $\mathrm{ReLU}(x) = \max\{0, x\}$, $\mathcal{D}^{\mathrm{online}}$ is the online replay buffer and $\beta > 0$ controls the strengths of the Bellman constraints. Besides, to improve the training stability, we use the target network $\overline{Q}$ to calculate the Bellman backup. This penalty term imposes a quadratic penalty whenever the learned Q-function violates the Bellman constraints, encouraging the solution to remain feasible. More importantly, it is evaluated over the *entire online dataset*, ensuring that Bellman constraints hold across a broad state-action space beyond the demonstrations, which is essential for generalization as suggested by our theoretical analysis.

---

**Algorithm 1** Practical Implementation of Dual Q-DM for Discrete Control

---

**Input:** Q-value $Q^0$, target Q-value $\overline{Q}^0 = Q^0$, and dataset $\mathcal{D}^{\mathrm{online}} = \emptyset$.
1: **for** $k = 1, 2, \ldots, K$ **do**
2:      Apply $\pi_{Q^{k-1}}$ to roll out a trajectory $\tau^{k-1}$ and append it to the online replay buffer $\mathcal{D}^{\mathrm{online}} = \mathcal{D}^{\mathrm{online}} \cup \{\tau^{k-1}\}$.
3:      Update the Q-value function by $Q^k \leftarrow Q^{k-1} + \alpha_Q \nabla \ell(Q)$ from Eq. (26).
4:      Update the target Q-value by $\overline{Q}^k \leftarrow \tau Q^k + (1 - \tau) \overline{Q}^{k-1}$.
5: **end for**

---

For tasks with continuous action spaces, it is difficult to directly calculate the log-partition function $(\mathrm{LSE}\, Q)(s) = \log(\sum_{a \in \mathcal{A}} \exp(Q(s, a)))$ and derive the softmax policy $\pi_Q = \mathrm{softmax}(Q)$. To circumvent this issue, we follow (Haarnoja et al., 2018; Garg et al., 2021) and trains an additional policy model $\pi$ via solving

$$\max_{\pi} \ell(\pi) := \mathbb{E}_{s \sim \mathcal{D}^{\mathrm{online}}, a \sim \pi(\cdot|s)} \left[ Q(s, a) - \log(\pi(a|s)) \right].$$

$$(27)$$

It is direct to prove that the solution to the above optimization problem is exactly $\mathrm{softmax}(Q)$. As a result, we can approximate the softmax policy by maximizing $\ell(Q)$. With a learned policy $\pi$, we can further approximate the log-partition function $(\mathrm{LSE}\, Q)(s)$ by $\mathbb{E}_{a \sim \pi(\cdot|s)} \left[ Q(s, a) - \log(\pi(a|s)) \right]$. Then the objective for the Q-function becomes

$$\ell(Q) = \mathbb{E}_{\tau \sim \mathcal{D}^{\mathrm{E}}} \left[ \sum_{h=1}^{H} Q(s_h, a_h) - \mathbb{E}_{a' \sim \pi(\cdot|s_{h+1})} \left[ Q(s_{h+1}, a') - \log(\pi(a'|s_{h+1})) \right] \right]$$

$$- \mathbb{E}_{s_1 \sim \rho, a' \sim \pi(\cdot|s_1)} \left[ Q(s_1, a') - \log(\pi(a'|s_1)) \right]$$

$$- \beta \mathbb{E}_{\tau \sim \mathcal{D}^{\mathrm{online}}} \left[ \sum_{h=1}^{H} \left( \mathrm{ReLU}(0 - Q(s_h, a_h) + \mathbb{E}_{a' \sim \pi(\cdot|s_{h+1})} \left[ \overline{Q}(s_{h+1}, a') - \log(\pi(a'|s_{h+1})) \right]) \right)^2 \right.$$

$$\left. + \left( \mathrm{ReLU}(Q(s_h, a_h) - \mathbb{E}_{a' \sim \pi(\cdot|s_{h+1})} \left[ \overline{Q}(s_{h+1}, a') - \log(\pi(a'|s_{h+1})) \right] - 1) \right)^2 \right]$$

$$(28)$$

---

**Algorithm 2** Practical Implementation of Dual Q-DM for Continuous Control

---

**Input:** Q-value $Q^0$, target Q-value $\overline{Q}^0 = Q^0$, policy $\pi^0$ and dataset $\mathcal{D}^0 = \emptyset$.
1: **for** $k = 1, 2, \ldots, K$ **do**
2:     Apply $\pi^{k-1}$ to roll out a trajectory $\tau^{k-1}$ and append it to the dataset $\mathcal{D}^k = \mathcal{D}^{k-1} \cup \{\tau^{k-1}\}$.
3:     Update the Q-value function by $Q^k \leftarrow Q^{k-1} + \alpha_Q \nabla \ell(Q)$ from Eq. (28).
4:     Update the policy by $\pi^k \leftarrow \pi^{k-1} + \alpha_\pi \nabla \ell(\pi)$ from Eq. (27).
5:     Update the target Q-value by $\overline{Q}^k \leftarrow \tau Q^k + (1 - \tau)\overline{Q}^{k-1}$.
6: **end for**

---

### C.6. A Detailed Comparison of Various Q-based IL Approaches

We perform a thorough comparison of Dual Q-DM with prior Q-based methods. The practical objective of Dual Q-DM is

$$\text{Dual Q-DM: } \max_Q \mathbb{E}_{\tau \sim \mathcal{D}^{\mathrm{E}}} \left[ \sum_{h=1}^{H} Q(s_h, a_h) - \text{LSE}(Q)(s_{h+1}) \right] - \mathbb{E}_{s_1 \sim \rho} \left[ \text{LSE}(Q)(s_1) \right]$$

$$- \beta \mathbb{E}_{\tau \sim \mathcal{D}^{\text{online}}} \left[ \sum_{h=1}^{H} \left( \text{ReLU}(0 - Q(s_h, a_h) + \text{LSE}(Q)(s_{h+1})) \right)^2 + \left( \text{ReLU}(Q(s_h, a_h) - \text{LSE}(Q)(s_{h+1}) - 1) \right)^2 \right].$$

For comparison, we consider several IQ-Learn variants. The standard IQ-Learn with TV divergence optimizes:

$$\text{IQ-Learn (TV): } \max_{Q \in \mathcal{Q}} \mathbb{E}_{\tau \sim \mathcal{D}^{\mathrm{E}}} \left[ \sum_{h=1}^{H} Q(s_h, a_h) - (\text{LSE}\, Q)(s_{h+1}) \right] - \mathbb{E}_{s_1 \sim \rho} \left[ (\text{LSE}\, Q)(s_1) \right].$$

The standard IQ-Learn with $\chi^2$-divergence adds a squared Bellman error term calculated on *demonstrations*.

$$\text{IQ-Learn } (\chi^2): \max_{Q \in \mathcal{Q}} \mathbb{E}_{\tau \sim \mathcal{D}^{\mathrm{E}}} \left[ \sum_{h=1}^{H} Q(s_h, a_h) - (\text{LSE}\, Q)(s_{h+1}) \right] - \mathbb{E}_{s_1 \sim \rho} \left[ (\text{LSE}\, Q)(s_1) \right]$$

$$- \frac{1}{4} \mathbb{E}_{\tau \sim \mathcal{D}^{\mathrm{E}}} \left[ \sum_{h=1}^{H} (Q(s_h, a_h) - (\text{LSE}\, Q)(s_{h+1}))^2 \right].$$

Garg et al. (2021) included a regularization term in its official implementation but did not introduce it in the paper. We call this variant Reg IQ-Learn, which is formulated as

$$\text{Reg IQ-Learn: } \max_{Q \in \mathcal{Q}} \mathbb{E}_{\tau \sim \mathcal{D}^{\mathrm{E}}} \left[ \sum_{h=1}^{H} Q(s_h, a_h) - (\text{LSE}\, Q)(s_{h+1}) \right] - \mathbb{E}_{s_1 \sim \rho} \left[ (\text{LSE}\, Q)(s_1) \right]$$

$$- \frac{1}{4} \mathbb{E}_{\tau \sim \mathcal{D}^{\text{online}}} \left[ \sum_{h=1}^{H} (Q(s_h, a_h) - (\text{LSE}\, Q)(s_{h+1}))^2 \right].$$

The additional regularization corresponds to the squared Bellman error evaluated on the *online dataset*. LS-IQ (Al-Hafez et al., 2023) mainly follows the Reg IQ-Learn formulation and is thus omitted from this comparison. We present the key distinctions as follows.

**Dual Q-DM V.S. IQ-Learn $(\chi^2)$ and IQ-Learn (TV).** First, compared to IQ-Learn (TV), Dual Q-DM incorporates Bellman constraints to establish temporal coupling across the Q-function. Section 4 demonstrates that Bellman constraints, by propagating Q-value information from visited to unvisited states, are essential for Q-based IL to generalize beyond demonstrations and mitigate compounding errors. Empirical results in Figure 4 and Figure 2 corroborate this theoretical finding.

Second, compared to IQ-Learn (TV), IQ-Learn $(\chi^2)$ adds a Bellman error term computed on expert data. While this superficially resembles our Bellman constraints by introducing coupling in the Q-function, its impact is limited because it only couples Q-values on state-action pairs visited in demonstrations. Since Theorem 1 establishes that both IQ-Learn (TV)

and BC can already recover expert actions in demonstrations, this additional constraint provides minimal benefit. In contrast, Dual Q-DM induces Q-function coupling across the broader online dataset, extending well beyond the demonstrations and thereby enabling generalization to unvisited states. Figure 4 empirically confirms that Dual Q-DM demonstrates substantially better performance than IQ-Learn ($\chi^2$).

**Reg IQ-Learn V.S. IQ-Learn ($\chi^2$).** Unlike IQ-Learn ($\chi^2$), Reg IQ-Learn computes the Bellman error term on the broader online data. From our theoretical perspective, this enables coupling across state-action pairs beyond the expert demonstrations, facilitating generalization to unvisited states. Figure 2 confirms that Reg IQ-Learn substantially outperforms standard IQ-Learn. While previous works (Al-Hafez et al., 2023; Karimi & Ebadzadeh, 2025) have interpreted this regularization as implicit reward regularization for training stability, we uncover a fundamental RL mechanism: the regularization propagates Q-value information through environment dynamics from visited to unvisited states, enabling effective generalization.

**Dual Q-DM V.S. Reg IQ-Learn.** Although Reg IQ-Learn's regularization establishes Q-function coupling, it introduces reward bias: the Bellman error term explicitly assigns a minimum reward of 0 to all online state-action pairs, even those closely resembling expert behavior. This biases Q-function learning. In contrast, Dual Q-DM's penalty term constrains the reward range without prescribing specific values, avoiding this bias. The experimental results in Figure 2 corroborate this advantage: Dual Q-DM outperforms Reg IQ-Learn.

# D. Extension to the Infinite-horizon Setting

In this part, we extend the main results of this paper to the infinite-horizon setting.

## D.1. Infinite-horizon MDP Setting

We consider the infinite-horizon discounted MDP $\mathcal{M} = (\mathcal{S}, \mathcal{A}, P, r^\star, \gamma, \rho)$, where $\gamma \in (0, 1)$ is the discount factor and all other notation follows Section 4. Unlike the finite-horizon setting, the state space $\mathcal{S}$ is stationary (not layered). A policy $\pi : \mathcal{S} \to \Delta(\mathcal{A})$ induces a discounted stationary occupancy measure:

$$d^\pi(s, a) := (1 - \gamma) \sum_{t=1}^{\infty} \gamma^{t-1} \mathbb{P}^\pi(s_t = s, a_t = a).$$

**BC in the infinite-horizon setting.** BC in the infinite-horizon setting directly applies the same MLE objective as in the finite-horizon case (Eq. (1)):

$$\pi^{\mathrm{BC}} = \underset{\pi \in \Pi}{\operatorname{argmax}} \ \mathbb{E}_{(s,a) \sim \mathcal{D}^{\mathrm{E}}} \left[ \log \pi(a|s) \right].$$

In function space, the BC policy recovers the empirical expert conditional distribution on visited states and assigns a uniform distribution on states uncovered by $\mathcal{D}^{\mathrm{E}}$. As a consequence, BC cannot infer expert actions at unvisited states and suffers from compounding errors. Its imitation gap scales as $\mathcal{O}\left(1/(1-\gamma)^2\right)$ (Xu et al., 2020), directly analogous to the $\mathcal{O}(H^2)$ bound in the finite-horizon case (Ross & Bagnell, 2010; Rajaraman et al., 2020), with the effective horizon $1/(1-\gamma)$ playing the role of $H$.

**AIL in the Infinite-horizon Setting.** AIL in the infinite-horizon setting minimizes the discrepancy between the learner's stationary occupancy measure and the expert's, analogous to the finite-horizon formulation in Eq. (3):

$$\min_{\pi \in \Pi} \ D_{\mathrm{TV}}\left(\widehat{d^{\pi^{\mathrm{E}}}}, d^\pi\right) - \overline{\mathcal{H}}(d^\pi), \tag{29}$$

where $\widehat{d^{\pi^{\mathrm{E}}}}(s, a)$ is the empirical estimation of the expert's state-action distribution based on finite demonstrations, and $\overline{\mathcal{H}}(d^\pi) = \mathbb{E}_{(s,a) \sim d^\pi}[-\log(\pi(a|s))]$ is the entropy regularization term. The equivalent minimax formulation is:

$$\min_{\pi \in \Pi} \max_{r \in \mathcal{R}} \ \mathbb{E}_{(s,a) \sim \mathcal{D}^{\mathrm{E}}} \left[ r(s, a) \right] - \mathbb{E}_{(s,a) \sim d^\pi} \left[ r(s, a) - \log \pi(a|s) \right], \tag{30}$$

where $\mathcal{R} = \{r : \mathcal{S} \times \mathcal{A} \to [0, 1]\}$ is the class of bounded reward functions. This is the direct analogue of the finite-horizon AIL minimax objective in Eq. (2), with the per-step summation replaced by the stationary occupancy measure expectation.

Through global occupancy measure matching, AIL achieves an imitation gap of $\mathcal{O}(1/(1-\gamma))$ (Xu et al., 2020), i.e., linear in the effective horizon, thereby mitigating compounding errors.

**IQ-Learn in the Infinite-horizon Setting.** Following Eq. (9) of Garg et al. (2021), IQ-Learn in the infinite-horizon setting optimizes the following objective over Q-functions:

$$\max_{Q \in \mathcal{Q}} \quad \mathbb{E}_{(s,a,s') \sim \mathcal{D}^{\mathrm{E}}} \left[ Q(s,a) - \gamma \, \mathrm{LSE}(Q)(s') \right] - (1 - \gamma) \, \mathbb{E}_{s_1 \sim \rho} \left[ \mathrm{LSE}(Q)(s_1) \right]. \tag{31}$$

After obtaining $\widehat{Q}$ by solving Eq. (31), IQ-Learn derives the corresponding softmax policy $\pi_{\widehat{Q}}(a|s) \propto \exp(\widehat{Q}(s,a))$.

**Dual Q-DM in the Infinite-horizon Setting.** The natural extension of Dual Q-DM to the infinite-horizon setting shares a similar objective to IQ-Learn in Eq. (31) but additionally imposes Bellman constraints:

$$\max_{Q} \quad \mathbb{E}_{(s,a,s') \sim \mathcal{D}^{\mathrm{E}}} \left[ Q(s,a) - \gamma \mathbb{E}_{s' \sim P(\cdot|s,a)} \left[ \mathrm{LSE}(Q)(s') \right] \right] - (1 - \gamma) \, \mathbb{E}_{s_1 \sim \rho} \left[ \mathrm{LSE}(Q)(s_1) \right],$$

$$\text{s.t.} \quad 0 \le Q(s,a) - \gamma \, \mathbb{E}_{s' \sim P(\cdot|s,a)} \left[ \mathrm{LSE}(Q)(s') \right] \le 1, \quad \forall (s,a) \in \mathcal{S} \times \mathcal{A}. \tag{32}$$

Here the Bellman constraints bound the implicit reward $r_Q(s,a) := Q(s,a) - \gamma \, \mathbb{E}_{s' \sim P(\cdot|s,a)}[\mathrm{LSE}(Q)(s')]$ to the interval $[0,1]$, consistent with the finite-horizon formulation.

### D.2. Extension of Theorem 1.

We now show that the reduction of IQ-Learn to BC established in Theorem 1 extends to the infinite-horizon setting.

**Theorem 3** (Extension of Theorem 1 to Infinite-horizon MDPs). *Consider infinite-horizon stationary MDPs. Suppose that $\widehat{Q}$ is the optimal solution to IQ-Learn in Eq. (31) and $\pi_{\widehat{Q}}$ is the derived policy. Assume the softmax policy class realizes the BC policy, i.e., $\pi^{\mathrm{BC}} \in \{\pi_Q : Q \in \mathcal{Q}\}$. Then the following holds:*

- *For any non-initial state $s$ with $\rho(s) = 0$, $\pi_{\widehat{Q}}(\cdot|s) = \pi^{\mathrm{BC}}(\cdot|s)$.*

- *For any initial state uncovered by demonstrations, i.e., $\rho(s_1) > 0$ and $s_1 \notin \mathcal{D}^{\mathrm{E}}$, $\pi_{\widehat{Q}}(\cdot|s_1) = \pi^{\mathrm{BC}}(\cdot|s_1) = \mathrm{Unif}(\mathcal{A})$.*

*Proof.* The proof applies the same re-parameterization technique as Theorem 1. We apply a telescoping transformation to rewrite the IQ-Learn objective. Using the relation $d^{\pi^{\mathrm{E}}}(s,a) = (1-\gamma) \sum_{t=1}^{\infty} \gamma^{t-1} \mathbb{P}^{\pi^{\mathrm{E}}}(s_t = s, a_t = a)$, we have:

$$\mathbb{E}_{(s,a,s') \sim \mathcal{D}^{\mathrm{E}}} \left[ Q(s,a) - \gamma \, \mathrm{LSE}(Q)(s') \right] - (1-\gamma) \, \mathbb{E}_{s_1 \sim \rho} \left[ \mathrm{LSE}(Q)(s_1) \right]$$

$$= (1-\gamma) \, \mathbb{E}_{\tau \sim \mathcal{D}^{\mathrm{E}}} \left[ \sum_{t=1}^{\infty} \gamma^{t-1} \left( Q(s_t, a_t) - \gamma \, \mathrm{LSE}(Q)(s_{t+1}) \right) \right] - (1-\gamma) \, \mathbb{E}_{s_1 \sim \rho} \left[ \mathrm{LSE}(Q)(s_1) \right]$$

$$\overset{(a)}{=} (1-\gamma) \, \mathbb{E}_{\tau \sim \mathcal{D}^{\mathrm{E}}} \left[ \sum_{t=1}^{\infty} \gamma^{t-1} \underbrace{\left( Q(s_t, a_t) - \mathrm{LSE}(Q)(s_t) \right)}_{\log \pi_Q(a_t|s_t)} \right]$$

$$+ (1-\gamma) \left( \mathbb{E}_{s_1 \sim \mathcal{D}^{\mathrm{E}}} \left[ \mathrm{LSE}(Q)(s_1) \right] - \mathbb{E}_{s_1 \sim \rho} \left[ \mathrm{LSE}(Q)(s_1) \right] \right)$$

$$= \underbrace{\mathbb{E}_{(s,a) \sim \widehat{d^{\pi^{\mathrm{E}}}}} \left[ \log \pi_Q(a|s) \right]}_{\text{BC objective}} + (1-\gamma) \left( \mathbb{E}_{s_1 \sim \mathcal{D}^{\mathrm{E}}} \left[ \mathrm{LSE}(Q)(s_1) \right] - \mathbb{E}_{s_1 \sim \rho} \left[ \mathrm{LSE}(Q)(s_1) \right] \right),$$

where step (a) follows the telescoping argument: for any trajectory $\tau$,

$$\sum_{t=1}^{\infty} \gamma^{t-1} \left( Q(s_t, a_t) - \gamma \, \mathrm{LSE}(Q)(s_{t+1}) \right) = \sum_{t=1}^{\infty} \gamma^{t-1} \left( Q(s_t, a_t) - \mathrm{LSE}(Q)(s_t) \right) + \mathrm{LSE}(Q)(s_1),$$

using $\sum_{t=1}^{\infty} \gamma^{t-1} \mathrm{LSE}(Q)(s_t) - \sum_{t=1}^{\infty} \gamma^{t} \mathrm{LSE}(Q)(s_{t+1}) = \mathrm{LSE}(Q)(s_1)$ and the fact that the boundary term $\gamma^{t} \mathrm{LSE}(Q)(s_{t+1}) \to 0$ as $t \to \infty$ (since $Q$ is bounded and $\gamma < 1$). Let $\mathcal{L}(Q)$ denote the full objective. We now analyze the optimal $Q$ at two types of states.

**Non-initial states.** For any state $s$ with $\rho(s) = 0$, the second term $(1 - \gamma) \left( \mathbb{E}_{s_1 \sim \mathcal{D}^{\mathrm{E}}}[\mathrm{LSE}(Q)(s_1)] - \mathbb{E}_{s_1 \sim \rho}[\mathrm{LSE}(Q)(s_1)] \right)$ does not depend on $Q(s, \cdot)$. Therefore:

$$\underset{Q(s,\cdot)}{\mathrm{argmax}}\, \mathcal{L}(Q) = \underset{Q(s,\cdot)}{\mathrm{argmax}}\, \mathbb{E}_{(s,a) \sim \widehat{d^{\pi^{\mathrm{E}}}}} \left[ \log \pi_Q(a|s) \right],$$

which is precisely the BC objective. Hence $\pi_{\widehat{Q}}(\cdot|s) = \pi^{\mathrm{BC}}(\cdot|s)$ for all such states.

**Unvisited initial states.** For any state $s$ with $\rho(s_1) > 0$, $s_1 \notin \mathcal{D}^{\mathrm{E}}$, so the BC term contributes no signal. Optimizing over $Q(s_1, \cdot)$ reduces to:

$$\underset{Q(s_1,\cdot)}{\mathrm{argmax}}\, \mathcal{L}(Q) = \underset{Q(s_1,\cdot)}{\mathrm{argmin}}\, (1 - \gamma)\, \mathbb{E}_{s_1 \sim \rho} \left[ \mathrm{LSE}(Q)(s_1) \right].$$

Since $\mathrm{LSE}(Q)(s_1) = \log \sum_{a \in \mathcal{A}} \exp(Q(s_1, a))$ is monotonically increasing in $Q(s_1, a)$ for every action $a$, the minimizer uniformly suppresses all action values, yielding $\pi_{\widehat{Q}}(a|s_1) = 1/|\mathcal{A}|$ for all $a \in \mathcal{A}$. This coincides with $\pi^{\mathrm{BC}}(\cdot|s_1)$ on unvisited initial states. $\qquad\square$

### D.3. Extension of Theorem 2

We now extend Theorem 2 to the infinite-horizon setting, showing that the infinite-horizon Dual Q-DM in Eq. (32) recovers the optimal solution to the infinite-horizon AIL objective in Eq. (29).

**Theorem 4** (Extension of Theorem 2 to Infinite-horizon MDPs)**.** *Consider infinite-horizon stationary MDPs. Suppose that $\widetilde{Q}$ is the optimal solution to Dual Q-DM in Eq. (32) and $\pi_{\widetilde{Q}} = \mathrm{softmax}(\widetilde{Q})$ is the derived policy. Then $\pi_{\widetilde{Q}}$ is the optimal solution to AIL in Eq. (29):*

$$\pi_{\widetilde{Q}} \in \underset{\pi \in \Pi}{\mathrm{argmin}}\, D_{\mathrm{TV}} \left( \widehat{d^{\pi^{\mathrm{E}}}}, d^{\pi} \right) - \overline{\mathcal{H}}(d^{\pi}).$$

The proof parallels the finite-horizon proof in Section C.2, extending the primal-dual framework to the infinite-horizon setting. We establish the equivalence through three formulations.

**Step 1: Primal Distribution Matching (Infinite-horizon).** We reformulate AIL in Eq. (29) as a constrained optimization over the stationary occupancy measure $d$:

$$
\begin{aligned}
\min_{d \geq 0}\ & D_{\mathrm{TV}} \left( \widehat{d^{\pi^{\mathrm{E}}}}, d \right) - \overline{\mathcal{H}}(d), \\
\mathrm{s.t.}\ & \sum_{a \in \mathcal{A}} d(s, a) = (1 - \gamma)\rho(s) + \gamma \sum_{(s', a') \in \mathcal{S} \times \mathcal{A}} d(s', a') P(s|s', a'), \quad \forall s \in \mathcal{S}.
\end{aligned}
\tag{33}
$$

The constraint is the discounted Bellman flow equation for occupancy measures, which characterizes all valid stationary occupancy measures induced by some policy $\pi$ (Puterman, 2014). This is the exact infinite-horizon analogue of the primal distribution matching in Eq. (6).

**Step 2: Dual V-Function Distribution Matching (Infinite-horizon).** We apply Lagrangian duality to Eq. (33), introducing a multiplier $V(s)$ for each flow constraint. The dual problem is:

$$
\begin{aligned}
\max_{r, V}\ & \mathbb{E}_{(s,a) \sim \mathcal{D}^{\mathrm{E}}} \left[ r(s, a) \right] - (1 - \gamma) \mathbb{E}_{s \sim \rho} \left[ V(s) \right], \\
\mathrm{s.t.}\ & V(s) = \mathrm{LSE}\left( r + \gamma P V \right)(s), \quad \forall s \in \mathcal{S}, \\
& 0 \leq r(s, a) \leq 1, \quad \forall (s, a) \in \mathcal{S} \times \mathcal{A},
\end{aligned}
\tag{34}
$$

where $(r + \gamma P V)(s, a) := r(s, a) + \gamma \mathbb{E}_{s' \sim P(\cdot|s,a)}[V(s')]$ denotes the discounted Bellman backup operator. The first constraint requires $V$ to satisfy the soft Bellman optimality equation with respect to $r$, i.e., $V$ is the soft optimal value function under $r$. Strong duality holds since Eq. (33) is a convex program in $d$ with feasible solutions.

**Step 3: Change of Variables to Dual Q-DM (Infinite-horizon).** Define the Q-function $Q(s, a) := r(s, a) + \gamma \mathbb{E}_{s' \sim P(\cdot|s,a)}[V(s')]$. Substituting into Eq. (34) and using $V(s) = \mathrm{LSE}(Q)(s)$ yields:

$$\mathbb{E}_{(s,a) \sim \mathcal{D}^{\mathrm{E}}} \left[ r(s, a) \right] - (1 - \gamma) \mathbb{E}_{s \sim \rho} \left[ V(s) \right]$$

$$= \mathbb{E}_{(s,a)\sim\mathcal{D}^{\mathrm{E}}} \left[ Q(s,a) - \gamma\mathbb{E}_{s'\sim P(\cdot|s,a)} \left[\mathrm{LSE}(Q)(s')\right]\right] - (1-\gamma)\,\mathbb{E}_{s\sim\rho}\left[\mathrm{LSE}(Q)(s)\right],$$

and the constraint $0 \leq r(s,a) \leq 1$ becomes $0 \leq Q(s,a) - \gamma\,\mathbb{E}_{s'\sim P(\cdot|s,a)}[\mathrm{LSE}(Q)(s')] \leq 1$.

We first establish the equivalence between primal distribution matching and dual V-Function distribution matching.

**Proposition 4** (Extension of Proposition 2 to Infinite-horizon MDPs). *Consider infinite-horizon stationary MDPs. Suppose that $(\widehat{r}, \widehat{V})$ is the optimal solution to Dual V-DM in Eq. (34) and $\widehat{\pi}$ is the derived policy, i.e.,*

$$\forall s \in \mathcal{S}, a \in \mathcal{A}, \quad \widehat{\pi}(a|s) \propto \exp\left(\widehat{r}(s,a) + \gamma\,\mathbb{E}_{s'\sim P(\cdot|s,a)}\left[\widehat{V}(s')\right]\right).$$

*Then $\widehat{\pi}$ is the optimal solution to AIL in Eq. (30).*

*Proof.* Let $(\widehat{r}, \widehat{V})$ be the optimal solution to Dual V-DM in Eq. (34) and let $\mathcal{R} = \{r : r(s,a) \in [0,1], \forall(s,a) \in \mathcal{S} \times \mathcal{A}\}$.

The constraint $V(s) = \mathrm{LSE}(r + \gamma PV)(s)$ for all $s \in \mathcal{S}$ in Eq. (34) requires $V$ to be exactly the soft optimal value function with respect to the reward $r$ in the infinite-horizon discounted setting, i.e., $V = V^{\star,\mathrm{soft},r}$. A standard result in discounted entropy-regularized RL gives:

$$(1-\gamma)\,\mathbb{E}_{s\sim\rho}\left[V(s)\right] = \max_{\pi\in\Pi}\,\mathbb{E}_{(s,a)\sim d^{\pi}}\left[r(s,a) - \log\pi(a|s)\right]. \tag{35}$$

Substituting Eq. (35) into the objective of Eq. (34), the Dual V-DM problem becomes:

$$\max_{r\in\mathcal{R}}\min_{\pi\in\Pi}\ \phi_{\infty}(\pi,r), \quad \text{where} \quad \phi_{\infty}(\pi,r) := \mathbb{E}_{(s,a)\sim\mathcal{D}^{\mathrm{E}}}\left[r(s,a)\right] - \mathbb{E}_{(s,a)\sim d^{\pi}}\left[r(s,a) - \log\pi(a|s)\right]. \tag{36}$$

The optimal $\widehat{r}$ solves the outer maximization: $\widehat{r} = \mathrm{argmax}_{r\in\mathcal{R}}\ \underline{\phi}_{\infty}(r)$, where $\underline{\phi}_{\infty}(r) = \min_{\pi\in\Pi}\phi_{\infty}(\pi,r)$. The derived policy $\widehat{\pi}$ is the soft optimal policy with respect to $\widehat{r}$, i.e., the unique solution to the inner minimization:

$$\widehat{\pi}(a|s) = \frac{\exp\left(\widehat{r}(s,a) + \gamma\,\mathbb{E}_{s'\sim P(\cdot|s,a)}\left[\widehat{V}(s')\right]\right)}{\sum_{a'\in\mathcal{A}}\exp\left(\widehat{r}(s,a') + \gamma\,\mathbb{E}_{s'\sim P(\cdot|s,a')}\left[\widehat{V}(s')\right]\right)},$$

which is the soft optimal policy $\pi^{\star,\mathrm{soft},\widehat{r}}$ in the infinite-horizon setting.

The AIL problem in Eq. (30) is the min-max counterpart: $\min_{\pi\in\Pi}\max_{r\in\mathcal{R}}\phi_{\infty}(\pi,r)$. We apply Sion's minimax theorem (Sion, 1958). Transitioning to the occupancy measure space, we define:

$$\mathrm{D}_{\infty} = \left\{d \geq 0 : \sum_{a\in\mathcal{A}} d(s,a) = (1-\gamma)\rho(s) + \gamma\sum_{(s',a')\in\mathcal{S}\times\mathcal{A}} d(s',a')P(s|s',a'),\ \forall s\in\mathcal{S}\right\},$$

and the objective in occupancy measure space:

$$J_{\infty}(d,r) = \mathbb{E}_{(s,a)\sim\mathcal{D}^{\mathrm{E}}}\left[r(s,a)\right] - \mathbb{E}_{(s,a)\sim d}\left[r(s,a) - \log\pi_d(a|s)\right],$$

where $\pi_d(a|s) = d(s,a)/\sum_{a'} d(s,a')$. There is a bijection between $\mathrm{D}_{\infty}$ and $\Pi$ (Syed et al., 2008). The objective $J_{\infty}(d,r)$ is linear (hence concave) in $r$ and strictly convex in $d$ due to the entropy term. Since $\mathrm{D}_{\infty}$ and $\mathcal{R}$ are both compact and convex, Sion's minimax theorem applies, yielding strong duality:

$$\min_{\pi\in\Pi}\max_{r\in\mathcal{R}}\phi_{\infty}(\pi,r) = \min_{d\in\mathrm{D}_{\infty}}\max_{r\in\mathcal{R}} J_{\infty}(d,r) = \max_{r\in\mathcal{R}}\min_{d\in\mathrm{D}_{\infty}} J_{\infty}(d,r) = \max_{r\in\mathcal{R}}\min_{\pi\in\Pi}\phi_{\infty}(\pi,r).$$

Let $\pi^{\mathrm{AIL}} \in \mathrm{argmin}_{\pi}\max_r\phi_{\infty}(\pi,r)$ be the AIL solution. By strong duality, $(\pi^{\mathrm{AIL}}, \widehat{r})$ is a saddle point of $\phi_{\infty}$, so $\pi^{\mathrm{AIL}} \in \mathrm{argmin}_{\pi}\phi_{\infty}(\pi,\widehat{r})$. Since the inner minimization given $\widehat{r}$ is a maximum-entropy RL problem with a unique solution $\widehat{\pi}$, we conclude $\pi^{\mathrm{AIL}} = \widehat{\pi}$. $\qquad\square$

**Proposition 5** (Extension of Proposition 3 to Infinite-horizon MDPs). *Consider infinite-horizon stationary MDPs. Suppose that $\widetilde{Q}$ is the optimal solution to Dual Q-DM in Eq. (32), and $\widetilde{r}$ and $\widetilde{V}$ are the derived reward and V-function, respectively, i.e.,*

$$\forall(s,a) \in \mathcal{S} \times \mathcal{A}, \quad \widetilde{r}(s,a) = \widetilde{Q}(s,a) - \gamma\,\mathbb{E}_{s'\sim P(\cdot|s,a)}\left[\mathrm{LSE}(\widetilde{Q})(s')\right],$$

$$\widetilde{V}(s) = \text{LSE}(\widetilde{Q})(s) = \log\left(\sum_{a' \in \mathcal{A}} \exp\left(\widetilde{Q}(s, a')\right)\right).$$

*Then $(\widetilde{r}, \widetilde{V})$ is the optimal solution to Dual V-DM in Eq. (34).*

*Proof.* We first show that $(\widetilde{r}, \widetilde{V})$ is a feasible solution to Dual V-DM in Eq. (34).

From the Bellman constraints of Dual Q-DM in Eq. (32):

$$0 \leq \widetilde{Q}(s, a) - \gamma \mathbb{E}_{s' \sim P(\cdot|s,a)}\left[\text{LSE}(\widetilde{Q})(s')\right] \leq 1, \quad \forall (s, a) \in \mathcal{S} \times \mathcal{A},$$

which directly gives $0 \leq \widetilde{r}(s, a) \leq 1$ for all $(s, a)$.

Next, we verify that $\widetilde{V}$ satisfies the soft Bellman optimality equation in Eq. (34):

$$
\begin{aligned}
\text{LSE}(\widetilde{r} + \gamma P\widetilde{V})(s) &= \log \sum_{a \in \mathcal{A}} \exp\left(\widetilde{r}(s, a) + \gamma \mathbb{E}_{s' \sim P(\cdot|s,a)}\left[\widetilde{V}(s')\right]\right) \\
&= \log \sum_{a \in \mathcal{A}} \exp\left(\widetilde{Q}(s, a) - \gamma \mathbb{E}_{s'}\left[\text{LSE}(\widetilde{Q})(s')\right] + \gamma \mathbb{E}_{s'}\left[\text{LSE}(\widetilde{Q})(s')\right]\right) \\
&= \log \sum_{a \in \mathcal{A}} \exp\left(\widetilde{Q}(s, a)\right) = \text{LSE}(\widetilde{Q})(s) = \widetilde{V}(s).
\end{aligned}
$$

Together, $(\widetilde{r}, \widetilde{V})$ is a feasible solution to Dual V-DM in Eq. (34).

We now prove that $(\widetilde{r}, \widetilde{V})$ achieves the optimal value of Dual V-DM in Eq. (34).

By the definition of $\widetilde{r}$ and $\widetilde{V}$:

$$
\begin{aligned}
&\mathbb{E}_{(s,a) \sim \mathcal{D}^{\text{E}}}\left[\widetilde{r}(s, a)\right] - (1 - \gamma)\mathbb{E}_{s \sim \rho}\left[\widetilde{V}(s)\right] \\
&= \mathbb{E}_{(s,a) \sim \mathcal{D}^{\text{E}}}\left[\widetilde{Q}(s, a) - \gamma\mathbb{E}_{s' \sim P(\cdot|s,a)}\left[\text{LSE}(\widetilde{Q})(s')\right]\right] - (1 - \gamma)\mathbb{E}_{s \sim \rho}\left[\text{LSE}(\widetilde{Q})(s)\right].
\end{aligned}
$$

For any feasible solution $(r, V)$ to Dual V-DM in Eq. (34), define the corresponding Q-function:

$$\forall (s, a) \in \mathcal{S} \times \mathcal{A}, \quad Q(s, a) = r(s, a) + \gamma \mathbb{E}_{s' \sim P(\cdot|s,a)}\left[V(s')\right].$$

Since $V$ is feasible and satisfies $V(s) = \text{LSE}(r + \gamma PV)(s)$:

$$V(s) = \log \sum_{a \in \mathcal{A}} \exp\left(r(s, a) + \gamma \mathbb{E}_{s'}\left[V(s')\right]\right) = \log \sum_{a \in \mathcal{A}} \exp(Q(s, a)) = \text{LSE}(Q)(s).$$

Therefore $r(s, a) = Q(s, a) - \gamma \mathbb{E}_{s' \sim P(\cdot|s,a)}[\text{LSE}(Q)(s')] \in [0, 1]$, confirming that $Q$ is feasible for Dual Q-DM in Eq. (32). Moreover:

$$
\begin{aligned}
&\mathbb{E}_{(s,a) \sim \mathcal{D}^{\text{E}}}\left[r(s, a)\right] - (1 - \gamma)\mathbb{E}_{s \sim \rho}\left[V(s)\right] \\
&= \mathbb{E}_{(s,a) \sim \mathcal{D}^{\text{E}}}\left[Q(s, a) - \gamma\mathbb{E}_{s' \sim P(\cdot|s,a)}\left[\text{LSE}(Q)(s')\right]\right] - (1 - \gamma)\mathbb{E}_{s \sim \rho}\left[\text{LSE}(Q)(s)\right].
\end{aligned}
$$

Since $\widetilde{Q}$ is the optimal solution to Dual Q-DM in Eq. (32) and $Q$ is a feasible solution:

$$
\begin{aligned}
&\mathbb{E}_{(s,a) \sim \mathcal{D}^{\text{E}}}\left[\widetilde{Q}(s, a) - \gamma\mathbb{E}_{s' \sim P(\cdot|s,a)}\left[\text{LSE}(\widetilde{Q})(s')\right]\right] - (1 - \gamma)\mathbb{E}_{s \sim \rho}\left[\text{LSE}(\widetilde{Q})(s)\right] \\
&\overset{(a)}{\geq} \mathbb{E}_{(s,a) \sim \mathcal{D}^{\text{E}}}\left[Q(s, a) - \gamma\mathbb{E}_{s' \sim P(\cdot|s,a)}\left[\text{LSE}(Q)(s')\right]\right] - (1 - \gamma)\mathbb{E}_{s \sim \rho}\left[\text{LSE}(Q)(s)\right] \\
&= \mathbb{E}_{(s,a) \sim \mathcal{D}^{\text{E}}}\left[r(s, a)\right] - (1 - \gamma)\mathbb{E}_{s \sim \rho}\left[V(s)\right],
\end{aligned}
$$

where inequality (a) holds because $\widetilde{Q}$ is optimal and $Q$ is feasible for Dual Q-DM in Eq. (32). This shows $(\widetilde{r}, \widetilde{V})$ achieves a value no less than any feasible $(r, V)$, and is therefore an optimal solution to Dual V-DM in Eq. (34). $\qquad\square$

*Proof of Theorem 4.* Suppose that $\widetilde{Q}$ is the optimal solution to Dual Q-DM in Eq. (32). Following Proposition 5, we define the derived reward and V-function:

$$\forall(s,a) \in \mathcal{S} \times \mathcal{A}, \quad \widetilde{r}(s,a) = \widetilde{Q}(s,a) - \gamma \, \mathbb{E}_{s' \sim P(\cdot|s,a)} \left[ \text{LSE}(\widetilde{Q})(s') \right],$$

$$\widetilde{V}(s) = \text{LSE}(\widetilde{Q})(s) = \log \left( \sum_{a' \in \mathcal{A}} \exp \left( \widetilde{Q}(s,a') \right) \right).$$

By Proposition 5, $(\widetilde{r}, \widetilde{V})$ is the optimal solution to Dual V-DM in Eq. (34). Proposition 4 further implies that

$$\forall(s,a) \in \mathcal{S} \times \mathcal{A}, \quad \widetilde{\pi}(a|s) \propto \exp \left( \widetilde{r}(s,a) + \gamma \, \mathbb{E}_{s' \sim P(\cdot|s,a)} \left[ \widetilde{V}(s') \right] \right)$$

is the optimal solution to AIL in Eq. (30). Since $\widetilde{r}(s,a) = \widetilde{Q}(s,a) - \gamma \, \mathbb{E}_{s'}[\text{LSE}(\widetilde{Q})(s')] = \widetilde{Q}(s,a) - \gamma \, \mathbb{E}_{s' \sim P(\cdot|s,a)}[\widetilde{V}(s')]$, the derived policy simplifies to:

$$\widetilde{\pi}(a|s) \propto \exp \left( \widetilde{Q}(s,a) \right),$$

which is exactly the softmax policy $\pi_{\widetilde{Q}} = \text{softmax}(\widetilde{Q})$. Therefore, $\pi_{\widetilde{Q}}$ is the optimal solution to AIL in Eq. (30), which finishes the proof. $\qquad \square$

## E. Technical Lemmas

### E.1. Basic Technical Lemmas

**Lemma 2.** *Consider two Q-functions $Q$ and $Q'$. For any timestep $h \in [H-1]$, in state-action pairs $(s_{h+1}, a_{h+1}) \in \mathcal{S}_{h+1}^{\uparrow} \times \mathcal{A}^{\uparrow}$, $Q'(s_{h+1}, a_{h+1}) = Q(s_{h+1}, a_{h+1}) + \delta(s_{h+1}, a_{h+1})$ for certain $\delta(s_{h+1}, a_{h+1}) > 0$ and in the remaining state-action pairs $(s_{h+1}, a_{h+1}) \notin \mathcal{S}_{h+1}^{\uparrow} \times \mathcal{A}^{\uparrow}$, $Q'(s_{h+1}, a_{h+1}) = Q(s_{h+1}, a_{h+1})$. Besides, we have that $\forall(s_h, a_h) \in \mathcal{S}_h \times \mathcal{A}, r_Q(s_h, a_h) = r_{Q'}(s_h, a_h)$. Then it holds that*

$$0 \leq \max_{(s_h, a_h) \in \mathcal{S}_h \times \mathcal{A}_h} Q'(s_h, a_h) - Q(s_h, a_h) \leq \max_{(s_{h+1}, a_{h+1}) \in \mathcal{S}_{h+1}^{\uparrow} \times \mathcal{A}^{\uparrow}} \delta(s_{h+1}, a_{h+1}).$$

*If $\mathcal{A}^{\uparrow} \subset \mathcal{A}$, we have that*

$$0 \leq \max_{(s_h, a_h) \in \mathcal{S}_h \times \mathcal{A}_h} Q'(s_h, a_h) - Q(s_h, a_h) < \max_{(s_{h+1}, a_{h+1}) \in \mathcal{S}_{h+1}^{\uparrow} \times \mathcal{A}^{\uparrow}} \delta(s_{h+1}, a_{h+1}).$$

*Proof.* We have that $Q'$ is greater than $Q$ in certain state-action pairs in time step $h + 1$. To maintain $\forall(s_h, a_h) \in \mathcal{S}_h \times \mathcal{A}, r_Q(s_h, a_h) = r_{Q'}(s_h, a_h)$, we obtain that

$$Q'(s_h, a_h) - Q(s_h, a_h) = \mathbb{E}_{s' \sim P(\cdot|s_h, a_h)} \left[ \text{LSE}(Q')(s') \right] - \mathbb{E}_{s' \sim P(\cdot|s_h, a_h)} \left[ \text{LSE}(Q)(s') \right]$$

$$= \sum_{s_{h+1} \in \mathcal{S}_{h+1}^{\uparrow}} P(s_{h+1}|s_h, a_h) \left( \text{LSE}(Q')(s_{h+1}) - \text{LSE}(Q)(s_{h+1}) \right).$$

Since $\text{LSE}(Q')(s_{h+1}) \geq \text{LSE}(Q)(s_{h+1})$, we can get the lower bound.

$$Q'(s_h, a_h) - Q(s_h, a_h) \geq \sum_{s_{h+1} \in \mathcal{S}_{h+1}^{\uparrow}} P(s_{h+1}|s_h, a_h) \geq 0.$$

For the upper bound, we have that

$$Q'(s_h, a_h) - Q(s_h, a_h) = \sum_{s_{h+1} \in \mathcal{S}_{h+1}^{\uparrow}} P(s_{h+1}|s_h, a_h) \left( \text{LSE}(Q')(s_{h+1}) - \text{LSE}(Q)(s_{h+1}) \right)$$

$$\leq \max_{s_{h+1} \in \mathcal{S}_{h+1}^{\uparrow}} \left( \text{LSE}(Q')(s_{h+1}) - \text{LSE}(Q)(s_{h+1}) \right)$$

$$\leq \max_{s_{h+1}\in\mathcal{S}_{h+1}^{\uparrow}} \log\left(\frac{\sum_{a\in\mathcal{A}}\exp(Q'(s_{h+1},a))}{\sum_{a\in\mathcal{A}}\exp(Q(s_{h+1},a))}\right)$$

$$= \max_{s_{h+1}\in\mathcal{S}_{h+1}^{\uparrow}} \log\left(\frac{\sum_{a\in\mathcal{A}^{\uparrow}}\exp(Q'(s_{h+1},a))+\sum_{a\notin\mathcal{A}^{\uparrow}}\exp(Q'(s_{h+1},a))}{\sum_{a\in\mathcal{A}^{\uparrow}}\exp(Q(s_{h+1},a))+\sum_{a\notin\mathcal{A}^{\uparrow}}\exp(Q(s_{h+1},a))}\right)$$

We define that $\delta_{h+1} := \max_{(s_{h+1},a_{h+1})\in\mathcal{S}_{h+1}^{\uparrow}\times\mathcal{A}^{\uparrow}}\delta(s_{h+1},a_{h+1})$. Then it holds that

$$Q'(s_h,a_h) - Q(s_h,a_h) = \max_{s_{h+1}\in\mathcal{S}_{h+1}^{\uparrow}} \log\left(\frac{\sum_{a\in\mathcal{A}^{\uparrow}}\exp(Q'(s_{h+1},a))+\sum_{a\notin\mathcal{A}^{\uparrow}}\exp(Q'(s_{h+1},a))}{\sum_{a\in\mathcal{A}^{\uparrow}}\exp(Q(s_{h+1},a))+\sum_{a\notin\mathcal{A}^{\uparrow}}\exp(Q(s_{h+1},a))}\right)$$

$$\leq \max_{s_{h+1}\in\mathcal{S}_{h+1}^{\uparrow}} \log\left(\frac{(\sum_{a\in\mathcal{A}^{\uparrow}}\exp(Q(s_{h+1},a)))\exp(d)+\sum_{a\notin\mathcal{A}^{\uparrow}}\exp(Q(s_{h+1},a))}{\sum_{a\in\mathcal{A}^{\uparrow}}\exp(Q(s_{h+1},a))+\sum_{a\notin\mathcal{A}^{\uparrow}}\exp(Q(s_{h+1},a))}\right)$$

$$\leq \delta.$$

Furthermore, if $\mathcal{A}^{\uparrow}\subset\mathcal{A}$, we have the following strict inequality.

$$Q'(s_h,a_h) - Q(s_h,a_h) < \delta.$$

We complete the proof. $\qquad\square$

**Lemma 3.** *Suppose that $\widetilde{Q}$ is the Q-function learned by Dual Q-DM via Eq. (8) and $r_{\widetilde{Q}}(s_h,a) := \widetilde{Q}(s_h,a) -$ $\mathbb{E}_{s'\sim P(\cdot|s_h,a)}\left[\mathrm{LSE}(\widetilde{Q})(s')\right]$. Then $r_{\widetilde{Q}}$ is the optimal solution to Eq. (37).*

$$\max_r \mathbb{E}_{\tau\sim\mathcal{D}^{\mathrm{E}}}\left[\sum_{h=1}^{H}r(s_h,a_h)\right] - \mathbb{E}_{s_1\sim\rho}\left[V^{\star,\mathrm{soft},r}(s_1)\right] \tag{37}$$

$$\text{s.t. } 0 \leq r(s_h,a) \leq 1, \forall(h,s_h,a)\in[H]\times\mathcal{S}_h\times\mathcal{A}.$$

*Proof.* $\widetilde{Q}$ is the optimal solution to

$$\max_Q \mathbb{E}_{\tau\sim\mathcal{D}^{\mathrm{E}}}\left[\sum_{h=1}^{H}Q(s_h,a_h)-\mathbb{E}_{s'\sim P(\cdot|s_h,a_h)}[\mathrm{LSE}(Q)(s')]\right]-\mathbb{E}_{s_1\sim\rho}[\mathrm{LSE}(Q)(s_1)]$$

$$\text{s.t. } 0 \leq Q(s_h,a)-\mathbb{E}_{s'\sim P(\cdot|s_h,a)}[\mathrm{LSE}(Q)(s')] \leq 1, \forall(h,s_h,a)\in[H]\times\mathcal{S}_h\times\mathcal{A}.$$

Bellman constraints ensure that $0 \leq r_{\widetilde{Q}}(s_h,a) \leq 1, \forall(h,s_h,a)\in[H]\times\mathcal{S}_h\times\mathcal{A}$, implying that $r_{\widetilde{Q}}$ is a feasible solution to Eq. (37). Notice that $\widetilde{Q}$ satisfies the soft Bellman optimality equation regarding $r_{\widetilde{Q}}$, implying that $\widetilde{Q}$ is the soft optimal Q-function regarding $r_{\widetilde{Q}}$, i.e., $\widetilde{Q} = Q^{\star,\mathrm{soft},r_{\widetilde{Q}}}$. Then the objective function can be reformulated as

$$\mathbb{E}_{\tau\sim\mathcal{D}^{\mathrm{E}}}\left[\sum_{h=1}^{H}\widetilde{Q}(s_h,a_h)-\mathbb{E}_{s'\sim P(\cdot|s_h,a_h)}[\mathrm{LSE}(\widetilde{Q})(s')]\right]-\mathbb{E}_{s_1\sim\rho}[\mathrm{LSE}(\widetilde{Q})(s_1)]$$

$$= \mathbb{E}_{\tau\sim\mathcal{D}^{\mathrm{E}}}\left[\sum_{h=1}^{H}r_{\widetilde{Q}}(s_h,a_h)\right]-\mathbb{E}_{s_1\sim\rho}[V^{\star,\mathrm{soft},r_{\widetilde{Q}}}(s_1)].$$

For any reward $r$ satisfying that $0 \leq r(s_h,a) \leq 1, \forall(h,s_h,a)\in[H]\times\mathcal{S}_h\times\mathcal{A}$, let $Q^{\star,\mathrm{soft},r}$ denote the corresponding soft optimal Q-function. It is direct to have that $Q^{\star,\mathrm{soft},r}$ is a feasible solution to Dual Q-DM in Eq. (8). Since $\widetilde{Q}$ is an optimal solution to Dual Q-DM in Eq. (8), we have that

$$\mathbb{E}_{\tau\sim\mathcal{D}^{\mathrm{E}}}\left[\sum_{h=1}^{H}\widetilde{Q}(s_h,a_h)-\mathbb{E}_{s'\sim P(\cdot|s_h,a_h)}[\mathrm{LSE}(\widetilde{Q})(s')]\right]-\mathbb{E}_{s_1\sim\rho}[\mathrm{LSE}(\widetilde{Q})(s_1)]$$

$$\geq \mathbb{E}_{\tau\sim\mathcal{D}^{\mathrm{E}}}\left[\sum_{h=1}^{H}Q^{\star,\mathrm{soft},r}(s_h,a_h)-\mathbb{E}_{s'\sim P(\cdot|s_h,a_h)}[\mathrm{LSE}(Q^{\star,\mathrm{soft},r})(s')]\right]-\mathbb{E}_{s_1\sim\rho}[\mathrm{LSE}(Q^{\star,\mathrm{soft},r})(s_1)].$$

This is equivalent to

$$\mathbb{E}_{\tau \sim \mathcal{D}^{\mathrm{E}}}\left[\sum_{h=1}^{H} r_{\widetilde{Q}}(s_h, a_h)\right] - \mathbb{E}_{s_1 \sim \rho}[V^{\star, \mathrm{soft}, r_{\widetilde{Q}}}(s_1)] \geq \mathbb{E}_{\tau \sim \mathcal{D}^{\mathrm{E}}}\left[\sum_{h=1}^{H} r(s_h, a_h)\right] - \mathbb{E}_{s_1 \sim \rho}[V^{\star, \mathrm{soft}, r}(s_1)].$$

This implies that $r_{\widetilde{Q}}$ is the optimal solution to Eq. (37), which completes the proof.

$\square$

**Lemma 4.** *In the Reset Cliff MDP with initial state distribution* $\rho = \{\frac{1}{N+1}, \cdots, \frac{1}{N+1}, 1 - \frac{S-2}{N+1}, 0\}$, *then* $\sum_{s_h \in S_h \setminus \{b_h\}} \rho(s_h)(1 - \rho(s_h))^N \geq \frac{S-2}{e(N+1)}$.

*Proof.* $\frac{S-2}{N+1}(\frac{N}{N+1})^N + (1 - \frac{S-2}{N+1})(\frac{S-2}{N+1})^N \geq \frac{S-2}{N+1}(\frac{N}{N+1})^N = \frac{S-2}{N+1}\frac{1}{(1+\frac{1}{N})^N} \geq \frac{S-2}{e(N+1)}$, where the last inequality follows that $(1 + \frac{1}{N})^N \leq e$. $\square$

### E.2. Proof of Lemma 1

We prove the first claim by contradiction. We assume that the first claim does not hold and $\exists h \in [H], \forall (s_h^{\mathrm{E}}, a_h^{\mathrm{E}}) \in \mathcal{D}^{\mathrm{E}}$,

$$\widetilde{Q}(s_h^{\mathrm{E}}, a_h^{\mathrm{E}}) - \mathbb{E}_{s' \sim P_h(\cdot | s_h^{\mathrm{E}}, a_h^{\mathrm{E}})}\left[\mathrm{LSE}(\widetilde{Q})(s')\right] < 1. \tag{38}$$

For ease of presentation, we define the reward induced by Q-function as

$$r_Q(s_h, a) = Q(s_h, a) - \mathbb{E}_{s' \sim P(\cdot | s_h, a)}\left[\mathrm{LSE}(Q)(s')\right].$$

We will prove that $\widetilde{Q}$ is not an optimal solution to Dual Q-DM in Eq. (8), thereby yielding a contradiction. We construct another Q-function $\overline{Q}$. In timestep $h + 1 \leq \ell \leq H$, $\overline{Q}$ is identical to $\widetilde{Q}$, i.e., $\overline{Q}(s_\ell, a) = \widetilde{Q}(s_\ell, a), \forall (s_\ell, a) \in \mathcal{S}_\ell \times \mathcal{A}$. This implies that $r_{\overline{Q}}(s_\ell, a) = r_{\widetilde{Q}}(s_\ell, a), \forall (s_\ell, a) \in \mathcal{S}_\ell \times \mathcal{A}$.

Notably, $\overline{Q}$ differs from $\widetilde{Q}$ in timestep $\ell \in [h]$. In step $h$, we define that

$$\delta(s_h, a) := 1 - \left(\widetilde{Q}(s_h, a) - \mathbb{E}_{s' \sim P_h(\cdot | s_h, a)}\left[\mathrm{LSE}(\widetilde{Q})(s')\right]\right).$$

According to Eq. (38), we have that $\forall (s_h^{\mathrm{E}}, a_h^{\mathrm{E}}) \in \mathcal{D}^{\mathrm{E}}, \delta(s_h^{\mathrm{E}}, a_h^{\mathrm{E}}) > 0$ and thus $\delta := \min_{(s_h^{\mathrm{E}}, a_h^{\mathrm{E}}) \in \mathcal{D}^{\mathrm{E}}} \delta(s_h^{\mathrm{E}}, a_h^{\mathrm{E}}) > 0$. Thus we construct $\overline{Q}$ as

$$\forall (s_h^{\mathrm{E}}, a_h^{\mathrm{E}}) \in \mathcal{D}^{\mathrm{E}}, \overline{Q}(s_h^{\mathrm{E}}, a_h^{\mathrm{E}}) = \widetilde{Q}(s_h^{\mathrm{E}}, a_h^{\mathrm{E}}) + \delta.$$

For other state-action pairs $(s_h, a_h) \notin \mathcal{D}^{\mathrm{E}}$,

$$\overline{Q}(s_h, a_h) = \widetilde{Q}(s_h, a_h).$$

We have finished the construction of $\overline{Q}$ in time step $h$ and obtain that

$$\forall (s_h^{\mathrm{E}}, a_h^{\mathrm{E}}) \in \mathcal{D}^{\mathrm{E}}, r_{\overline{Q}}(s_h^{\mathrm{E}}, a_h^{\mathrm{E}}) = r_{\widetilde{Q}}(s_h^{\mathrm{E}}, a_h^{\mathrm{E}}) + \delta, \forall (s_h, a_h) \notin \mathcal{D}^{\mathrm{E}}, r_{\overline{Q}}(s_h, a_h) = r_{\widetilde{Q}}(s_h, a_h),$$

where $\delta > 0$.

Then we further construct $\overline{Q}$ in preceding time steps $1 \leq \ell \leq h - 1$ such that $r_{\overline{Q}}(s_\ell, a_\ell) = r_{\widetilde{Q}}(s_\ell, a_\ell), \forall (s_\ell, a_\ell) \times \mathcal{S}_\ell \times \mathcal{A}$. We perform the construction via backward induction. In time step $h - 1$, we apply the second claim in Lemma 2 and obtain that

$$0 \leq \max_{(s_{h-1}, a_{h-1}) \in \mathcal{S}_{h-1} \times \mathcal{A}_h} \overline{Q}(s_{h-1}, a_{h-1}) - Q(s_{h-1}, a_{h-1}) < \max_{(s_h, a_h) \in \mathcal{S}_h^{\uparrow} \times \mathcal{A}^{\uparrow}} \delta(s_h, a_h) = \delta.$$

This means that the growth magnitude in time step $h - 1$ is strictly less than that in time step $h$. Then we consider the following two cases.

- **Case I:** $\max_{(s_{h-1},a_{h-1})\in\mathcal{S}_{h-1}\times\mathcal{A}_h}\overline{Q}(s_{h-1},a_{h-1})-Q(s_{h-1},a_{h-1})=0$. As $\overline{Q}$ is identical to $Q$ in time step $h-1$, we can simply keep that $Q(s_\ell,a_\ell)=\overline{Q}(s_\ell,a_\ell),\forall(s_\ell,a_\ell)\in\mathcal{S}_\ell\times\mathcal{A}$ in preceding timesteps $1\leq\ell\leq h-2$.

- **Case II:** $\max_{(s_{h-1},a_{h-1})\in\mathcal{S}_{h-1}\times\mathcal{A}}\overline{Q}(s_{h-1},a_{h-1})-Q(s_{h-1},a_{h-1})>0$. We can apply the first claim in Lemma 2 and obtain that

$$\max_{(s_{h-2},a_{h-2})\in\mathcal{S}_{h-2}\times\mathcal{A}}\overline{Q}(s_{h-2},a_{h-2})-Q(s_{h-2},a_{h-2})\leq\max_{(s_{h-1},a_{h-1})\in\mathcal{S}_{h-1}^{\uparrow}\times\mathcal{A}^{\uparrow}}\delta(s_{h-1},a_{h-1})<\delta.$$

In both cases, we obtain that

$$\max_{(s_{h-2},a_{h-2})\in\mathcal{S}_{h-2}\times\mathcal{A}}\overline{Q}(s_{h-2},a_{h-2})-Q(s_{h-2},a_{h-2})<\delta.$$

Then we can repeat the above analysis in the backward direction and obtain that

$$\forall 1\leq\ell\leq h-2,\ \max_{(s_\ell,a_\ell)\in\mathcal{S}_\ell\times\mathcal{A}}\overline{Q}(s_\ell,a_\ell)-Q(s_\ell,a_\ell)<\delta.$$

We have finished the construction of $\overline{Q}$, which satifies that

$$\forall h+1\leq\ell\leq H,r_{\overline{Q}}(s_\ell,a_\ell)=r_{\widetilde{Q}}(s_\ell,a_\ell),\forall(s_\ell,a_\ell)\in\mathcal{S}_\ell\times\mathcal{A},$$
$$\forall(s_h^{\mathrm{E}},a_h^{\mathrm{E}})\in\mathcal{D}^{\mathrm{E}},r_{\overline{Q}}(s_h^{\mathrm{E}},a_h^{\mathrm{E}})=r_{\widetilde{Q}}(s_h^{\mathrm{E}},a_h^{\mathrm{E}})+\delta,\forall(s_h,a_h)\notin\mathcal{D}^{\mathrm{E}},r_{\overline{Q}}(s_h,a_h)=r_{\widetilde{Q}}(s_h,a_h),$$
$$\forall 1\leq\ell\leq h-1,r_{\overline{Q}}(s_\ell,a_\ell)=r_{\widetilde{Q}}(s_\ell,a_\ell),\forall(s_\ell,a_\ell)\in\mathcal{S}_\ell\times\mathcal{A},$$
$$\forall(s_1,a_1)\in\mathcal{S}_1\times\mathcal{A},\overline{Q}(s_1,a_1)-\widetilde{Q}(s_1,a_1)<\delta.$$

Then we compare the objectives of $\overline{Q}$ and $\widetilde{Q}$. We denote the objective as

$$\mathcal{L}(Q)=\mathbb{E}_{\tau\sim\mathcal{D}^{\mathrm{E}}}\left[\sum_{h=1}^{H}Q(s_h,a_h)-\mathbb{E}_{s'\sim P(\cdot|s_h,a_h)}\left[\mathrm{LSE}(Q)(s')\right]\right]-\mathbb{E}_{s_1\sim\rho}\left[\mathrm{LSE}(Q)(s_1)\right]$$

Then we can have that

$$\mathcal{L}(\overline{Q})-\mathcal{L}(\widetilde{Q})$$
$$=\mathbb{E}_{\tau\sim\mathcal{D}^{\mathrm{E}}}\left[\sum_{h=1}^{H}r_{\overline{Q}}(s_h,a_h)\right]-\mathbb{E}_{\tau\sim\mathcal{D}^{\mathrm{E}}}\left[\sum_{h=1}^{H}r_{\widetilde{Q}}(s_h,a_h)\right]+\mathbb{E}_{s_1\sim\rho}\left[\mathrm{LSE}(\widetilde{Q})(s_1)\right]-\mathbb{E}_{s_1\sim\rho}\left[\mathrm{LSE}(\overline{Q})(s_1)\right]$$
$$=\sum_{h=1}^{H}\sum_{(s_h,a_h)\in\mathcal{S}_h\times\mathcal{A}}\widehat{d_h^{\pi^{\mathrm{E}}}}(s_h,a_h)\left(r_{\overline{Q}}(s_h,a_h)-r_{\widetilde{Q}}(s_h,a_h)\right)+\mathbb{E}_{s_1\sim\rho}\left[\mathrm{LSE}(\widetilde{Q})(s_1)\right]-\mathbb{E}_{s_1\sim\rho}\left[\mathrm{LSE}(\overline{Q})(s_1)\right]$$
$$=\delta\sum_{(s_h,a_h)\in\mathcal{S}_h\times\mathcal{A}}\widehat{d_h^{\pi^{\mathrm{E}}}}(s_h,a_h)+\sum_{s_1\in\mathcal{S}_1}\rho(s_1)\left(\mathrm{LSE}(\widetilde{Q})(s_1)-\mathrm{LSE}(\overline{Q})(s_1)\right)$$
$$=\delta+\sum_{s_1\in\mathcal{S}_1}\rho(s_1)\left(\mathrm{LSE}(\widetilde{Q})(s_1)-\mathrm{LSE}(\overline{Q})(s_1)\right)$$
$$>0.$$

This implies that $\overline{Q}$ achieves a strictly larger objective value than $\widetilde{Q}$, which contradicts the fact that $\widetilde{Q}$ is the optimal solution. Therefore, the assumption does not hold and $\forall h\in[H],\exists(s_h^{\mathrm{E}},a_h^{\mathrm{E}})\in\mathcal{D}^{\mathrm{E}}$ such that $\widetilde{Q}(s_h^{\mathrm{E}},a_h^{\mathrm{E}})-\mathbb{E}_{s'\sim P(\cdot|s_h^{\mathrm{E}},a_h^{\mathrm{E}})}\left[\mathrm{LSE}(\widetilde{Q})(s')\right]=1$. We finish the proof of the first claim.

Then we prove the second claim by leveraging the connection between Eq. (8) and Eq. (37). According to Lemma 3, we have that $r_{\widetilde{Q}}$ is the optimal solution to Eq. (37). Then we prove the desired claim by characterizing the optimal solution to Eq. (37). In particular, we denote the objective function as

$$\mathcal{L}(r)=\mathbb{E}_{\tau\sim\mathcal{D}^{\mathrm{E}}}\left[\sum_{h=1}^{H}r(s_h,a_h)\right]-\mathbb{E}_{s_1\sim\rho}\left[V^{\star,\mathrm{soft},r}(s_1)\right].$$

Then we calculate its gradient

$$\frac{\partial \mathcal{L}(r)}{\partial r(s_h, a_h)} = \widehat{d_h^{\pi^{\mathrm{E}}}}(s_h, a_h) - d_h^{\pi^{\star,\mathrm{soft},r}}(s_h, a_h).$$

Here $\pi^{\star,\mathrm{soft},r}$ is the soft-optimal policy w.r.t $r$. For any $(s_h, a_h) \notin \mathcal{D}^{\mathrm{E}}$, we have that $\frac{\partial \mathcal{L}(r)}{\partial r(s_h, a_h)} = -d_h^{\pi^{\star,\mathrm{soft},r}}(s_h, a_h) < 0$. The strict inequality holds because the soft-optimal policy supports on the whole action space, i.e., $\forall a_h \in \mathcal{A}, \pi^{\star,\mathrm{soft},r}(a_h|s_h) > 0$. Then we obtain that $\mathcal{L}(r)$ is a monotonically decreasing function w.r.t $r(s_h, a_h)$ and thus achieves the optimal solution on the lower bound $r(s_h, a_h) = 0$. Then we have that $\forall (s_h, a_h) \notin \mathcal{D}^{\mathrm{E}}, r_{\widetilde{Q}}(s_h, a_h) = 0$, which finishes the proof of the second claim.

# F. Experiment Details

## F.1. Implementation Details

The experiments are conducted on a machine with 32 CPU cores and 1 RTX5090 GPU cores. Each experiment is replicated three times using different random seeds. For each task, we adopt SAC (Haarnoja et al., 2018) to train an agent with sufficient environment interactions and use the resultant policy as the expert policy. We use 1M environment interactions for each task and then we roll out this expert policy to collect expert demonstrations. We use small amount of expert demonstrations to train each method. The number and subsample ratio of expert trajectories used for training in each task is provided in 3. The architecture and training details of Dual Q-DM and all baselines are listed below.
**Dual Q-DM:** Our codebase of Dual Q-DM extends the open-sourced framework of IQ-Learn. We retain the structure and parameter design of the critic from the original framework, and employ SAC with a fixed temperature coefficient for policy update. A comprehensive enumeration of the hyperparameters of Dual Q-DM is provided in Table 4.
**BC:** We implement BC based on our codebase. The actor model is trained using Mean Squared Error (MSE) loss over 50k training steps.
**IQ-Learn and Reg IQ-Learn:** We use the author's codebase, which is avaiable at https://github.com/Div99/IQ-Learn. A comprehensive enumeration of the hyperparameters of IQ-Learn is provided in Table 5.
**LS-IQ:** We use the author's codebase, which is avaiable at https://github.com/robfiras/ls-iq. A comprehensive enumeration of the hyperparameters of LS-IQ is provided in Table 6.
**ReCOIL:** We use the author's codebase, which is avaiable at https://github.com/hari-sikchi/DVL. A comprehensive enumeration of the hyperparameters of ReCOIL is provided in Table 9 .
**HyPE:** We use the author's codebase, which is available at https://github.com/gkswamy98/hyper. A comprehensive enumeration of the hyperparameters of HyPE is provided in Table 7.
**DAC:** We use the codebase which is available at https://github.com/Kaixhin/imitation-learning. A comprehensive enumeration of the hyperparameters of DAC is provided in Table 8.

*Table 3.* Number and subsample ratio of expert trajectories for each task.

| Task | Expert Trajectories |
|---|---|
| Ant-v2 | 1 |
| HalfCheetah-v2 | 1 |
| Hopper-v2 | 5 |
| Humanoid-v2 | 5 |
| Walker2d-v2 | 1 |

*Table 4.* Dual Q-DM Hyper-parameters.

| Parameter | Value |
|---|---|
| discount factor | 0.99 |
| replay buffer size | $1 \cdot 10^6$ |
| batch size | 256 |
| optimizer | Adam |
| *Actor* | |
| learning rate | $3 \cdot 10^{-5}$ |
| number of hidden layers | 2 |
| number of hidden units per layer | 256 |
| activation | ReLU |
| *Critic* | |
| learning rate | $3 \cdot 10^{-4}$ |
| number of hidden layers | 2 |
| number of hidden units per layer | 256 |
| activation | ReLU |

*Table 5.* IQ-Learn Hyper-parameters.

| Parameter | Value |
|---|---|
| discount factor | 0.99 |
| replay buffer size | $1 \cdot 10^6$ |
| batch size | 256 |
| optimizer | Adam |
| *Actor* | |
| learning rate | $3 \cdot 10^{-5}$ |
| number of hidden layers | 2 |
| number of hidden units per layer | 256 |
| activation | ReLU |
| *Critic* | |
| learning rate | $3 \cdot 10^{-4}$ |
| number of hidden layers | 2 |
| number of hidden units per layer | 256 |
| activation | ReLU |

*Table 6.* LS-IQ Hyper-parameters.

| Parameter | Value |
|---|---|
| discount factor | 0.99 |
| replay buffer size | $1 \cdot 10^6$ |
| batch size | 256 |
| optimizer | Adam |
| *Actor* | |
| learning rate | $3 \cdot 10^{-5}$ |
| number of hidden layers | 2 |
| number of hidden units per layer | 256 |
| activation | ReLU |
| *Critic* | |
| learning rate | $3 \cdot 10^{-4}$ |
| number of hidden layers | 2 |
| number of hidden units per layer | 256 |
| activation | ReLU |

*Table 7.* HyPE Hyper-parameters.

| Parameter | Value |
|---|---|
| discount factor | 0.98 |
| gradient penalty coefficient | 10 |
| replay buffer size | $1 \cdot 10^6$ |
| batch size | 256 |
| optimizer | Adam |
| *Reward* | |
| learning rate | $8 \cdot 10^{-4}$ |
| batch size | 4096 |
| number of hidden layers | 2 |
| number of hidden units per layer | 256 |
| *Actor* | |
| learning rate | $7.3 \cdot 10^{-4}$ |
| number of hidden layers | 2 |
| number of hidden units per layer | 256 |
| activation | ReLU |
| *Critic* | |
| learning rate | $7.3 \cdot 10^{-4}$ |
| number of hidden layers | 2 |
| number of hidden units per layer | 256 |
| activation | ReLU |

*Table 8.* DAC Hyper-parameters.

| Parameter | Value |
|---|---|
| discount factor | 0.97 |
| gradient penalty coefficient | 1 |
| replay buffer size | $1 \cdot 10^6$ |
| batch size | 256 |
| optimizer | Adam |
| *Reward* | |
| learning rate | $3 \cdot 10^{-5}$ |
| batch size | 256 |
| number of hidden layers | 1 |
| number of hidden units per layer | 64 |
| *Actor* | |
| learning rate | $3 \cdot 10^{-4}$ |
| number of hidden layers | 2 |
| number of hidden units per layer | 256 |
| activation | ReLU |
| *Critic* | |
| learning rate | $3 \cdot 10^{-4}$ |
| number of hidden layers | 2 |
| number of hidden units per layer | 256 |
| activation | ReLU |

*Table 9.* ReCOIL Hyper-parameters.

| Parameter | Value |
|---|---|
| discount factor | 0.99 |
| batch size | 256 |
| optimizer | Adam |
| *Actor* | |
| learning rate | $1 \cdot 10^{-4}$ |
| number of hidden layers | 2 |
| number of hidden units per layer | 256 |
| temperature | 1 |
| *Critic* | |
| learning rate | $1 \cdot 10^{-4}$ |
| number of hidden layers | 2 |
| number of hidden units per layer | 256 |
| *Value* | |
| learning rate | $1 \cdot 10^{-4}$ |
| number of hidden layers | 2 |
| number of hidden units per layer | 256 |
| $\tau$ | 1 |

## F.2. Additional Experimental Results

**Experiments in the Offline Setting.** We evaluate BC, IQ-Learn, and Reg IQ-Learn in the offline setting, where no online environment interactions are available during training. In the offline setting, IQ-Learn utilizes the initial states in demonstrations to estimate the last term in its objective of Eq. (4), while Reg IQ-Learn uses expert demonstrations to establish the regularization. Figure 6 reports the training curves across 5 MuJoCo tasks. All three methods exhibit similarly poor performance and fail to learn an effective policy. This result directly corroborates Theorem 1 that IQ-Learn reduces to BC.

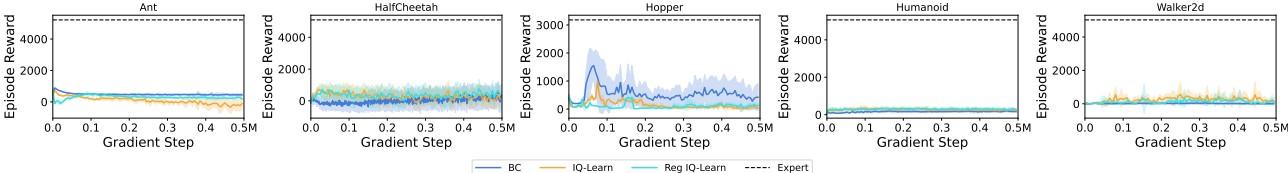

*Figure 6.* Training curves of BC, IQ-Learn, and Reg IQ-Learn across 5 MuJoCo tasks under the offline setting, where solid lines denote the mean and shaded regions denote the standard deviation over 5 seeds. The x-axis represents the number of gradient update steps and the y-axis represents return. All three methods exhibit similarly poor performance, failing to learn an effective policy, providing direct empirical support for Theorem 1.

**Sensitivity Analysis.** We analyze the sensitivity of Dual Q-DM to its two key hyperparameters: the Bellman constraint bound and the penalty coefficient $\beta$. Specifically, the bound controls the upper and lower limits of the Bellman constraints (i.e., $-\text{Bound} \leq Q(s,a) - \mathbb{E}_{s'}[\text{LSE}(Q)(s')] \leq \text{Bound}$), while $\beta$ controls the strength of the Bellman constraint penalty. Figure 7 reports training curves on HalfCheetah and Humanoid under a wide range of values for each hyperparameter. Dual Q-DM maintains strong performance across most configurations, with performance degradation observed only under extreme settings—specifically, very large bounds or very small $\beta$ in the Humanoid environment. These results confirm that Dual Q-DM is robust to hyperparameter choices and does not require careful tuning in practice.

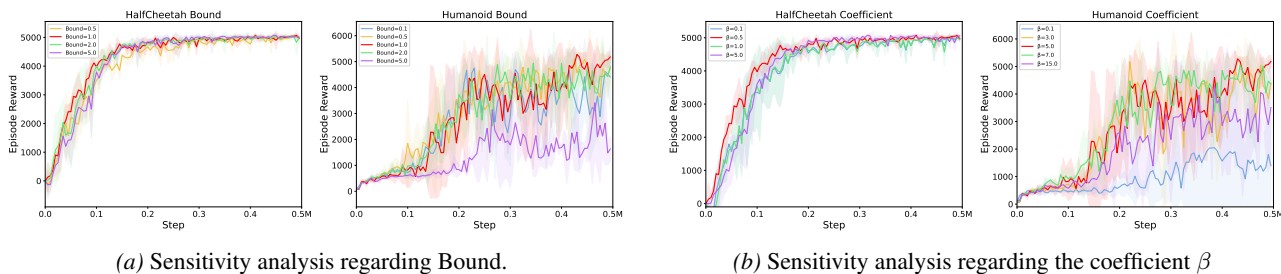

*(a)* Sensitivity analysis regarding Bound.                                    *(b)* Sensitivity analysis regarding the coefficient $\beta$

*Figure 7.* Hyperparameter sensitivity analysis of Dual Q-DM on the HalfCheetah and Humanoid MuJoCo tasks. For HalfCheetah, we vary Bound $\in [0.5, 1.0, 2.0, 5.0]$ and $\beta \in [0.1, 0.5, 1.0, 5.0]$; for Humanoid, we vary Bound $\in [0.1, 0.5, 1.0, 2.0, 5.0]$ and $\beta \in [0.1, 3.0, 5.0, 7.0, 15.0]$. The left and right panels show sensitivity to the bound and coefficient, respectively, with the x-axis representing environment interactions and the y-axis representing return. Dual Q-DM maintains strong and robust performance across a wide range of hyperparameter values, with degradation observed only in the Humanoid environment under extremely large bounds or small coefficients. These results confirm that Dual Q-DM is robust to hyperparameter variations.

