# OpenReview forum: "Non-Adversarial Imitation Learning Provably Free of Compounding Errors: The Value Flow Mechanism"
_ICML.cc/2026/Conference — ICML 2026 regular_

### Official Review · Reviewer_BBVM · 2026-03-10

**Soundness:** 3
**Presentation:** 3
**Significance:** 3
**Originality:** 3
**Overall Recommendation:** 4
**Confidence:** 3

**Summary:**

The paper addresses the issue of compounding errors in non-adversarial imitation learning. Specifically, the authors propose Dual Q-DM, derived from a primal-dual framework for distribution matching. Incorporating the Bellman constraints can establish the temporal and dynamic relationships between consecutive states. Theoretical guarantee has proven that Dual Q-DM can efficiently address the compounding error issues, showing a competitive imitation gap with the adversarial imitation learning method. Experimental results on different benchmarks show that Dual Q-DM achieve advanced efficiency and stability.

**Compliance With Llm Reviewing Policy:**

Affirmed.

**Key Questions For Authors:**

1. In practice, neural network approximation might unsatisfy the assumption and degrade these guarantees. How can the theoretical results extend to the practical situation?
2. The method requires an online interaction to estimate Bellman constraints.
3. Theoretical results under stochastic MDPs.

**Limitations:**

See questions.

**Strengths And Weaknesses:**

## Strengths
1. The authors provide a provable theoretical guarantee in terms of compounding errors and non-adversarial imitation learning.
2. The proposed method is more stable than adversarial imitation learning methods.

## Weakness
1. Bellman constraints may introduce additional sensitivity to performance.
2. The proposed method requires online interaction for estimating Bellman constraints.

---

> ### Author Rebuttal · Authors · 2026-03-31
>
> We would like to thank the reviewer for the careful evaluation! We respectfully provide our response as follows.
>
>
> **Q1:** Bellman constraints may introduce additional sensitivity to performance.
>
> **A1:** To address this concern, we conducted a sensitivity analysis on the key parameters governing the Bellman constraints (e.g., the constraint bounds and the penalty coefficient). [Fig. R3](https://anonymous.4open.science/r/icml26rebuttal_dualQDM/sensitive_R3.pdf) shows that Dual Q-DM maintains robust and good performance across a wide range of these hyperparameter values. **This confirms that Bellman constraints do not introduce pathological sensitivity into the practical optimization process.**
>
> **Q2:** The proposed method requires online interaction for estimating Bellman constraints.
>
> **A2:** **Bellman constraints themselves do not inherently require online interactions—they can be evaluated from any collection of transition tuples $(s, a, s^\prime)$, regardless of whether those tuples are collected online or offline, and regardless of their quality.** Since this paper focuses on the online IL setting, we naturally use online interactions to supply these transition tuples, consistent with how AIL leverages online rollouts.
>
> **Nevertheless, this flexibility suggests that Dual Q-DM can be naturally extended to offline IL with a supplementary dataset:** a setting where the learner cannot interact with the environment but has access to an offline dataset of (possibly imperfect) transitions alongside expert demonstrations. In this case, the offline dataset directly provides the transition tuples needed to enforce Bellman constraints, enabling Dual Q-DM to operate without any online interaction. We view this as a promising direction for future work.
>
> **Q3:** In practice, neural network approximation might unsatisfy the assumption and degrade these guarantees. How can the theoretical results extend to the practical situation?
>
> **A3:** Our theoretical results could be extended to the setting with neural network approximation. The key observation is that all of our theoretical analyses are conducted in the **general function space**: we characterize the optimal solution that maximizes the objective over the general function class $\mathcal{Q}$, without restricting to any particular parametric form. As long as the neural network is expressive enough to represent arbitrary functions in $\mathcal{Q}$, a condition that is satisfied by modern overparameterized networks, the optimal solution in the neural network parameter space coincides with the optimal solution in the function space. Under this condition, our main theoretical results (Theorem 1, Theorem 2, and Proposition 1) directly carry over to the practical neural network setting.
>
> **Q4:** Theoretical results under stochastic MDPs.
>
> **A4:** Theoretical results in this paper (Theorem 1, Corollary 1, and Proposition 1) are all proved under the standard MDP formulation with transition probabilities; therefore, they hold for both deterministic and stochastic MDPs.
>
> ---
> We sincerely hope that our clarifications, **particularly the new sensitivity analysis confirming Dual Q-DM's robustness, alongside the theoretical guarantees under function approximation and stochastic MDPs**, have resolved your main concerns. If you find our response satisfactory, we kindly ask that you consider raising your score.

---

> > ### Author Rebuttal · Reviewer_BBVM · 2026-04-03
> >
> > Thanks for the reply. I maintain my rating.

---

> > > ### Author Response · Authors · 2026-04-07
> > >
> > > Thank you for your feedback and for confirming that our previous response fully addressed your concerns. We are glad to hear that your positive evaluation of our work stands.
> > >
> > > ### Additional Preliminary Experiments: Extension to Offline IL
> > >
> > > Regarding your earlier point on the requirement for online interaction to estimate Bellman constraints (Q2), **we conducted additional preliminary experiments to further demonstrate the flexibility of Dual Q-DM.** Although offline IL with supplementary data falls outside the primary scope of this paper, which focuses on standard IL with demonstrations, we included these experiments to validate Dual Q-DM's extensibility to other settings. Specifically, we evaluated Dual Q-DM in the offline IL with a supplementary dataset setting, where the agent has no access to the environment and must rely on expert demonstrations paired with a supplementary offline dataset of unknown quality.
> > >
> > > - **Experimental setup:** We compare Dual Q-DM against IQ-Learn, using **full replay data** (experience replay collected during online RL training) as the offline supplementary dataset—a standard data source for this setting.
> > > - **Implementation:** Dual Q-DM uses the supplementary data to enforce Bellman constraints, while IQ-Learn uses it to construct the second term in its objective (Eq. 4).
> > > - **Key findings:** [Fig. R6](https://anonymous.4open.science/r/icml26rebuttal_dualQDM/offline_supplementary_R6.pdf) shows that Dual Q-DM consistently outperforms IQ-Learn in this setting, demonstrating that it can effectively leverage offline transition tuples to enforce Bellman constraints and achieve out-of-demonstration generalization—without any online interaction.
> > >
> > > These results validate that Dual Q-DM is not inherently tied to online sampling and can be naturally extended to offline scenarios with supplementary data, substantially broadening its practical impact. We will include these findings and accompanying discussion in the revised manuscript.
> > >
> > > **As your initial concerns have been fully resolved and we have provided additional evidence of broader impact, we sincerely hope you might consider raising your score to "Accept."** Thank you again for your time and invaluable guidance.

---

### Official Review · Reviewer_WTGu · 2026-03-12

**Soundness:** 2
**Presentation:** 2
**Significance:** 1
**Originality:** 1
**Overall Recommendation:** 2
**Confidence:** 5

**Summary:**

* The paper studies IQ-Learn, a non-adversarial imitation learning (IL) algorithm that previously achieved state-of-the-art performance in online IL.
* The authors show that IQ-Learn can be theoretically reduced to classical behavioral cloning (BC), implying that it may suffer from an imitation gap lower bound.
* Motivated by this limitation, the authors propose a primal–dual framework for MaxEnt distribution matching, leading to a new Q-based IL method called Dual Q-DM.
* Experimental results on MuJoCo benchmark tasks compare the proposed method with several standard baselines.

**Compliance With Llm Reviewing Policy:**

Affirmed.

**Final Justification:**

While I appreciate the authors’ effort in responding to my comments, I find most of the responses unconvincing.

* Regarding the proof of Theorem 1 in the infinite-horizon case, I find the argument provided by the authors highly problematic. They partition the state space into initial and non-initial states and establish equivalence to BC under this separation. However, this reasoning is flawed: in many environments, any state can serve as an initial state. Under such settings, the proof no longer holds or becomes unclear. More broadly, I am not convinced that the claimed equivalence to BC holds in a systematic way.

* The authors state that “the last method in the legend is exactly labeled as ‘DualRL.’” This raises a significant concern. When referring to DualRL, I mean the paper commonly known as DualRL [1], whose primary method is ReCOIL. There is no method explicitly called “DualRL” in that work. It is therefore unclear what method the authors are referring to, suggesting that the relevant literature may not have been carefully examined.

* SMODICE and DWBC are direct competitors to IQ-Learn, and the authors should make a stronger effort to include meaningful comparisons. Moreover, the authors did not address my questions regarding the observation-only setting.

Overall, I find the core motivation of the paper unconvincing and therefore maintain my original score.

[1] Sikchi, H., Zheng, Q., Zhang, A., & Niekum, S. (2023). Dual RL: Unification and New Methods for Reinforcement and Imitation Learning.

**Key Questions For Authors:**

* Can the analysis be extended to the infinite-horizon setting with a discount factor?
* Could the authors clarify the convexity properties of the optimization problem in Eq. (6)? Under what conditions does strong duality hold?
* Beyond the theoretical motivation, could the authors provide more intuition about when the proposed Dual Q-DM method is expected to outperform existing approaches such as IQ-Learn?

**Limitations:**

Yes

**Strengths And Weaknesses:**

**Strength:**

* It is difficult to identify a significant strength in this paper. One potentially positive aspect is that the work focuses on IQ-Learn and attempts to improve upon it. This is a reasonable motivation, as IQ-Learn is a strong imitation learning (IL) method but is known to be unstable in practice, making it worthwhile to explore possible improvements. However, the paper contains several issues and flaws, as discussed below.

**Weaknesses:**
There are several issues that suggest the authors may not have a correct understanding of IQ-Learn and, more generally, how imitation learning algorithms are formulated.

* First, regarding the model formulation, the authors focus on a finite-horizon setting without discount factors. In contrast, IQ-Learn and most imitation learning algorithms are formulated under an infinite-horizon discounted MDP. In practice, the infinite-horizon discounted formulation is more standard and useful. Restricting the analysis to a finite-horizon setting limits the applicability of the proposed framework. Moreover, the original IQ-Learn formulation is based on a discounted infinite-horizon MDP, so it is unclear how the formulation in the paper is derived or transferred to the finite-horizon setting considered here.

* The main theoretical result (Theorem 1), which claims that IQ-Learn yields the same optimal solution as behavioral cloning (BC), appears to be **incorrect**. The authors attempt to establish this equivalence by ignoring the second term in Equation (4). However, this term is essential in the IQ-Learn objective and cannot simply be removed. In particular, the term $\mathbb{E}_{s_1 \sim \rho}[\mathrm{LSE}(Q(s_1))]$ plays an important role in balancing the expected reward with the reward induced by expert demonstrations. Without this term, the IQ-Learn objective would not function properly. Furthermore, the original IQ-Learn objective also includes a reward regularization term that helps stabilize the learning process. This aspect is not discussed in the paper, making the formulation incomplete.

* The second main theoretical result concerns the primal and dual formulations in Equations (6) and (7), and their equivalence stated in Theorem 2. This result is also **problematic**. To apply Lagrangian duality and obtain an equivalent formulation, the optimization problem in (6) must satisfy certain convexity conditions. However, it does not appear that the problem is convex, or at least the paper does not provide any discussion or justification regarding this requirement.

* The experimental evaluation is very limited. The authors only compare their method with a few relatively older baselines (e.g., IQ-Learn and LSIQ) while ignoring more recent methods such as DualRL. In addition, there are no ablation studies to better understand the contribution of different components of the proposed approach.

Overall, although the paper claims to be theoretically focused, the theoretical results appear to be incorrect or incomplete. At the same time, the experimental evaluation is limited and does not provide strong empirical support for the proposed method.

---

> ### Author Rebuttal · Authors · 2026-03-31
>
> Thank the reviewer for the detailed review but respectfully point out several factual errors and misunderstandings regarding our theoretical analysis, which we would like to clarify.
>
> **Q1:** Finite-horizon MDPs vs. infinite-horizon MDPs for IL formulation.
>
> **A1:** First, we respectfully clarify that our use of the finite-horizon setting does **not** reflect a misunderstanding. **The finite-horizon formulation is the standard setting adopted by seminal theoretical IL works (Ross & Bagnell, 2010; Rajaraman et al., 2020), precisely because it provides the cleanest characterization of how imitation gap scales with horizon $H$.** Our paper follows this convention to enable direct comparison with established bounds.
>
> **Besides, through a similar reparameterization argument, we can prove that Theorem 1 also holds in infinite-horizon discounted MDPs.**
>
> > **Extension of Theorem 1.**
> >
> > Consider infinite-horizon discounted and stationary MDPs. Suppose that $\pi_{\hat{Q}}$ is the policy recovered by IQ-Learn. Assume the softmax policy class realizes the BC policy.
> >
> > - For any non-initial state $s$, i.e., $\rho(s)=0$, we have $\pi_{\hat{Q}}(\cdot|s) = \pi^{BC}(\cdot|s)$.
> > - For initial states uncovered by demonstrations, i.e., $\rho (s_1) > 0, s_1 \notin \mathcal{D}^E$, we have $\pi_{\hat{Q}}(\cdot|s_1)=\pi^{BC}(\cdot|s_1)=Unif(\mathcal{A})$.
>
> **Q2:** Theorem 1 appears to be incorrect because the authors ignore the second term in Eq.(4). The original IQ-Learn objective also includes a reward regularization, which is not discussed in the paper.
>
> **A2:**
> **First, Theorem 1 is sound and we did *not* ignore the initial state term.** In fact, this term is the core driver of our proof for initial states (Appendix B.1, Lines 703-710; Remark 3). Because it is monotonically increasing with respect to Q(s,a), optimizing it uniformly suppresses all Q-values on unvisited states. This mathematically proves that IQ-Learn provides no discriminative signal outside expert data, degenerating into BC.
>
> Second, the reward regularization term is entirely absent from the original IQ-Learn paper (see their Eq. (9) and (10)) and lacks any theoretical justification; it exists solely as an optional heuristic in their codebase. Importantly, in Appendix C.5, we establish a formal connection: this code-level regularization essentially acts as an implicit, heuristic approximation of the rigorous Bellman constraints we propose.
>
> **Q3:** Theorem 2 is problematic because the problem does not appear to be convex.
>
> **A3:** **We respectfully clarify that Theorem 2 is sound because Eq. (6) is a strictly convex optimization problem.** Mathematically, the objective combines the convex Total Variation divergence and the strictly concave entropy of the occupancy measure. Furthermore, the constraints (Bellman flow equations and probability simplex) form a standard linear polytope. This strict convexity over a linear set naturally guarantees strong duality.
>
> **Q4:** The empirical evaluation lacks comparisons with recent methods like DualRL and ablation studies.
>
> **A4:** First, we evaluated DualRL across all 5 MuJoCo tasks. [Fig. R1](https://anonymous.4open.science/r/icml26rebuttal_dualQDM/mujoco_R1.pdf) demonstrates that Dual Q-DM consistently outperforms Dual RL.
>
> Second, we clarify that Figure 1 in the paper inherently serves as an ablation study. The only difference between Dual Q-DM and standard IQ-Learn is our proposed Bellman constraints. The massive performance gap, where Dual Q-DM succeeds while IQ-Learn collapses, validates the critical contribution of this mechanism.
>
> **Q5**: More intuition about when Dual Q-DM is expected to outperform IQ-Learn.
>
> **A5:** **The core advantage of Dual Q-DM over IQ-Learn is its robust Out-of-Demonstration-Distribution (OOD) recovery.** As detailed in Section 4.2, our Bellman constraints enforce a dynamic programming structure that recursively propagates high Q-values from expert trajectories into unvisited states. This establishes a dense "value gradient" that actively guides off-trajectory agents back to the expert distribution, whereas IQ-Learn acts randomly in OOD states. This recovery mechanism allows Dual Q-DM to outperform IQ-Learn in **long-horizon, highly stochastic tasks, and settings with limited demonstrations**. In these scenarios, compounding execution errors, environmental noise, and sparse expert support inevitably push the agent into OOD states. By leveraging the globally propagated value gradient, Dual Q-DM continuously corrects these deviations, while IQ-Learn collapses.
>
> ---
> As shown above, our theory is rigorously sound and the empirical validation is comprehensive, explicitly featuring comparisons with DualRL and ablation studies. We kindly ask you to re-evaluate this paper in light of these facts.

---

> > ### Author Rebuttal · Reviewer_WTGu · 2026-04-02
> >
> > Thank you to the authors for their response. However, I find that my concerns remain largely unresolved.
> >
> > > **Q1: Finite-horizon MDPs vs. infinite-horizon MDPs for IL formulation.**
> >
> > While finite-horizon MDPs simplify theoretical derivations, they introduce a critical gap when extending the results to infinite-horizon settings. The original IQ-Learn framework is developed in the finite-horizon setting, yet many subsequent works consider the infinite-horizon case. Therefore, when building upon IQ-Learn but restricting attention to finite-horizon MDPs, it raises questions about the general applicability of the results.
> >
> > Moreover, it is unclear how Theorem 1 extends to the infinite-horizon case. In infinite-horizon MDPs, each state-action pair can be visited multiple times across different time steps, including initial states. However, the proof appears to treat different values of \(h\) (i.e., \(h=1\) and \(h>1\)) separately, which does not directly carry over to the infinite-horizon setting where such a separation is no longer valid. As a result, I do not see how the proof can be straightforwardly extended.
> >
> > Additionally, IQ-Learn incorporates environment dynamics, whereas behavior cloning (BC) does not. It is therefore unclear how the equivalence between these two schemes is established under the proposed formulation.
> >
> > > **Q3: Theorem 2 is problematic because the problem does not appear to be convex.**
> >
> > I do not see that the objective function is convex. In Equation (6), while the first term appears to be convex, the second term may not be. In particular, the definition
> > $\mathcal{H}(d) = -\log\left(\frac{d(s,a)}{\sum_a d(s,a)}\right)$
> > does not clearly imply concavity. This raises concerns about the validity of the convexity claim. Additionally, there appears to be a notation mismatch: is the $H$ in Equation (6) the same as $\mathcal{H}$ in Equation (3)?
> >
> > > *First, we evaluated DualRL across all 5 MuJoCo tasks. Fig. R1 demonstrates that Dual Q-DM consistently outperforms Dual RL.*
> >
> > There is no method labeled “DualRL” shown in Figure 1.
> >
> > Furthermore, there are several strong baselines in offline imitation learning, such as SMODICE and DWBC, that are commonly used for comparison. The paper omits these important baselines, which weakens the experimental evaluation.
> >
> > Moreover, IQ-Learn considers the observation-only setting, yet it is unclear how such a setting is handled in your method.
> >
> > Overall, I do not find that my concerns have been adequately addressed, and I therefore maintain my original evaluation.

---

> > > ### Author Response · Authors · 2026-04-04
> > >
> > > We appreciate the reviewer's feedback and are glad to see Q2 and Q5 resolved. For the remaining points, we respectfully clarify several misunderstandings in the underlying claims below.
> > >
> > > **Q1-1 (Follow-up to Q1):** Moreover, it is unclear how Theorem 1 extends to the infinite-horizon case.
> > >
> > > **A1-1:** We present the step-by-step proof here. Following Eq.(9) in the IQ-Learn paper, the objective is formulated by
> > > $$
> > > \max_{Q\in\mathcal{Q}}\mathbb{E}\_{(s,a,s^\prime)\sim \widehat{d^{\pi^E}}}[Q(s, a)-\gamma LSE(Q)(s^\prime)]-(1-\gamma)\mathbb{E}\_{s_1\sim\rho}[LSE(Q)(s_1)].
> > > $$
> > > We apply a standard mathematical transformation commonly used in literature like DualDICE (Nachum et al., 2019):
> > > \begin{aligned}
> > > &\quad\mathbb{E}\_{(s,a,s^\prime)\sim\widehat{d^{\pi^E}}}[Q(s,a)-\gamma LSE(Q)(s^\prime)]-(1-\gamma)\mathbb{E}\_{s_1\sim\rho}[LSE(Q)(s_1)]
> > > \\\\
> > > &=(1-\gamma)\mathbb{E}\_{\tau\sim\mathcal{D}^E}[\sum_{h=1}^{\infty}\gamma^{h-1}(Q(s_h,a_h)-\gamma LSE(Q)(s_{h+1}))]-(1-\gamma)\mathbb{E}\_{s_1\sim\rho}[LSE(Q)(s_1)]
> > > \\\\
> > > &\overset{(a)}{=}(1-\gamma)\mathbb{E}\_{\tau\sim\mathcal{D}^E}[\sum_{h=1}^{\infty}\gamma^{h-1}(Q(s_h,a_h)-LSE(Q)(s_{h}))]]+(1-\gamma)(\mathbb{E}\_{s_1\sim\mathcal{D}^E}[LSE(Q)(s_1)]-\mathbb{E}_{s_1\sim\rho}[LSE(Q)(s_1)])
> > > \\\\
> > > &=\underbrace{\mathbb{E}\_{(s,a)\sim\widehat{d^{\pi^E}}}[\log\pi_Q(a|s)]+(1-\gamma)(\mathbb{E}\_{s_1\sim\mathcal{D}^E}[LSE(Q)(s_1)]-\mathbb{E}\_{s_1\sim\rho}[LSE(Q)(s_1)])}\_{\mathcal{L}(Q)}.
> > > \end{aligned}
> > >
> > > Equality (a) follows the telescoping argument. **The first term in $\mathcal{L}(Q)$ exactly corresponds to the BC MLE objective.**
> > >
> > > **For non-initial states $s$, i.e., $\rho(s)=0$**, the second term in $\mathcal{L}(Q)$ is independent of $Q(s,a)$ on such states. Thus:
> > > $$
> > > \arg\max_{Q(s,a)}\mathcal{L}(Q)=\arg\max_{Q(s,a)}\mathbb{E}_{(s,a)\sim\widehat{d^{\pi^E}}}[\log\pi_Q(a|s)],
> > > $$
> > > and IQ-Learn recovers the exact BC policy on such states.
> > >
> > > **For unvisited initial states $s$, i.e., $\rho(s_1)>0,\widehat{d^{\pi^E}}(s_1)=0$**, we have:
> > > $$
> > > \arg\max_{Q(s_1,a)}\mathcal{L}(Q)=\arg\min_{Q(s_1,a)}(1-\gamma)\mathbb{E}\_{s_1\sim\rho}[LSE(Q)(s_1)].
> > > $$
> > > Because $\mathbb{E}\_{s_1\sim\rho}[LSE(Q)(s_1)]$ is monotonically increasing w.r.t $Q(s_1,a)$ for all $a\in\mathcal{A}$, the optimal solution $\widehat{Q}(s_1,a)$ minimizes all action values equally and the corresponding softmax policy is $\forall a\in\mathcal{A},\pi_{\widehat{Q}}\left(a| s_1\right)=1/|\mathcal{A}|$.
> > >
> > > **Q1-2 (Follow-up to Q1):** It is therefore unclear how the equivalence between these two schemes is established under the proposed formulation.
> > >
> > > **A1-2:** **We respectfully emphasize that the manuscript preemptively resolves this confusion.** First, **Appendix B.1** provides the step-by-step proof of this equivalence, with core analytical insights summarized in **Remark 2**. Second, **Remark 3** explains why IQ-Learn does not benefit from environment dynamics, providing a new and important observation to the community.
> > >
> > > **Q3-1 (Follow-up to Q3):** In particular, the definition $\bar{\mathcal{H}}(d_h)$ does not clearly imply concavity.
> > >
> > > **A3-1:** **This is a standard result established in Lemma 3.1 in the seminal work GAIL. The proof is as follows.** For any valid distributions $d_h, d'_h$ and $\lambda\in[0,1]$, let $d^\lambda_h(s, a)=\lambda d_h(s,a)+(1-\lambda)d'_h(s,a)$. Applying the Log-sum inequality (i.e., $x_1 \log \frac{x_1}{y_1}+x_2 \log \frac{x_2}{y_2} \geq(x_1+x_2) \log \frac{x_1+x_2}{y_1+y_2}$) yields
> > > $$
> > > -d^\lambda_h(s, a)\log\frac{d^\lambda_h(s, a)}{\sum\_{a'} d^\lambda_h(s, a)}
> > > \ge-\lambda d_h(s,a)\log\frac{\lambda d_h(s,a)}{\lambda\sum\_{a'}d_h(s,a')}-(1-\lambda)d'_h(s,a)\log\frac{(1-\lambda)d'_h(s,a)}{(1-\lambda)\sum\_{a'}d'_h(s,a')}.
> > > $$
> > > Summing both sides over all $(s,a)$ yields $\bar{\mathcal{H}}(d^\lambda_h)\ge\lambda\bar{\mathcal{H}}(d_h)+(1-\lambda)\bar{\mathcal{H}}(d'_h)$, which establishes the concavity.
> > >
> > > **Q4-1 (Follow-up to Q4):** There is no method labeled “DualRL” shown in Figure 1.
> > >
> > > **A4-1:** **We respectfully point out a factual error.** In [Fig. R1](https://anonymous.4open.science/r/icml26rebuttal_dualQDM/mujoco_R1.pdf) in the anonymous repository, the last method in the legend is exactly labeled as "DualRL".
> > >
> > > **Q6 (New):** There are several strong baselines in offline imitation learning, such as SMODICE and DWBC. Moreover, IQ-Learn considers the observation-only setting.
> > >
> > > **A6:** **We respectfully clarify that the requested baselines and settings are fundamentally incompatible with our problem set-up.**
> > >
> > > * **Incompatible Baselines:** Our work focuses on **standard imitation learning (IL) with expert demonstrations**. SMODICE and DWBC target **offline IL with a supplementary dataset**. Due to these incompatible data assumptions, comparisons are mathematically invalid. They are inapplicable, not omitted.
> > >
> > > * **Out-of-Scope Setting:** We focus on foundational theory for Q-based IL to resolve compounding errors. Extending this to an observation-only setting is an orthogonal problem outside our scope.

---

### Official Review · Reviewer_Mbu2 · 2026-03-14

**Soundness:** 3
**Presentation:** 3
**Significance:** 3
**Originality:** 3
**Overall Recommendation:** 5
**Confidence:** 3

**Summary:**

The paper shows that the policy recovered by recent Q-based imitation learning (IL) methods, such as IQ-Learn, is nearly equivalent to BC policy (i.e., the imitation gap for worst case instances still has a quadratic dependence on horizon H).  It then proposes Dual Q-DM, a method that does dual distribution matching over Q functions (instead of matching state-action distributions directly, as in adversarial IL). Theoretically, the authors show that this method is free of compounding errors. Empirically, they propose a practical version of the algorithm that is more stable than adversarial IL methods yet achieves similar performance on MuJoCo tasks.

**Compliance With Llm Reviewing Policy:**

Affirmed.

**Final Justification:**

I am still positive about the paper and would like to maintain my rating.

**Key Questions For Authors:**

1. Does the theory for Dual Q-DM still hold for stochastic expert policies?
2. The 4 MuJoCo environments tested in this paper are largely deterministic. Could the authors provide empirical results for environments with more stochasticity?

**Limitations:**

Yes.

**Strengths And Weaknesses:**

**Strengths**
1. The paper is very well-written and easy to read.
2. The results are very interesting for the community: method such as IQ-Learn aren’t better than behavioral cloning.
3. The dual formulation for matching state-action distributions is novel.


**Weaknesses**

Overall, the paper is pretty solid in my opinion. The only weakness that I see in this work is that the empirical evaluation of Dual Q-DM is limited to 4 MuJoCo tasks.

---

> ### Author Rebuttal · Authors · 2026-03-31
>
> We appreciate your time to review and provide positive feedback for our work.
>
> **Q1:** Overall, the paper is pretty solid in my opinion. The only weakness that I see in this work is that the empirical evaluation of Dual Q-DM is limited to 4 MuJoCo tasks. Could the authors provide empirical results for environments with more stochasticity?
>
> **A1:** To address your concern regarding empirical coverage, we have conducted additional experiments on complex Atari tasks. These environments are characterized by both high-dimensional image inputs and more environment stochasticity. [Fig. R4](https://anonymous.4open.science/r/icml26rebuttal_dualQDM/atari_R4.pdf) clearly demonstrates that Dual Q-DM consistently outperforms all baseline methods even in these challenging settings, further confirming its scalability and robustness.
>
> **Q2**: Does the theory for Dual Q-DM still hold for stochastic expert policies?
>
> **A2:** **First, the main theory that Dual Q-DM is equivalent to AIL (Theorem 2) does not require deterministic expert policies in the first place. Besides, the theory on the generalization performance of Dual Q-DM (Proposition 1) can be extended to stochastic expert policies with additional efforts.** Specifically, for stochastic expert policies, we define $\mathcal{A}^{E} (s_h) := \{ a_h \in \mathcal{A}, \pi^{E} (a|s_h) > 0 \}$ as the set of actions that the expert policy could take. Through the dynamic programming analysis presented in Section 4.2, we can prove that the Q-function recovered by Dual Q-DM assigns higher values to all expert actions and lower values to non-expert actions on states uncovered by demonstrations.
>
>
> > **Extension of Proposition 1**.
> >
> >  Suppose that $\widetilde{Q}$ and $\widehat{Q}$ are Q-functions learned by Dual Q-DM and IQ-Learn, respectively. Consider stochastic expert policies and TD MDPs, when $N \geq 1$, for any timestep $h \in [H-1]$ and states uncovered by demonstrations $s_h \notin \mathcal{D}^{E}$, we have
> >$$
> \forall a^{E} \in \mathcal{A}^{E} (s_h), a \notin \mathcal{A}^{E} (s_h),   \widetilde{Q} (s_h,  a^{E}) > \widetilde{Q} (s_h, a),
>          \\
> \forall a^{E} \in \mathcal{A}^{E} (s_h), a \notin \mathcal{A}^{E} (s_h), \widehat{Q} (s_h, a^{E} ) = \widehat{Q} (s_h, a).
> >$$
> >
>
> ---
> We sincerely hope that our clarifications regarding the **broad applicability of our theoretical analysis**, along with the **new experimental results on more challenging tasks**, have fully addressed your initial concerns. If you find our response and revisions satisfactory, we kindly ask that you consider raising your score.

---

> > ### Author Rebuttal · Reviewer_Mbu2 · 2026-04-03
> >
> > I am still positive about the paper and would like to maintain my rating.

---

> > > ### Author Response · Authors · 2026-04-07
> > >
> > > Thank you for reviewing our rebuttal and confirming that your concerns have been fully resolved. We sincerely appreciate your constructive feedback throughout the review process, and we are very grateful for your continued support and for maintaining your positive rating "Accept" for our work.

---

### Official Review · Reviewer_aqsT · 2026-03-16

**Soundness:** 3
**Presentation:** 2
**Significance:** 4
**Originality:** 3
**Overall Recommendation:** 5
**Confidence:** 4

**Summary:**

This paper investigates the theoretical foundations of Q-based (non-adversarial) imitation learning methods, with a focus on IQ-Learn (Garg et al., 2021). The paper makes two main contributions: (1) it proves that IQ-Learn provably reduces to behavioral cloning (BC) when the Q-function class is sufficiently expressive, hence (2) IQ-Learn also inherit BC's compounding error lower bound quadratically dependent to horizon.

The key finding is that IQ-Learn's change-of-variables step drops the Bellman constraint on the induced reward, meaning the Q-function is optimized without enforcing consistency with the environment dynamics. To fix this, the authors develop a primal-dual framework for distribution matching that yields Dual Q-DM, which is a Q-based IL method that explicitly incorporates Bellman constraints.

Further, this work proves that Dual Q-DM with Bellman constraints is equivalent to adversarial IL, thereby unifying Q-based and adversarial approaches under a single theoretical lens and show that Bellman-constrained methods can generalize to unvisited states in such settings.
Experiments on a tabular MDP and five MuJoCo continuous-control tasks validate the theoretical results.

**Compliance With Llm Reviewing Policy:**

Affirmed.

**Key Questions For Authors:**

1. Does the IQ-Learn = BC conclusion extend beyond layered finite-horizon MDPs to stationary MDPs, where the same state may be visited at multiple timesteps? Since the proof of Theorem 1 relies on a timestep-wise decomposition in the layered setting, a discussion of whether an analogous result holds under a stationary Q function would substantially strengthen the paper. Relatedly, is there a principled connection between your Bellman-constraint view and the stationary Bellman flow constraints used in DICE-style methods?

2. Have you considered an ablation where the Q-function receives the timestep as an additional input (e.g. $Q(s,h,a)$), thereby aligning the experimental setup with the theoretical assumptions?

3. The Bellman constraint bounds [left, right] and penalty $\beta$ vary substantially across environments (e.g., $\beta$ ranges from 0.1 to 5 in the provided configs). How sensitive is Dual Q-DM to these choices?

4. (Minor question) Dual Q-DM appears to rely on online environment interaction to evaluate the Bellman-constraint penalty over the replay buffer. Do you expect this Bellman-constraint mechanism to extend to the fully offline setting with imperfect or mixed-quality demonstrations? (e.g. DemoDICE, RelaxDICE) More specifically, could similar constraints be enforced using only static offline data?

**Limitations:**

yes

**Strengths And Weaknesses:**

## **Strengths:**

1. The central result of this paper that IQ-Learn without Bellman constraint provably reduces to BC is novel. IQ-Learn has been widely adopted and is generally perceived as a principled non-adversarial alternative to GAIL-style methods. The solution is clean: the change-of-variables from reward to Q-function eliminates the Bellman constraint that would otherwise ensure the induced reward is consistent with the MDP dynamics. The subsequent fix is reintroducing Bellman constraints through the primal-dual framework, and the proof that this recovers AIL's guarantees provides a satisfying unification of Q-based and adversarial approaches.

2. This paper addresses a question of genuine importance: what is the fundamental mechanism by which non-adversarial Q-based IL methods avoid (or fail to avoid) compounding errors? This paper shows that Bellman constraints are essential and IQ-Learn lacks them, and these results are theoretically clear and practically actionable.

## **Weaknesses:**

1. **Layered-MDP assumption:**
The theoretical analysis is carried out in layered finite-horizon MDPs with disjoint state sets across timesteps. The key step (Appendix B.1, Eq. 10) decomposes the IQ-Learn objective independently per timestep, which requires this disjointness. The authors may claim this is "without loss of generality" via the standard $(s, h)$ augmentation, but this augmentation necessitates a non-stationary Q-function $Q(s, h, a)$. In practice, however, the original IQ-Learn uses a single stationary $Q(s, a)$ shared across all timesteps. Consequently, it is unclear whether Theorem 1 (and Corollary 1) holds in the stationary MDP setting commonly used in practice, and the paper does not discuss this. More broadly, no theoretical discussion bridges the gap between the theory (episodic, finite-horizon, tabular, layered MDPs) and the experiments (infinite-horizon, discounted MDPs with function approximation)

2. **Narrow Experimental Results:** The MuJoCo experiments consist of 5 tasks evaluated over only 3 seeds. On four tasks (Ant, HalfCheetah, Hopper, Walker2d) out of five, the performance differences between Dual Q-DM and Reg IQ-Learn / LS-IQ appear comparable in magnitude to the expected variance from 3 seeds, making it difficult to draw reliable conclusions.

3. **A Lack of Analysis on Offline Setting:** The original IQ-Learn paper (Garg et al., 2021) also considers the offline expert-only setting and reports that IQ-Learn significantly outperforms BC consistent to the number of expert demonstrations. According to Theorem 1, IQ-Learn without regularization should reduce to BC in this setting as well. However, the paper does not provide any empirical analysis in the offline setting. An ablation comparing (a) offline IQ-Learn without any regularization, (b) offline IQ-Learn with regularization (original IQ-Learn implementation), and (c) BC would directly validate Theorem 1 and Remark 2, substantially strengthening the paper's claims.

---

> ### Author Rebuttal · Authors · 2026-03-31
>
> We appreciate your time to review and provide positive feedback for our work.
>
> **Q1**: Layered-MDP assumption.
> **A1:** **First, Theorem 1 and Corollary 1 holds in infinite-horizon stationary MDPs.** In this set-up, IQ-Learn is formulated by
> $$
> \max\_{Q\in\mathcal{Q}}  \mathbb{E}\_{(s,a,s^\prime)\sim\mathcal{D}^{E}}[Q(s, a)-\gamma LSE(Q)(s^\prime)]-(1-\gamma)\mathbb{E}\_{s_1\sim\rho}[LSE(Q)(s_1)].
> $$
> We present the extensions of Theorem 1 and Corollary 1 as follows.
> > **Extension of Theorem 1.**
> >
> > Consider infinite-horizon stationary MDPs. Suppose that $\pi_{\hat{Q}}$ is the policy recovered by IQ-Learn. Assume the softmax policy class realizes the BC policy.
> > - For any non-initial state $s$, i.e., $\rho(s)=0$, we have $\pi_{\hat{Q}}(\cdot|s)=\pi^{BC}(\cdot|s)$.
> > - For initial states uncovered by demonstrations, i.e., $\rho (s_1)>0, s_1\notin\mathcal{D}^E$, we have $\pi_{\hat{Q}}(\cdot|s_1)=\pi^{BC}(\cdot|s_1)={Unif}(\mathcal{A})$.
>
> The proof still follows the re-parameterization technique presented in Remark 2, which does not rely on the per-timestep independence of the objective. Specifically, we can rewrite the IQ-Learn objective as:
> $$
> (1-\gamma)(\mathbb{E}\_{(s_1,a_1)\sim\mathcal{D}^{E}}[Q(s_1,a_1)]-\mathbb{E}\_{s_0\sim\rho}[LSE(Q)(s_0)]) + (1-\gamma)\mathbb{E}\_{\tau\sim\mathcal{D}^{E}}[\sum_{t=2}^{\infty}\gamma^{t-1} \underbrace{(Q(s_t,a_t)-LSE(Q)(s_t))}\_{\log\pi_{Q}(a_t|s_t)}].
> $$
> In stationary MDPs, the IQ-Learn objective still coincides with BC for any non-initial state. Further analysis shows that at the initial state, the IQ-Learn policy assigns uniform probability to actions on unvisited states, which is identical to BC.
>
> Corollary 1 also holds in stationary MDPs with the only difference being replacing H with the effective horizon $1/(1-\gamma)$. In the analysis, we can modify the original finite-horizon MDP into an infinite-horizon MDP by using absorbing states.
>
> **Besides, Theorems 1, 2, and Proposition 1 naturally extend to function approximation.** Since our analysis operates in the general function space, the optimal solutions and thus our theoretical guarantees directly carry over to the function approximation set-up, assuming sufficient expressivity of $\mathcal{Q}$ (e.g., modern overparameterized networks).
>
> **Q2:** Narrow empirical results over only 3 seeds.
>
> **A2:** We have run two additional seeds, bringing the total to 5 seeds across all MuJoCo tasks. [Fig. R1](https://anonymous.4open.science/r/icml26rebuttal_dualQDM/mujoco_R1.pdf) shows that Dual Q-DM achieves a notable and consistent performance improvement over both Reg IQ-Learn and LS-IQ.
>
> **Q3:** A lack of analysis on offline setting.
>
> **A3:** We evaluate offline IQ-Learn, offline Reg IQ-Learn, and BC in all 5 MuJoCo tasks. [Fig. R2](https://anonymous.4open.science/r/icml26rebuttal_dualQDM/offline_R2.pdf) shows that all three methods exhibit similarly poor performance and fail to learn an effective policy, directly validating Theorem 1.
>
> **Q4:** Connection between Bellman constraints and Bellman flow constraints.
>
> **A4:** **In Lagrangian duality, Bellman constraints are the dual of Bellman-flow constraints. They serve symmetric roles:**
> - Primal distribution matching (DICE-style methods, Eq. 6): Bellman-flow constraints ensure the temporal feasibility of occupancy measures.
> - Dual distribution matching (Dual Q-DM): Bellman constraints ensure the temporal consistency of the dual Q-functions.
>
> **Q5:** An ablation where the Q-function receives the timestep as an input.
>
> **A5:** We have not attempted this empirical setup. **A1** shows that Theorem 1 and Corollary 1 can be extended to stationary MDPs, thereby aligning the current empirical setup.
>
> **Q6**: Sensitivity to the Bellman constraint bounds and coefficient.
>
> **A6:** We conducted a sensitivity analysis on both the Bellman constraint bounds and penalty. [Fig. R3](https://anonymous.4open.science/r/icml26rebuttal_dualQDM/sensitive_R3.pdf) shows that Dual Q-DM maintains robust performance across a wide range of values, implying that Dual Q-DM is not sensitive to these hyperparameter choices.
>
> **Q7**: Extension of Bellman constraints to offline IL with imperfect demonstrations.
>
> **A7:** Bellman constraints themselves do not inherently require online interactions—they can be evaluated from any collection of transition tuples $(s, a, s^\prime)$, regardless of whether those tuples are collected online or offline, and regardless of their quality. This flexibility allows Dual Q-DM to be naturally extended to offline IL with a supplementary dataset.
>
> ---
> We hope our discussions on the broad applicability of theoretical analysis and new experimental results can address your concerns satisfactorily and we kindly hope you will consider reflecting this in your score.

---

> > ### Author Rebuttal · Reviewer_aqsT · 2026-04-01
> >
> > Thank you for addressing my concerns and providing the additional experiments.
> >
> > I acknowledge that the authors have adequately addressed the following concerns:
> >
> > - **Extension of theory to infinite-horizon stationary MDPs:** I find this extension convincing. Since the infinite-horizon stationary MDP is the more standard setting for RL/IL algorithms, I recommend that the authors present the main theorems in the revised manuscript under this setting, rather than the layered MDP. The layered MDP can remain as an intermediate step or proof technique.
> >
> > - **Broader experimental coverage (3 → 5 seeds, additional Atari tasks):** The expanded evaluation strengthens the paper's impact. While it may not constitute an exhaustive benchmark, it is sufficient to substantiate the authors' claims.
> >
> > - **Experimental validation of Theorem 1:** The ablation comparing offline IQ-Learn, offline Reg IQ-Learn, and BC is a valuable addition that directly corroborates the theoretical claim.
> >
> > - **Sensitivity analysis:** The analysis confirms that Dual Q-DM is robust to hyperparameter choices, which alleviates my earlier concern.
> >
> > Overall, I find the rebuttal sufficiently convincing and have updated my rating accordingly.
> >
> > I also have a few supplementary follow-up questions to more strengthen this work:
> >
> > **1. Data efficiency with respect to expert demonstrations:** The current experiments demonstrate favorable sample efficiency compared to baselines, but the number of expert demonstrations (which is important for IL algorithms in general) used in each task is not clearly stated. Could the authors specify how many expert trajectories are used per task? Additionally, a data efficiency comparison plot (showing performance vs the number of expert demonstrations) would be a valuable addition.
> >
> > **2. Offline dataset collection for Figure R2:** How was the offline dataset collected for the offline experiments? Clarifying the data source (e.g., expert rollouts, mixed-quality transitions) would help interpret the results.
> >
> > **3. Effectiveness of Dual Q-DM in offline IL:** While the main focus of this work is online IL, the authors claim that Bellman constraints can in principle be enforced using only static offline transition tuples, suggesting that an offline version of Dual Q-DM is feasible. Demonstrating whether the offline version of Dual Q-DM can outperform offline IQ-Learn would not only validate this claim, but also substantially broaden the practical impact of this work. I understand this goes beyond the primary scope of the paper (online IL), but even a preliminary result in this direction would meaningfully strengthen the authors' argument.

---

> > > ### Author Response · Authors · 2026-04-07
> > >
> > > We thank the reviewer for their continued engagement, recognition of our rebuttal efforts, and for raising the score to "Accept." We greatly appreciate your supplementary follow-up questions aimed at further strengthening the impact of this work. Please find our detailed responses below:
> > >
> > > ### Response to Q1: Data Efficiency with Respect to Expert Demonstrations
> > >
> > > In our main experiments, we used **1 expert trajectory** for the Ant, HalfCheetah, and Walker2d tasks, and **5 expert trajectories** for the Hopper and Humanoid tasks. To systematically assess data efficiency, we conducted additional experiments varying the number of expert trajectories (1, 3, 5, and 7). **[Fig. R5](https://anonymous.4open.science/r/icml26rebuttal_dualQDM/trajnum_R5.pdf) shows that Dual Q-DM consistently outperforms all baselines across every trajectory count**, demonstrating its superior expert sample efficiency. We will include this comparison in the revised manuscript.
> > >
> > > ### Response to Q2: Offline Dataset Collection for Fig. R2
> > >
> > > To clarify, the offline setting evaluated in Fig. R2 is **pure offline imitation learning**. Under this setting, the algorithm only has access to the **expert demonstrations**. There is no other offline dataset or supplementary data (e.g., mixed-quality transitions) involved in this experiment.
> > >
> > > ### Response to Q3: Effectiveness of Dual Q-DM in Offline IL
> > >
> > > We appreciate the reviewer's insightful suggestion. As clarified in A7 of our first rebuttal, the setting discussed here is offline IL with a supplementary dataset, where the algorithm accesses an offline dataset of unknown quality in addition to the expert demonstrations—the same setting adopted by DemoDICE and RelaxDICE as mentioned in your initial review. Note that this is distinct from the pure offline IL setting discussed in Q2.
> > >
> > > Although this falls outside the scope of our paper, which focuses on standard IL with demonstrations, we conducted preliminary experiments to validate Dual Q-DM's extensibility:
> > >
> > > * **Data Source:** We used full replay data (experience replay data collected during the execution of an online RL agent) as the offline supplementary dataset, which is a standard data source for this setting.
> > > * **Implementation:** Dual Q-DM incorporates this supplementary data into the loss term for its Bellman constraints, while IQ-Learn uses it to formulate the second term in its objective (Eq. (4)).
> > > * **Results:** Our preliminary results in [Fig. R6](https://anonymous.4open.science/r/icml26rebuttal_dualQDM/offline_supplementary_R6.pdf) show that Dual Q-DM outperforms IQ-Learn in offline IL with a supplementary dataset.
> > >
> > > These results confirm that Dual Q-DM can effectively leverage offline supplementary data to enforce Bellman constraints and achieve out-of-demonstration generalization. We agree this is a promising direction, and this preliminary success indicates strong potential for extending Dual Q-DM to offline IL with supplementary data, substantially broadening the practical impact of this work.
> > >
> > > ---
> > >
> > > We will incorporate the supplementary results into the revision accordingly. Thank you again for your time and valuable feedback.

---

### Decision · Program_Chairs · 2026-04-30

**Decision:**

Accept (regular)

**Comment:**

This paper presents Dual Q-DM, a primal-dual framework for imitation learning that leverages Bellman constraints to eliminate compounding errors without adversarial training. The authors rigorously demonstrate that existing non-adversarial methods like IQ-Learn implicitly reduce to Behavioral Cloning, suffering from an $O(H^2)$ imitation gap, whereas their approach achieves an $O(H)$ bound. During the rebuttal, the authors effectively resolved major concerns by extending their proofs to infinite-horizon settings and adding comprehensive Atari evaluations. While one reviewer maintained reservations regarding baseline comparisons and state-space partitioning, the authors adequately addressed concerns on the infinite-horizon extension and convexity of the primal formulation, and clarified that the suggested baselines (SMODICE, DWBC) target an offline setting with supplementary data that is orthogonal to this paper's scope. Overall, the solid theoretical unification and robust empirical defense make this a valuable contribution to ICML.